**Improving the Sectional MOSAIC Aerosols of WRF-Chem with the Revised Gridpoint Statistical Interpolation System and Multi-wavelength Aerosol Optical Measurements: DAO-K Experiment 2019 at Kashi, near the Taklamakan Desert, northwestern China**

Wenyuan Chang[1], Ying Zhang[2], Zhengqiang Li[2], Jie Chen[3], Kaitao Li[2]

[1] State Key Laboratory of Atmospheric Boundary Layer Physics and Atmospheric Chemistry (LAPC), Institute of Atmospheric Physics, Chinese Academy of Sciences, Beijing 100029, China.

[2] State Environment Protection Key Laboratory of Satellite Remote Sensing, Aerospace Information Research Institute, Chinese Academy of Sciences, Beijing 100101, China

[3] National Meteorological Information Center, China Meteorological Administration, Beijing 100081, China

Corresponding authors:
Wenyuan Chang (changwy@mail.iap.ac.cn)
Zhengqiang Li (lizq@radi.ac.cn)

**Abstract**

The Gridpoint Statistical Interpolation data assimilation (DA) system was developed for the four-size bin sectional Model for Simulating Aerosol Interactions and Chemistry (MOSAIC) aerosol mechanism in the Weather Research and Forecasting-Chemistry (WRF-Chem) model. The forward and tangent linear operators for the aerosol optical depth (AOD) analysis were derived from WRF-Chem aerosol optical code. We applied three-dimensional variational DA to assimilate the multi-wavelength AOD, ambient aerosol scattering coefficient, and aerosol absorption coefficient, measured by the sun-sky photometer, nephelometer, and aethalometer, respectively. These were undertaken during a dust observation field campaign at Kashi in northwestern China in April 2019. The results showed that the DA analyses decreased the model aerosols' low biases; however, it had some deficiencies. Assimilating the surface particle concentration increased the coarse particles in the dust episodes, but AOD, and the coefficients for aerosol scattering and absorption, were still lower than those observed. Assimilating aerosol scattering coefficient separately from AOD improved the two optical quantities. However, it caused an overestimation of the particle concentrations at the surface. Assimilating the aerosol absorption coefficient yielded the highest positive bias in the surface particle concentration, aerosol scattering coefficient, and AOD. The positive biases in the DA analysis were caused by the forward operator underestimating aerosol mass scattering and absorption efficiency. As a compensation, the DA system increased particle concentrations excessively to fit the observed optical values. The best overall improvements were obtained from the simultaneous assimilation of the surface particle concentration and AOD. The assimilation did not substantially change the aerosol chemical fractions. After DA, the clear-sky aerosol radiative forcing at Kashi was $-10.4$ Wm$^{-2}$ at the top of the atmosphere, which was 55% higher than the radiative forcing value before DA.

**1. Introduction**

Data assimilation (DA) blends the information from observations with *a priori* background fields from deterministic models to obtain an optimal analysis (Wang et al., 2001; Bannister, 2017). With lagged emission inventories and unsatisfactory model chemistry mechanisms, there are notable discrepancies between model aerosols and observed levels (He et al., 2017; Chen L. et al., 2019). The DA technology incorporates aerosol measurements into the models to optimize emissions (Peng et al., 2017; Ma et al., 2019) and cyclically updates the background fields in forecasts. This technology effectively improves the air quality forecasts in China (Bao et al., 2019; Cheng et al., 2019; Feng et al., 2018; Hong et al., 2020; Liu et al., 2011; Pang et al., 2018; Peng et al., 2018; Xia et al., 2019a, 2019b).

Variational DA minimizes the distant scalar function that measures the misfit between model states and a set of observations in each assimilation window. An effective variational DA requires appropriate tangent linear and adjoint operators, which describe the gradient or sensitivity of the observed parameter to the control variable (Wang et al., 2001; Bannister 2017). The operator is highly dependent on the types of assimilated observations and the selection of control variables; it is also sometimes dependent on the aerosol mechanism. For $PM_{2.5}$ (particulate matter with a dynamic radius less than 2.5 μm) DA, the tangent linear operator is the ratio of the $PM_{2.5}$ concentration to each aerosol composition (Pagowski et al., 2010). For the aerosol optical depth (AOD) DA, the operator is generated through Mie theory (Liu et al., 2011; Saide et al., 2013). With the development of aerosol mechanisms and the growing body of novel aerosol observations from ground-based networks and satellites, appropriate tangent linear and adjoint operators are in demand.

The community gridpoint statistical interpolation (GSI) system (Wu et al., 2002; Purser et al., 2003a, 2003b) is often used to modify regional aerosol simulations with three-dimensional variational (3D-Var) DA. The official GSI (version 3.7 in this study) can incorporate observations of surface particulate matter concentration and AOD to constrain the aerosols simulated within the aerosol mechanism of Goddard Chemistry Aerosol Radiation and Transport (GOCART, Liu et al., 2011; Pagowski et al., 2014). The tangent linear operator and adjoint operator for AOD were determined using the Community Radiative Transfer Model (CRTM). The official GSI version incorporated the Moderate Resolution Imaging Spectroradiometer (MODIS) AOD in East Asia (Liu et al., 2011) and revealed the simultaneous DA effects of $PM_{2.5}$ and AOD in the continental United States (Schwartz et al., 2012). This GSI identified DA effects that weakened during the succeeding model's running as the model error grew (Jiang et al., 2013) and assessed the radiative forcing of the aerosols released by wildfires (Chen et al., 2014). This version was also utilized to improve air quality forecasts in China by assimilating a variety of satellite AOD data retrieved from the Geostationary Ocean Color Imager (Pang et al., 2018), Visible Infrared Imaging Radiometer Suite (Pang et al., 2018); Advanced Himawari-8 Imager (Xia et al., 2019a), and the Fengyun-3A/medium-resolution spectral imager (Bao et al., 2019; Xia et al., 2019b).

The GOCART mechanism cannot simulate nitrate and secondary organic aerosols (SOA), and the GOCART aerosol size distribution uses a bulk assumption for radiative transfer

calculation. Strictly speaking, the lack of aerosol components violates the model states'
unbiased requirements in the DA system. Lack of size-segregated aerosols may introduce a
bias in the calculation of aerosol optics. The official GSI can assimilate the surface particle
concentration from the aerosol mechanism apart from GOCART, but its AOD DA is tightly
bound with the GOCART aerosols. If one wished to use GSI to assimilate AOD for the other
aerosol mechanisms, a compromise solution was to either integrate the map of the speciated
aerosols of other mechanisms into that of the GOCART aerosols or use a simple observation
operator to convert aerosol chemical mass concentrations to AOD. For example, Tang et al.
(2017) used the official GSI to assimilate MODIS AOD with the aerosols from the
Community Multi-scale Air Quality Model (CMAQ). They incorporated the map of the 54
aerosol components of CMAQ into the five CRTM aerosols and repartitioned each CMAQ
aerosol's mass increments according to the ratios of aerosol chemical components in the
background field. This repartitioning is called the "ratio approach." Cheng et al. (2019)
assimilated the lidar extinction coefficient profiles measured in Beijing to modify the Weather
Research and Forecasting-Chemistry (WRF-Chem) Model for Simulating Aerosol
Interactions and Chemistry (MOSAIC) aerosols. They used the ratio approach to map eight
MOSAIC aerosols based on five GOCART aerosols. This mapping strategy is readily
implemented but introduces inconsistent size-segregated aerosol information (e.g.,
hygroscopicity and extinction efficiency) between the aerosol model and the DA system.
Kumar et al. (2019) analyzed the CMAQ aerosols by assimilating MODIS AOD with GSI.
Their forward operator converted aerosol chemical composition into AOD based on the well-
known IMPROVE aerosol extinction model (Malm and Hand, 2007). The IMPROVE model
predicts AOD with a linear combination of aerosol chemical masses, with the hydrophilic
particles multiplied by a tuning factor associated with relative humidity. Since building a DA
system for a new aerosol mechanism is quite technical, the official GSI for the GOCART
aerosols is a primary choice for recent aerosol DA studies (Bao et al., 2019; Xia et al., 2019;
Hong et al., 2020).
Because of the shortcomings, the official GSI has been extended to cooperate with other
aerosol mechanisms in WRF-Chem. The MOSAIC mechanism in WRF-Chem simulates
aerosol mass and number concentrations in either four- or eight-size bins. This sectional
aerosol mechanism involves nitrate chemistry and can simulate SOA with the volatility basis
set scheme. Li et al. (2013) developed a 3D-Var scheme for assimilating the surface $PM_{2.5}$ and
speciated aerosol chemical concentrations for the WRF-Chem MOSACI aerosols. Zang et al.
(2016) applied this scheme to incorporate aircraft speciated aerosols in California. They
proved that the assimilation of aircraft profile extended the DA benefit to aerosol forecast.
Saide et al. (2013) proposed a revised GSI version that performed variational DA for the
MOSAIC aerosols. The authors generated the adjoint operator code with the automatic
differentiation tool (ADT), TAPENADE v3.6. The ADT used the chain rule of derivative
calculus on the AOD source code in WRF-Chem. They assimilated multi-source AOD data
with the MOSAIC aerosols over the continental United States and found that incorporating
multi-wavelength fine-mode AOD redistributed the aerosols' particulate mass concentration
in sizes. Their GSI system also assimilated Korean ground-based and geostationary satellite
AOD datasets to improve local aerosol simulations (Saide et al., 2014, 2020). Pang et al.
(2020) developed the official GSI to work with the Modal Aerosol Dynamics Model for
Europe with the Secondary Organic Aerosol Model (MADE/SORGAM) aerosols in WRF-
Chem. They used the WRF-Chem AOD code as the forward operator to calculate the essential
aerosol optical properties and employed the CRTM adjoint operator. Because aerosols were
externally mixed in CRTM, their scheme abandoned the aerosol internal mixture in WRF-
Chem but computed the AOD of each aerosol component separately.
This study provides a solution to improve the GSI 3D-Var DA system's capability for the
sectional MOSAIC aerosols in WRF-Chem. We designed the tangent linear operator code for
AOD DA based on the WRF-Chem intrinsic aerosol optical subroutine (Fast et al., 2006). The
operator code is programmed based on the analytical equations of the tangent linear model for
AOD. As our revised GSI does not use the CRTM module, it avoids the problem of needing
to eliminate WRF-Chem aerosols characteristics (e.g., aerosol mixture state and size
distribution) to meet the CRTM input requirements. The forward and tangent linear operators
are coordinated and written in a single subroutine, coupled to the GSI at the place of invoking
CRTM for the AOD calculation. In addition to AOD DA, our tangent linear operator has two
variants to assimilate the aerosol scattering and absorption coefficients, measured using a
nephelometer and aethalometer, respectively.
This study verifies our revised GSI system's effectiveness by incorporating multi-wavelength
aerosol optical observations that were measured during an international field campaign, the
Dust Aerosol Observation-Kashi, in April 2019 at Kashi city, neighboring the Taklamakan
Desert, northwestern China. This desert is the second largest globally and is the primary
source of dust aerosols in East Asia. The dust from the desert affects the nearby Tibetan
Plateau (Ge et al., 2014; Jia et al., 2015; Zhao et al., 2020), air quality and climate in East
Asia (Huang et al., 2014), and the biogeochemical cycles in the western Pacific Ocean (Calil
et al., 2011). A successful DA analysis will help improve the local air quality forecast and
enhance our understanding of local dust storms' environmental impacts. The remainder of this
paper is organized as follows. Section 2 describes the revised GSI system, the experimental
design, and the observed data. Section 3 presents the DA results when assimilating different
observations. Section 4 discusses the impact of DA on aerosol chemical composition and
aerosol direct radiative forcing. Finally, Section 5 provides the conclusions and limitations
that need further research.
**2. Methodology and Data**
**2.1 Forecast Model**
The background aerosol fields were simulated using the WRF-Chem model version 4.0 (Grell
et al., 2005; Fast et al., 2006). The model configurations included the Purdue Lin
microphysics scheme (Chen and Sun, 2002), the unified Noah land surface model (Tewari et
al., 2004), the Yonsei University scheme for planetary boundary layer meteorological
conditions (Hong et al., 2006), and the rapid radiative transfer model for general circulation
models (RRTMG) scheme for shortwave and longwave radiation (Iacono et al., 2008). The
gas-phase chemistry was simulated using the carbon bond mechanism (Zaveri and Peters,
1999), including aqueous-phase chemistry. The aerosol chemistry was simulated using the

MOSAIC mechanism (Zaveri et al., 2008), which simulated sulfate, nitrate, ammonium, black carbon (BC), organic carbon (OC), sodium, calcium, chloride, carbonate, and other inorganic matter (OIN, e.g., trace metals and silica). The experiments did not simulate SOA to accelerate model integration. The influence of ignoring SOA was assumed to be small because of low anthropogenic and biogenic emissions in the desert's vicinity. The dust emission was simulated using the GOCART dust scheme (Ginoux et al., 2001; Zhao et al., 2010), and the dust mass was included in the OIN concentration. We performed the MOSAIC aerosol simulations with four-size bins (0.039–0.156 μm, 0.156–0.625 μm, 0.625–2.500 μm, and 2.5–10.0 μm dry diameters). The sectional aerosol data in the hourly model output were the aerosol dry mass mixing ratios of chemical compositions, aerosol number concentration, and aerosol water content. The aerosol compositions included hydrophilic particulates (i.e., $SO_4^{2-}$, $NO_3^-$, $NH_4^+$, $Cl^-$, $Na^+$) and hydrophobic particulates (i.e., BC, OC, and OIN). According to the Mie theory, we used the spherical particulate assumption and computed the aerosol optics. The aerosol compositions were internally mixed in each size bin and were externally mixed between the size bins. The internal mixing refractive index was the volume-weighted mean complex refractive index of each composition. The WRF-Chem model computed the aerosol optics at 300, 400, 600, and 999 nm and interpolated the aerosol optical parameters (AOD, SSA, asymmetry factor) to eleven shortwave lengths with Ångström exponents for the radiative transfer calculation.

**2.2 Assimilation System**

The revised GSI DA system is based on the official GSI (https://dtcenter.org/community-code/gridpoint-statistical-interpolation-gsi, Wu et al., 2002; Liu et al., 2011; Schwartz et al., 2012; Pagowski et al., 2014) version 3.7. The 3D-Var DA minimizes the cost function:

$$J(\mathbf{x}) = \frac{1}{2}(\mathbf{x} - \mathbf{x}_b)^T \mathbf{B}^{-1}(\mathbf{x} - \mathbf{x}_b) + \frac{1}{2}(H(\mathbf{x}) - \mathbf{y})^T \mathbf{R}^{-1}(H(\mathbf{x}) - \mathbf{y})$$

(1)

where $\mathbf{x}$ is the state vector composed of the model control variables; the subscript $b$ denotes that $\mathbf{x}$ is the background state vector; $\mathbf{y}$ is the vector of the observations; $H$ is the forward operator or observation operator that transfers the gridded control variables into the observed quantities at the observation locations; and $\mathbf{B}$ and $\mathbf{R}$ are the background and observation error covariance matrices, respectively.

The official GSI version only works with the GOCART aerosols for assimilating the surface-layer $PM_{2.5}$ and $PM_{10}$ (denoted as PMx in the context) concentrations and the 550 nm MODIS AOD. Our revised GSI system assimilates PMx concentrations, multi-wavelength aerosol scattering/absorption coefficients, and AOD. Figure 1 shows the workflow of our DA system. According to the AOD calculation in WRF-Chem, we can either choose the aerosol number concentration (option 1) or aerosol mass concentration (option 2) as control variables. Li et al. (2020) describes option 1. This study selects option 2 and describes it in the following subsections.

| Figure 1 |
| --- |


**2.2.1 Control Variables**

The control variables were the mass mixing ratio of each aerosol composition per size bin,
which corresponded to the WRF-Chem output data. This set differed from previous studies
that lumped aerosols per size bin as control variables (Li et al., 2013; Pagowski et al., 2014).
The control variables were six aerosol mass mixing ratios of $SO_4^{2-}$, $NH_4^+$, $NO_3^-$, OC, BC, and
OIN per size bin, a total of which were twenty-four for the four-size bin simulations. They
substantially contributed to the total aerosol mass concentrations. Chlorine and sodium had
minuscule background concentrations and remained the background values. In Kashi's case
near the desert, the OIN was predominant, accounting for 62% of $PM_{2.5}$ and 82% of $PM_{10}$.

Our design of the control variables was different from the AOD assimilation in Saide et al.
(2013), with theirs being the natural logarithm of the total mass mixing ratio per size bin,
multiplied by the thickness of the model layer. This multiplication of layer thickness
prevented many modifications for the high model layers, where aerosols were low in
concentrations. The logarithmic transformation decreased the extensive value range in the
control variables caused by multiplication. However, since the AOD value is often smaller
than one, their transformation leads to a significant negative logarithm value and an
unconstrained DA system. To handle this disadvantage, Saide et al. (2013) introduced two
weak constraints in the cost function to cut off the user-defined "extraordinarily high" and
"extraordinarily low" concentrations. They repartitioned the total mass per size bin's
increments for the composition of each aerosol using the ratio approach. In this study, neither
the logarithmic transformation nor the multiplication using layer thickness was set. Our
control variable was restricted to the WRF-Chem output variable, and the DA system changed
the composition of each aerosol per size bin, depending on the aerosol background errors.

Consistent with Pang et al. (2020), aerosol water content (AWC) was not one of the control
variables in our GSI. Otherwise, the AWC might have increased as a mathematical artifact,
contrary to the physical constraints imposed by the loading of hydrophilic particles. The
AWC was diagnosed in each outer loop according to the analyzed aerosol mass concentration
and the background relative humidity, using the WRF-Chem's hygroscopic growth scheme
coupled into the revised GSI.

**2.2.2 Tangent Linear Operator for PMx**

The $PM_{10}$ is the sum of all aerosol dry mass concentrations over the size bins, and the sum of
the first three is the $PM_{2.5}$ (Chen et al., 2019; Wang et al., 2020). The tangent linear operator
for PMx is the gradient of the PMx concentration to the aerosol chemical mass concentration
per size bin:

$$\frac{\delta[PM_x]}{\delta[C_{aer,k}]}, k = 1, \dots, n_{size}$$

267                                                                                                            (2)


where $n_{size}$ is the number of size bins and is equal to four in this study; [.] denotes the mass
concentration ($\mu g\ m^{-3}$ for PMx); $C_{aer,\ k}$ is the aerosol mass mixing ratio ($\mu g\ kg^{-1}$) of $SO_4^{2-}$,
$NO_3^-$, $NH_4^+$, OC, BC, and OIN at the $k$-th size bin. Because we did not multiply the chemical
mass with a scaling factor to represent some unknown compositions in the summation of
PMx, Eq (2) always equals one. It means that we equally distribute the PMx increment to
each aerosol composition per size bin. The $PM_{2.5}$ and $PM_{10}$ are assimilated in the same way.
When the observed fine and coarse particle concentrations are assimilated simultaneously, we
assimilate the concentrations of $PM_{2.5}$ and the coarse particulate ($PM_{10}$-$PM_{2.5}$).

**2.2.3 Forward Operator for Aerosol Optics in WRF-Chem**
We used the original forward operator in WRF-Chem for the aerosol optical parameters (Fast
et al., 2006). AOD is calculated as a function of wavelength according to Mie theory. The
columnar AOD $\tau$ is the sum of layer AOD across the $n_z$ model layers:

$$\tau = \sum_{z=1}^{n_z} \tau_z = \sum_{z=1}^{n_z} \sum_{k=1}^{n_{size}} e_{ext,z,k} \cdot n_{z,k} \cdot H_z$$

$\hfill$ (3)

where $e_{ext,z,k}$ is the extinction cross section of a single mixing particle in the $k$-th size bin at the
$z$-th model layer, $n_{z,k}$ is the aerosol number concentration, and $H_z$ is the layer thickness. At the
surface, the ambient aerosol scattering ($E_{sca}$) and absorbing ($E_{abs}$) coefficients that are
measured by the nephelometer and aethalometer, respectively, are represented as

$$E_{sca} = \sum_{k=1}^{n_{size}} e_{sca,1,k} \cdot n_{1,k}$$

$$E_{abs} = \sum_{k=1}^{n_{size}} e_{abs,1,k} \cdot n_{1,k}$$

$\hfill$ (4)

where $e_{sca,1,k}$ and $e_{abs,1,k}$ are the scattering and absorption cross section of a particle at the
surface. There is a relationship:

$$e_{ext,z,k} = e_{sca,z,k} + e_{abs,z,k}$$

$\hfill$ (5)

The extinction cross section $e_{ext,z,k}$ of a wet particle with radius $r_{wet,z,k}$ is:

$$e_{ext,z,k} = p_{ext,z,k} \cdot \pi \cdot r_{wet,z,k}^2$$

$\hfill$ (6)

where $p_{ext,z,k}$ is the extinction efficiency, given the desired mixing refractive indexes and the wet particle radius. The $p_{ext,z,k}$ is attained through the Chebyshev polynomial interpolation:

$$p_{ext,z,k} = \exp\left\{\sum_{j=1}^{n_{coef}} c_{ch}(j) \cdot c_{ext,z,k}(j)\right\}$$

(7)

where $c_{ch}$ is the coefficient of $n_{coef}$ order Chebyshev polynomials, $c_{ext,z,k}$ is the polynomial value for the particle's extinction efficiency, which is an internal mixture of all aerosol compositions (i.e., the control variables plus chlorine, sodium, and AWC). The radius is in a logarithmic transform in the AOD subroutine code to handle the broad particle size range from 0.039 μm to 10 μm. The exponential function in Eq. (7) transforms the logarithm radius back to the normal radius. The aerosol number concentration $n_{z,k,}$ and the aerosol dry (wet) mass concentration $m_{i,z,k}$ have a linkage through the dry (wet) particle radius $r_{dry,z,k}$ ($r_{wet,z,k}$) and the aerosol density $\rho_i$:

$$n_{z,k} = \sum_{i}^{n_{wet\_aer}} \frac{m_{i,z,k}}{\rho_i} \cdot \frac{3}{4\pi \cdot r_{wet,z,k}^3} = \sum_{i}^{n_{dry\_aer}} \frac{m_{i,z,k}}{\rho_i} \cdot \frac{3}{4\pi \cdot r_{dry,z,k}^3}$$

(8)

Both the dry and wet particle radius appear in the tangent linear operator. The difference between the second and the third terms in Eq (8) is whether aerosol water content is counted. $n_{wet\_aer}$ is the number of aerosol chemical composition plus aerosol water content ($n_{wet\_aer} = n_{dry\_aer} + 1$).

**2.2.4 Tangent Linear Operator Developed for AOD**

As per the forward operator in Eq. (3) in WRF-Chem, we developed the tangent linear operator for AOD, which requires the derivative of $\tau$ in Eq. (3) to the aerosol dry mass concentration (aerosol water content is not a control variable), $m_{i,z,k}$:

$$\frac{\delta\tau}{\delta m_{i,z,k}} = \frac{\delta\tau_z}{\delta m_{i,z,k}} = \frac{\delta e_{ext,z,k} \cdot n_{z,k} \cdot H_z}{\delta m_{i,z,k}} + \frac{e_{ext,z,k} \cdot \delta n_{z,k} \cdot H_z}{\delta m_{i,z,k}} + \frac{e_{ext,z,k} \cdot n_{z,k} \cdot \delta H_z}{\delta m_{i,z,k}}$$

(9)

The first term on the righthand side of Eq. (9) indicates the change in AOD as the perturbation of extinction cross section. According to Eq. (6), considering that the particle radius is constant, $\delta e_{ext,z,k}$ is represented as:

$$\delta e_{ext,z,k} = \delta p_{ext,z,k} \cdot \pi \cdot r_{wet,z,k}^2$$

(10)

where $\delta c_{ch}(j) = 0$ assuming that the particle radius is constant. This assumption simplifies the tangent linear operator and is also employed in Saide et al. (2013).


Equation (10) is expanded with the derivative of Eq. (7):

$$\delta p_{ext,z,k} = p_{ext,z,k} \cdot \left\{ \sum_{j=1}^{n_{coef}} c_{ch}(j) \cdot \delta c_{ext,z,k}(j) \right\}$$

349 (11)

By expanding $\delta c_{ext,z,k}$ in Eq. (11), we have:

$$\delta c_{ext,z,k}(j) = \delta w_{00} \cdot E_{ext,00}(j) + \delta w_{01} \cdot E_{ext,01}(j) + \delta w_{10} \cdot E_{ext,10}(j) + \delta w_{11} \cdot E_{ext,11}(j)$$

353 (12)


The four parameters of $E_{ext}$ indicate the extinction efficiencies in the Mie lookup table
surrounding the point with the desired mixing refractive indexes and the wet particle radius.
The interpolation weights $\delta w$ are determined as:

$$\delta w_{00} = (v-1)\delta u + (u-1)\delta v \qquad \delta w_{01} = (1-v)\delta u - u\delta v$$
$$\delta w_{10} = (1-u)\delta v - v\delta u \qquad \delta w_{11} = u\delta v + v\delta u$$

361 (13)


where

$$u = \frac{R_{mix} - R_{low}}{R_{up} - R_{low}} \qquad \delta u = \frac{\delta R_{mix}}{R_{up} - R_{low}}$$
$$v = \frac{I_{mix} - I_{low}}{I_{up} - I_{low}} \qquad \delta v = \frac{\delta I_{mix}}{I_{up} - I_{low}}$$

367 (14)


In Eq. (14), $R_{mix}$ and $I_{mix}$ are the aerosol volume-weighted mean real and imaginary parts of
complex refractive indices, respectively. $R_{up}$ ($I_{up}$) and $R_{low}$ ($I_{low}$) are the nearest upper and
lower limits for $R_{mix}$ ($I_{mix}$) in the Mie table. Considering $V_{wet,z,k}$ is the volume of all aerosol dry
masses plus aerosol water content, the real and imaginary parts, and their derivatives are:

$$R_{mix,z,k} = \sum_{i}^{n_{wet\_aer}} R_i \cdot \frac{m_{i,z,k}}{\rho_i \cdot V_{wet,z,k}} \qquad \delta R_{mix,z,k} = \frac{R_i}{\rho_i \cdot V_{wet,z,k}} \cdot \delta m_{i,z,k}$$
$$I_{mix,z,k} = \sum_{i}^{n_{wet\_aer}} I_i \cdot \frac{m_{i,z,k}}{\rho_i \cdot V_{wet,z,k}} \qquad \delta I_{mix,z,k} = \frac{I_i}{\rho_i \cdot V_{wet,z,k}} \cdot \delta m_{i,z,k}$$

376 (15)


where
$$V_{wet,z,k} = \sum_{i}^{n_{wet\_aer}} \frac{m_{i,z,k}}{\rho_i}$$
$\qquad$ (16)

Put Eq. (12), Eq. (13) into Eq. (11) leads to:

$\delta p_{ext,z,k} = \left[(v-1)\alpha_{sca,00} + (1-v)\alpha_{sca,01} - v\alpha_{sca,10} + v\alpha_{sca,11}\right]\delta u +$
$\qquad \left[(u-1)\alpha_{abs,00} - u\alpha_{abs,01} + (1-u)\alpha_{abs,10} + u\alpha_{abs,11}\right]\delta v$
$\qquad$ (17)
where
$\alpha_{sca,00} = p_{sca,1,k} \cdot \sum_{j=1}^{n_{coef}} c_{ch}(j) \cdot E_{sca,00}(j) \qquad \alpha_{sca,01} = p_{sca,1,k} \cdot \sum_{j=1}^{n_{coef}} c_{ch}(j) \cdot E_{sca,01}(j)$
$\alpha_{sca,10} = p_{sca,1,k} \cdot \sum_{j=1}^{n_{coef}} c_{ch}(j) \cdot E_{sca,10}(j) \qquad \alpha_{sca,11} = p_{sca,1,k} \cdot \sum_{j=1}^{n_{coef}} c_{ch}(j) \cdot E_{sca,11}(j)$
$\alpha_{abs,00} = p_{abs,1,k} \cdot \sum_{j=1}^{n_{coef}} c_{ch}(j) \cdot E_{abs,00}(j) \qquad \alpha_{abs,01} = p_{abs,1,k} \cdot \sum_{j=1}^{n_{coef}} c_{ch}(j) \cdot E_{abs,01}(j)$
$\alpha_{abs,10} = p_{abs,1,k} \cdot \sum_{j=1}^{n_{coef}} c_{ch}(j) \cdot E_{abs,10}(j) \qquad \alpha_{abs,11} = p_{abs,1,k} \cdot \sum_{j=1}^{n_{coef}} c_{ch}(j) \cdot E_{abs,11}(j)$

$\qquad$ (18)

The subscripts of *sca* and *abs* in Eq. (17) and (18) denote "scattering" and "absorption",
respectively. The first term on the righthand side of Eq. (9) is determined using Eq. (10) and
Eq. (17). The second term on the righthand side of Eq. (9) indicates the linkage of the aerosol
number and mass concentrations. It is the derivative of the dry particle in Eq. (8) by assuming
a constant radius:

$\delta n_{z,k} = \dfrac{3 \cdot \delta m_{i,z,k}}{4\pi \cdot r_{dyr,z,k}^3 \cdot \rho_i}$
$\qquad$ (19)

The third term on the righthand side of Eq. (9) contains the layer thickness's derivative to the
concentrations in this layer. It indicates that the light attenuation length is based on per unit
concentration, which can be intuitively represented by the ratio of layer thickness to the
aerosol mass concentration in this layer. Putting Eq. (10) and Eq. (19) into Eq. (9), we have
the original formula of the tangent linear operator for AOD for the aerosol dry mass
concentration:


$\dfrac{\delta \tau}{\delta m_{i,z,k}} = \dfrac{\delta \tau_z}{\delta m_{i,z,k}} = \dfrac{\delta e_{ext,z,k} \cdot n_{z,k} \cdot H_z}{\delta m_{i,z,k}} + \dfrac{e_{ext,z,k} \cdot \delta n_{z,k} \cdot H_z}{\delta m_{i,z,k}} + \dfrac{e_{ext,z,k} \cdot n_{z,k} \cdot \delta H_z}{\delta m_{i,z,k}} =$
$$\left\{\left[(v-1)\alpha_{sca,00}+(1-v)\alpha_{sca,01}-v\alpha_{sca,10}+v\alpha_{sca,11}\right]\cdot\frac{\pi\cdot r_{wet,z,k}^2\cdot R_i\cdot n_{z,k}\cdot H_z}{\rho_i\cdot V_{wet,z,k}\cdot\left(R_{up,z,k}-R_{low,z,k}\right)}\right.$$
$$+$$
$$\left[(u-1)\alpha_{abs,00}-u\alpha_{abs,01}+(1-u)\alpha_{abs,10}+u\alpha_{abs,11}\right]\cdot\frac{\pi\cdot r_{wet,z,k}^2\cdot I_i\cdot n_{z,k}\cdot H_z}{\rho_i\cdot V_{wet,z,k}\cdot\left(I_{up,z,k}-I_{low,z,k}\right)}+$$
$$\frac{3e_{ext,z,k}\cdot H_z}{4\pi\cdot r_{dry,z,k}^3\cdot\rho_i}+\frac{e_{ext,z,k}\cdot n_{z,k}\cdot H_z}{m_{i,z,k}}\Big\}\cdot\beta$$

$$(20)$$

where $\beta$ changes the mass unit from µg kg$^{-1}$ to µg m$^{-3}$. The last righthand term in Eq. (20)
may not have a quick convergence in the DA outer loops because the aerosol mass
concentration $m_{i,z,k}$ in the denominator often has a low bias, introducing an error into the
operator. The error is further amplified by the layer thickness $H_z$ in the numerator. Thus, Eq.
(20) cannot lead to a stable analysis. For this reason, we changed the tangent linear operator to
account for the columnar mean aerosol extinction coefficient, which is described as follows:

$$\frac{\delta\overline{(e_{ext}\cdot n)}}{\delta m_{i,z,k}}=\frac{H_z}{\sum H_z}\cdot\frac{\delta\left(e_{ext,z,k}\cdot n_{z,k}\right)}{\delta m_{i,z,k}}=\frac{H_z}{\sum H_z}\cdot\left[\frac{\delta e_{ext,z,k}\cdot n_{z,k}}{\delta m_{i,z,k}}+\frac{e_{ext,z,k}\cdot\delta n_{z,k}}{\delta m_{i,z,k}}\right]=$$
$$\left\{\left[(v-1)\alpha_{sca,00}+(1-v)\alpha_{sca,01}-v\alpha_{sca,10}+v\alpha_{sca,11}\right]\cdot\frac{\pi\cdot r_{wet,z,k}^2\cdot R_i\cdot n_{z,k}}{\rho_i\cdot V_{wet,z,k}\cdot\left(R_{up,z,k}-R_{low,z,k}\right)}\right.$$
$$+$$
$$\left[(u-1)\alpha_{abs,00}-u\alpha_{abs,01}+(1-u)\alpha_{abs,10}+u\alpha_{abs,11}\right]\cdot\frac{\pi\cdot r_{wet,z,k}^2\cdot I_i\cdot n_{z,k}}{\rho_i\cdot V_{wet,z,k}\cdot\left(I_{up,z,k}-I_{low,z,k}\right)}+$$
$$\frac{3e_{ext,z,k}}{4\pi\cdot r_{dry,z,k}^3\cdot\rho_i}\Big\}\cdot\beta\cdot\frac{H_z}{\sum H_z}$$

$$(21)$$

In Eq. (21), the operator is based on the extinction coefficient at each layer, weighted by the
layer thickness normalized to the total model layer thickness. Correspondingly, the AOD
observations and AOD observation error are divided by the total layer thickness at the
observation location. Note that the dry ($r_{dry,z,k}$) and wet ($r_{wet,z,k}$) particle radiuses are both
present in Eq (21). Because aerosol water content is not a control variable, $r_{dry,z,k}$ is used in Eq
(19) and appears in Eq (21). Aerosol water content participates in the computation of internal
mixing refractive indexes, and $r_{wet,z,k}$ is also present in Eq (21). Equation (21) is the final
tangent linear operator for AOD DA in this study.

**2.2.5 Tangent Linear Operator Developed for Surface Aerosol Attenuation Coefficients**
The aerosol scattering and absorption coefficients measured by the nephelometer and
aethalometer, respectively, are similar to the aerosol extinction coefficient at the surface in
Eq. (21). Neither of the two coefficients addresses the layer thickness. The operator for the
aerosol scattering coefficient measured by a nephelometer is described as follows:

$$\frac{\delta(e_{sca,1,k} \cdot n_{1,k})}{\delta m_{i,1,k}} = \{[(v-1)\alpha_{sca,00} + (1-v)\alpha_{sca,01} - v\alpha_{sca,10} + v\alpha_{sca,11}]$$

$$\cdot \frac{\pi \cdot r_{wet,1,k}^2 \cdot R_i \cdot n_{1,k}}{\rho_i \cdot V_{wet,1,k} \cdot (R_{up,1,k} - R_{low,1,k})} + \frac{3e_{sca,1,k}}{4\pi \cdot r_{dry,1,k}^3 \cdot \rho_i}\} \cdot \beta$$

$$(22)$$


The symbols have the same meaning as before, and the subscript one in Eq. (22) denotes the
surface layer. The operator for the aerosol absorption coefficient measured by an aethalometer
is

$$\frac{\delta(e_{abs,1,k} \cdot n_{1,k})}{\delta m_{i,1,k}} = \{[(u-1)\alpha_{abs,00} - u\alpha_{abs,01} + (1-u)\alpha_{abs,10} + u\alpha_{abs,11}]$$

$$\cdot \frac{\pi \cdot r_{wet,1,k}^2 \cdot I_i \cdot n_{1,k}}{\rho_i \cdot V_{wet,1,k} \cdot (I_{up,1,k} - I_{low,1,k})} + \frac{3e_{abs,1,k}}{4\pi \cdot r_{dry,1,k}^3 \cdot \rho_i}\} \cdot \beta$$

$$(23)$$


As shown in the operators, the aerosol mass concentrations' gradients rely on the aerosol
number concentration; meanwhile, the number concentration is estimated according to the
mass concentration and the particle radius. The two concentrations are intertwined in the DA
system, indicating the operator's nonlinearity. This nonlinearity is handled with a succeeding
minimization of the cost function within the GSI. The cost function is first minimized with the
number concentration in the background field, and the number concentration is updated with
the first analyzed aerosol mass concentrations. In the second minimization, the first analysis's
number concentration constructs a new operator value, resulting in a new analysis of mass
concentrations. This iterative process is denoted as the "outer loop," which is repeated several
times to attain the final analysis (Massart et al., 2010). We set ten maximum iterations in the
experiments. The cost function in most analyses reaches the minimum in two or three outer
loops. The WRF-Chem AOD code is coupled into the GSI subroutine at the place of invoking
CRTM. The tangent linear operators of Eq. (21), Eq. (22), and Eq. (23) are simultaneously
determined in the subroutines, which are cyclically invoked in the outer loops.

**2.2.6 Aerosol Complex Refractive Indexes in GSI**
Table S1 in the supplementary document shows the complex refractive indexes for each
aerosol chemical composition in the revised GSI. The refractive indexes are for eleven
wavelengths, including four for CE318, three for nephelometer, three for aethalometer, and
one for 550 nm MODIS AOD (not assimilated in this study). The real parts of refractive
indexes of sulfate, nitrate, and ammonium are similar and refer to Toon et al.'s (1976) data.
The real part is 1.53 at 440 nm and decreases to 1.52 at 1020 nm. The refractive indexes of
OC and BC are constant across the wavelengths, being 1.55–0.001$i$ for OC (Chen and Bond,
2010) and 1.95–0.79*i* for BC (Bond and Berstrom, 2006). The dust refractive index's real part
is a constant value of 1.54 (Zhao et al., 2010). The dust refractive index's imaginary part
depends on the dust mineralogy, size distribution, and shape associated with the dust sources.
Cheng et al. (2006) reported the desert dust refractive index in winter and spring at
Dunhuang, a city adjacent to the Taklamakan desert's northeast side. Their imaginary part
value was approximately in the ranges of 0.0008 to 0.0028 at 440 nm, 0.0006 to 0.0030 at 670
nm, 0.0005 to 0.0036 at 870 nm, and 0.0005 to 0.0040 at 1020 nm (See Figure 9 in their
paper). Di Biagio et al. (2019) retrieved the dust's imaginary part in the Taklimakan desert's
north edge (41.83°N, 85.88°E). Their dust imaginary part decreased from 0.0018±0.0008 at
370 nm to 0.0005±0.0002 at 950 nm, much lower than the generic values in climate models.
The imaginary part's retrieval uncertainty is related to the iron oxide in dust samples, the
cutoff coarse particle size (<10 μm in Di Biagio et al., 2019), and the spherical particle
assumption applied in the retrieval algorithm. Here, we admit the high uncertainty and use the
imaginary part following the generic model values (Table S1), which are higher than the
upper data limits of Di Biagio et al. (2019) and are close to the values of Cheng et al. (2006).
The desert dust has a stronger absorption at shortwave wavelengths. The refractive index of a
wavelength without exact literature data uses the nearby wavelength's data in the literature.
Aerosol density is necessitated to compute aerosol optical parameters in the AOD forward
operator and construct our tangent linear operator. The supplement also shows the aerosol
density (Table S2) that follows the data in Barnard et al. (2010).
**2.3 Background Error Covariance (BEC)**
Many aerosol DA studies used the National Meteorological Center (NMC) method (Parrish
and Derber, 1992) to model the BEC matrix. The NMC method uses long-term archived
weather data created in forecast cycles. It computes the statistical differences between two
forecasts with different leading lengths (e.g., 24 h and 48 h), but which are valid at the same
time. The NMC method is workable because solving global weather forecasts is an initial
value problem of mathematical physics. A slight difference in the initial atmospheric state
would lead to a substantially different prediction because of the chaos in the atmosphere.
However, a regional model is a boundary value problem (Giorgi and Mearns, 1999). As the
regional model runs, the influence of the initial conditions becomes weak, while lateral
boundary conditions always take effect. The reanalysis data that drive the paring regional
model simulations are similar and leads to a limited difference between the paring
simulations. The NMC method's BEC would therefore underestimate the aerosol error in
WRF-Chem. Kumar et al. (2019) assimilated AOD in the contiguous United States based on
the NMC method's BEC. They perturbed the background emissions by adding the gridded
mean differences of four emission inventories. Their BEC accounting for meteorology and
emissions uncertainties reduced the AOD bias by 38%, superior to 10% bias reduction,
counting the meteorology uncertainty alone.
Some aerosol DA studies have created background error variance using the ensemble
simulations by randomly disturbing model lateral boundary conditions and surface emissions
(Peng et al., 2017; Ma et al., 2020). The ensemble experiments better represent the model
error but significantly increase the computational burden. Here, we used the variance of the

background hourly aerosol concentrations in April to represent the background error variance. The rationale of this approach is that the Tarim Basin acts as a "dust reservoir" and traps dust particles for a period before the dust being carried long-distance by wind (Fan et al., 2020). The model bias in dust concentration is correlated with aerosol concentration variation as the weather fluctuates. The model bias is small on clear days when the aerosol concentration is low. The bias is large when the concentration is high on heavily-polluted days. The mean aerosol concentration correlated positively with the aerosol variation. Using aerosol concentration variance to represent the aerosol error prioritizes DA modification of aerosols having high background mean concentrations. It was similar to the way in Sič et al. (2016), which set a percentage of the first guess field for the background error variance.

We calculated the background error statistics, including the aerosol standard deviation and the horizontal and vertical correlation length scales, using the GENerate the Background Errors (GEN-BE) software (Descombes et al., 2015), based on the one-month hourly aerosol concentrations in WRF-Chem. We obtained the statistics of four static BECs for the four DA analysis hours (i.e., 0000, 0600, 1200, and 1800 UTC), respectively. The DA procedures for the four analysis times a day in April 2019 repeatedly use the background error statistics at the corresponding analysis time.

A usual strategy to enrich the samples of model results for the error statistics is to gather model grid points with similar atmosphere characteristics, referred to as "binning." The statistics are spatially averaged over the binned grid points. The GEN_BE default strategy for GSI is latitude-binning, which creates a latitude-dependent error correlation function (Figure 2a). The latitude binning is generally used for latitude flow dependency and works for large and global domains (Wu et al., 2002). However, we found that using the latitude-binning strategy overestimated the PMx concentration when assimilating aerosol optical observations. One reason for this overestimation was related to the model's low bias in particle extinction efficiency, as discussed in Section 3.3. Another plausible reason is related to the background model error's vertical profile. The maximum dust error in the desert occurred at the surface (Figure 2e) because of the local dust emissions, while the maximum error at Kashi was at the dust transporting layer above the surface (Figure 2d). Owing to the Taklamakan Desert's vast extent, the latitude-binning suppressed the local error characteristics at Kashi and led to a vertical error profile (Figure 2c) similar to that over the desert (Figure 2e).

For this reason, we used the standard deviation of the control variable at each model grid to replace the latitude-binning standard deviation. The horizontal and vertical correlation length scales were calculated based on the latitude-binning data. Figure 3 shows the background error statistics generated by the GEN_BE software, which provided the input to the GSI. The OIN component showed high background errors in the third and fourth particle sizes at the transporting layer above the surface (Figure 3f). The aerosol compositions related to anthropogenic emissions (i.e., sulfate, nitrate, ammonium, OC, and BC, referred to here as 'anthropogenic aerosols') that had maximum errors in the second particle size, with the greatest vertical error at the surface. The background error for OIN composition was a factor

of 2–3 higher than that for anthropogenic aerosols because of the high background dust
concentration.
The horizontal and vertical correlation length scales determine the range of observation
innovations spreading from the observation locations. The horizontal influences had small
changes in altitude within the lowest 15 model layers (below a height of ~5 km above
ground), indicating that the dust transport layer was well-mixed in the lower atmosphere. This
deep dust layer was consistent with Meng et al. (2019). They showed that the dust in spring
was vertically mixed in a thick boundary layer to a height of 3–5 km in the Tarim Basin. The
vertical correlation length scales first increased from low values at the surface to high values
at ~2.5 km in height (for the 8–9 layers), indicating upward aerosol flux in windy days. The
vertical correlation length scale quickly decreased from the maximum value with a further
altitude rise. The maximum correlation length above the ground indicates a laminar air motion
during the dust storm.
Because the background model error per size bin is independent, the DA modification of an
aerosol concentration would be quite large in a single size bin with the maximum background
error (e.g., the OIN in the fourth particle size). To avoid the excessive accumulation of
increment, we added a one-dimensional recursive filter for the background covariances of
control variables across the size bins, with a correlation length scale of four bin units.

Figure 2, Figure 3

**2.4 Observational Data and Errors**
The Dust Aerosol Observation–Kashi field campaign was performed at Kashi from 0000
UTC 25 March to 0000 UTC 1 May 2019. The site was placed in the Kashi campus of the
Aerospace Information Research Institute, Chinese Academy of Sciences (39.50°N, 75.93°E;
Li et al., 2018), about 4 km in the northwest to the Kashi city. The site aerosol observations
included: (1) the multi-wavelength AOD measured by the sun-sky photometer (Cimel
CE318); (2) the multi-wavelength aerosol scattering and absorption coefficients at the surface,
measured with a nephelometer (Aurora 3000) and aethalometer (Magee AE-33), respectively;
and (3) the hourly $PM_{2.5}$ and $PM_{10}$ observations, measured with a METONE BAM-1020
continuous particulate monitor. All the instruments were deployed at the roof of a three
stories height building on the campus. Please refer to Li et al. (2020) for more details about
the field campaign.
Table 1 summarizes the observation periods, the aerosol optical data's wavelengths, and the
observation errors. The multi-wavelength data of each type of optical observation were
assimilated simultaneously. The observation errors of PMx are handled in the conventional
way (Schwartz et al., 2012; Chen et al., 2019), which contains the measurement error ($e_1$) and
the representation error ($e_2$). The measurement error is the sum of a baseline error of 1.5 μg
$m^{-3}$ and 0.75% of the observed PMx concentration. The representation error is the
measurement error multiplied by the half-squared ratio of the grid spacing to the scale
distance. The scale distance denotes the site representation in GSI and has four default values
of 2, 3, 4, and 10 km, corresponding to the urban, unknown, suburban, and rural sites. We
used 3 km for the scale distance in this study. As we had a single site in Kashi, it is difficult to
estimate the site representation error. Since the DA analysis was based on the inner model
domain with a horizontal resolution of 5 km, close to the site distance to the Kashi urban area,
we assumed the aerosol optical measurement had good representativeness of the model grid.
The observation error of CE318 AOD took the AERONET AOD uncertainty of 0.01 in cloud-
free conditions (Holben et al., 1998). The AOD observational error was further divided by the
total model layer thickness in GSI. It is difficult to determine instrumental errors in
nephelometers and aethalometers, and we set their instrumental errors to 10 Mm$^{-1}$, equivalent
to the magnitude of the Rayleigh extinction coefficient. The observational errors were
uncorrelated, with **R** being a diagonal matrix.

| Table 1 |
| --- |


**2.5 Experimental Design**
The WRF-Chem simulations were configured in a two-nested domain centered at 82.9 °E,
41.5 °N. The coarse domain was a 120×100 (west-east × north-south) grid with a horizontal
resolution of 20 km covering the Taklamakan Desert, and the fine domain was an 81×61 grid
with a resolution of 5 km, focusing on Kashi and environs (Figure 4a). Both domains had 41
vertical levels extending from the surface to 50 hPa. The lowest model layer at the site was
approximately 25-meter height from the ground. The two domains were two-way coupled.
The coarse domain covered the entire dust emission source, providing dust transport fluxes at
the fine domain's lateral boundaries. The aerosol radiative effect was set to provide feedback
on the meteorology. The indirect effect of aerosols was not set in the experiments. Initial and
lateral boundary meteorological conditions for WRF-Chem were the one-degree resolution of
the National Centers for Environmental Prediction Final Analysis data created by the Global
Forecast System model. The meteorological lateral boundary conditions for the coarse domain
were updated every six hours and were linearly interpolated between the updates in WRF-
Chem. We did not set the chemical boundary conditions for the coarse domain. The
Multiresolution Emission Inventory of China (MEIC) for 2010 (www.meicmodel.org)
provided anthropogenic emission levels. The yearly emission differences in 2010-2019 may
bias the aerosol chemical simulation, but this bias is hard to be quantified as lack of aerosol
chemical observations in this city. As the significant pollutant at Kashi is dust, we just ignore
the model uncertainties due to the yearly differences in anthropogenic emission inventories.
The biogenic emission levels were estimated online using the Model of Emissions of Gases
and Aerosols from Nature (Guenther et al., 2006). Wildfire emissions were not set in the
experiments.

We conducted a one-month WRF-Chem simulation for April 2019, starting at 0000 UTC on
27 March and discarding the first five days for spin-up. The revised GSI system modified the
aerosols in the fine domain at 0000, 0600, 1200, and 1800 UTC each day starting from 0000
UTC 1 April until the end of the month. We assimilated the observations four times a day
because the reanalyzed meteorological data were available for the four-time slices, facilitating
the model restarting from the DA analyses. The hourly PMx observations were assimilated at
the exact time of analysis. The observed AOD and aerosol scattering/absorption coefficients
were assimilated when they fell within 3 hours before the time of analysis. Table 2 shows the
DA experiments, in which the multi-wavelength AOD (440 nm, 675 nm, 870 nm, and 1020
nm) in DA_AOD, the aerosol scattering coefficients (450 nm, 525 nm, and 635 nm) in
DA_Esca, and the aerosol absorption coefficients (470 nm, 520 nm, and 660 nm) in DA_Eabs
were assimilated simultaneously in each experiment. The literal meanings of the experimental
names denote the observations that were assimilated. To study the impact of DA on aerosol
direct radiative forcing (ADRF), we modified the WRF-Chem code to calculate the shortwave
irradiance with and without aerosols at each model integration step. The modified WRF-
Chem model restarted from each DA analysis and ran to the next analysis time. Each running
performed the radiation transfer calculation twice, and each calculation saw the aerosols and
clean air, respectively. The irradiance difference between the two pairing calls was aerosol
radiative forcing. Section 4.2 shows the DA effects on the clear-sky ADRF values.

| Table 2, Figure 4 |
| --- |

**3. Results**
**3.1 Evaluation of Control Experiment**
Table 2 shows the monthly mean values and correlations between the observed data and the
model results. The statistical values were based on the pairing data between the model results
and the observations. Figures 6 show the surface PMx concentrations, aerosol scattering
coefficients, and AOD when assimilating the observations at 0000, 0600, 1200, and 1800
UTC each day in April.
Kashi is in the junction between the Tian Shan Mountains to the west and the Taklamakan
Desert to the east (Figure 5a). In the Tarim Basin, the prevailing surface wind is easterly or
northeasterly, which raises dust levels and carries the particles westward (Figure 5b). An
intense dust storm hit the city at noon on 24 April 2019, with a peak $PM_{10}$ concentration
exceeding 3,000 $\mu g\ m^{-3}$. The dust storm traveled across the northern part of the desert and
carried the dust particles to Kashi and the mountainous area (Figure 5c, d). A few mild dust
storms occurred at Kashi on April 3–5, April 8–11, and April 14–17, and the maximum $PM_{10}$
concentrations were in the range of 400–600 $\mu gm^{-3}$. The time series of $PM_{2.5}$, aerosol
scattering/absorption coefficient, and AOD showed patterns similar to those for $PM_{10}$ (Figure
690 6).
WRF-Chem captured the main dust episodes but significantly underestimated the aerosols at
Kashi (Table 2). The monthly mean background concentrations of $PM_{2.5}$ and $PM_{10}$ were 17%
and 41% lower than the observed values, respectively, with a low correlation ($R < 0.3$). The
simulated dust storm on 24 April was a mild dust event and had a maximum $PM_{10}$ of ~300 $\mu g$
$m^{-3}$, one-tenth of the observed value. The model underestimates the aerosol
scattering/absorption coefficients and AOD by 40–70%.
The OIN component accounted for the model bias in $PM_{10}$ on dusty days. Zhao et al. (2020)
proposed that the GOCART scheme reproduced dust emission fluxes under weak wind
erosion conditions but underestimated the emissions in conditions of strong wind erosion. We
did not assimilate meteorology. The model bias in the surface wind could introduce an error
in dust emission and a bias in the number of dust particles entering the city. In the non-dust
days with the $PM_{10}$ lower than the $25^{th}$ percentile $PM_{10}$ in April, the model $PM_{2.5}$ on average
accounted for 60% of the observed data levels. The $PM_{2.5}$ low bias could be due to the lack of
SOA chemistry in our experiments and the low emission bias in the residential sector, a major
source of anthropogenic emissions for $PM_{2.5}$, BC, and OC in the developing western area. The
residential sector accounts for 36–82% of the primary particle emissions, according to the
MEIC emission inventory (Li et al., 2017), and is the primary source of uncertainty in
anthropogenic emissions inventories in China.

| Figure 5 |
| --- |


**3.2 Assimilating $PM_{2.5}$ and $PM_{10}$ Concentrations**
Simultaneous assimilation of the observed PMx (DA_PMx) improved both the fine and
coarse particle concentrations, with a substantial increase in the third and fourth particle sizes
of the OIN composition (Figure 8f). The analyzed monthly mean $PM_{10}$ increased to 329.3 µg
$m^{-3}$, with a high correlation of 0.99. The analyzed monthly mean $PM_{2.5}$ was improved to 89.3
µg $m^{-3}$, although it was still lower than the observed levels, with a high correlation of 0.89.
The low bias in $PM_{2.5}$ and the high bias in $PM_{10}$ in the analyses were mainly in the dust storm
on 24–25 April (Figure 6a, d).

Applying the inter-size bin correlation length caused the interlinked analyses of $PM_{2.5}$ and
$PM_{10}$. In the desert area, the coarse and fine dust is readily affected by BEC's magnitude of
the fourth size-bin OIN (oin_a04). We performed a few sensitivity tests decreasing the BEC
of oin_a04 by 10% each time until the BEC was 30% of its original value. The magnitude of
30% of oin_a04 was comparable to the magnitude of the third size-bin (oin_a03) OIN's
background error. As shown in Table S3, because the oin_a04's BEC reduction relaxes the
constraint on the coarse particle, the $PM_{10}$ bias becomes more negative along with the
decrease in on_a04's BEC. The $PM_{2.5}$ bias meanwhile becomes more positive.
Correspondingly, the ratio of $PM_{2.5}$ to $PM_{10}$ was increased to 0.33 with 30% of oin_a04's
BEC, higher than the observed value of 0.28. According to these experiments, the original
BEC of oin_a04 is a reasonable tradeoff. The inter-size bin correlation length tunes the cross
size-bin modifications and also affects the analyses of $PM_{2.5}$ and $PM_{10}$. The experiment's
correlation length is a little bit arbitrary, but we found that our DA analyses were not very
sensitive to the inter-size bin correlation length.

According to our BEC modeling strategy, the DA system preferentially modified the coarse
particle concentrations because of the coarse particles' high background model error.
Intuitively, our modification that mainly focused on the highest concentration of coarse
particles was reasonable. It decreased the model biases by raising the heaviest loading
aerosols. As a result, the ratio of $PM_{2.5}$ to $PM_{10}$ decreased from 0.39 in the background to 0.27
in DA_PMx, approaching the observed ratio of 0.28. Such improvement was consistent with
the correction required to the model desert dust in literature. Kok et al. (2011) found that
regional and global circulation models underestimate the fraction of emitted coast dust (>~5
µm), overestimates the fraction of fine dust (<2µm diameter). Adebiyi and Kok (2020)
claimed that too rapid deposition of coarse dust out of the atmosphere accounts for the
missing coarse dust in models. According to Kashi's AOD between 440 nm and 1020 nm, the
observed Ångström exponent (AE) was 0.18, while the background value was 0.54 (Table 3),
showing too many fine particles in the background field. DA_PMx reduced the AE value to
0.30, a little improvement but not sufficient.

As the particle concentration increased, the 635 nm aerosol scattering coefficient in DA_PMx
moderately increased to 170.4 $Mm^{-1}$, still lower than the observed level of 231.5 $Mm^{-1}$, with a
high correlation of 0.89. The scattering AE was 1.32 in the background and decreased to 0.96
(Table 3), indicating a more reasonable wavelength dependence of the coarse particles'
scattering in the analysis. The analyzed 660 nm absorption coefficient had a small
improvement, which was 15.8 $Mm^{-1}$, 67% lower than observed levels, with a correlation of
0.42. There was no improvement in absorption AE, which increased to 1.84 in DA_PMx, far
higher than the observation value of 1.65. The analyzed 870 nm AOD showed a monthly
mean value of 0.38 in DA_PMx, 42% lower than observed levels, with a low correlation of

761    0.35.


Figure 9a shows the diurnal concentrations of $PM_{10}$ in the analyses in April. The observed
$PM_{10}$ showed a substantial variation at 1800 UTC, the local midnight. This substantial
nocturnal variation was partly owing to the dust storm that started on 24 April and ended the
next day. This midnight variation was also related to a nocturnal low-level jet. Ge et al.
(2016) pointed out a nocturnal low-level jet at the height of 100–400 m, with a wind speed of
4–10 m $s^{-1}$ throughout the year in the Tarim Basin. They stressed that the low-level jet broke
down in the morning, transporting its momentum toward the surface, and increased dust
emissions. The nocturnal low-level jet increased the possibility of dust particles moving
towards the city at night, causing a high $PM_{10}$ variation at 1800 UTC. The diurnal changes in
the DA analyses followed the observed levels but had higher mean values.

**3.3 Assimilating AOD**
Assimilating AOD (DA_AOD) improved the monthly mean of 870 nm AOD to 0.59,
approaching the observed value of 0.66, with a high correlation of 0.98 (Figure 6h). The
monthly mean $PM_{2.5}$ was improved to 92.6 µg $m^{-3}$, quite close to the observed level of 91 µg
$m^{-3}$, but the analyzed $PM_{10}$ was 541.7 µg $m^{-3}$, 68% higher than the observed value. The DA
system improved the AOD at the price of deteriorating the data quality of surface coarse
particle concentrations. Such surface particle overestimations have been reported in previous
studies (Liu et al., 2011; Ma et al., 2020; Saide et al., 2020). As a result, the ratio of $PM_{2.5}$ to
$PM_{10}$ reduced to 0.17 in DA_AOD, which was too far compared with the observed ratio of
0.28. The overestimation of aerosol mass concentration also inclines to raise
scattering/absorption coefficients. The analyzed 635 nm scattering coefficient in DA_AOD
increased to 222.6 $Mm^{-1}$, slightly lower than the observed value. The analyzed 660 nm
absorption coefficient slightly increased to 17.0 $Mm^{-1}$, 64% lower than the observed value.

The scattering and absorption AE values in DA_AOD had the responses as those in
DA_PMx. As shown in Table 3, the scattering AE decreased to 0.44 in DA_AOD, which was
slightly better than the AE value of 0.96 in DA_PMx. On the contrary, the absorption AE
increased to 1.97 in DA_AOD, far deviating from the observed value. The analysis fit to the
aerosol scattering overwhelmed the fit to the aerosol absorption. The AE based on AOD was
reduced to –0.01 in DA_AOD, in line with the decrease in DA_PMx, but the reduction in
DA_AOD was much lower than the observation value of 0.18.
Table 4 shows the ratios of the AOD and aerosol scattering/absorption coefficients to the
surface $PM_{10}$ concentrations. The ratio of AOD to $PM_{10}$ in the background model result was
one-third of the observed levels. The observed mass scattering coefficient (Esca/$PM_{10}$) was
1.05 $Mm^{-1}$ $\mu g^{-1}$ $m^3$, while the background value was only 0.65 $Mm^{-1}$ $\mu g^{-1}$ $m^3$. DA_AOD
did not eliminate the low bias but lowered the ratio to 0.51 $Mm^{-1}$ $\mu g^{-1}$ $m^3$. The same thing
occurred for Eabs/$PM_{10}$, which was 0.09 in the background and 0.05 in DA_AOD, much
lower than the observed value of 0.25. Figure 10 shows these mean ratios at the other
wavelengths. The low bias in AOD/$PM_{10}$ was comparable at each wavelength with a slightly
stronger low bias in short wavelengths (Figure 10a). The ratios' low biases indicated the low
scattering and absorption efficiencies, and the DA system overestimated the $PM_{10}$ to fit the
observed AOD data.
We computed the surface single scattering albedo (SSAsrf) with the 525 nm scattering
coefficient and 520 nm absorption coefficient. We did not use the Ångström exponent to
interpolate the scattering/absorption coefficients to a similar wavelength because the AE itself
had a large model bias even after DA (Table 3). The observed SSAsrf value was 0.78,
indicating an emphatic absorption particle, probably due to the mixture of anthropogenic
black carbon and natural desert dust in the local air. The model background SSAsrf was 0.86.
The DA analyses gave even higher SSAsrf (0.88 to 0.9).
The low bias in mass scattering/absorption efficiency is related to the aerosol optical module
based on Mie theory in WRF-Chem. First, the simulations used four-size bin particle
segregation. This coarse size representation aggregated many aerosols in the accumulation
mode (Figure 8f). Because small particles have a strong light attenuation capability, according
to the Mie theory, too many coarse particles would not effectively increase the AOD. Saide et
al. (2020) linked the aerosol optics to the size bin representation (from 4 to 16 bins) for hazes
in South Korea. They showed that WRF-Chem underestimated the dry aerosol extinction, and
the underestimation could be relieved when using a finer size bin than four. Okada and Kai
(2004) found that the dust particle radius in the Taklamakan Desert was in the range of 0.1–4
μm, indicating the dominant fine-mode particles in the desert.
Second, the dust particles are irregular in shape (Okada and Kai, 2004), while the spherical
particle is a common assumption for the aerosol optics in the Mie theory in current models,
which is an essential source of uncertainty in the forward operator of WRF-Chem when the
assumption of spherical particles for dust fails. The irregular morphology has a significant
influence on the dust simulation. Okada et al. (2001) found that the aspect ratio (the ratio of

the longest dimension to its orthogonal width) of the mineral dust particles (0.1-6 µm) in China's arid regions exhibited a median of 1.4. Dubovik et al. (2006) suggested the aspect ratio of ~1.5 and higher in desert dust plumes. Kok et al. (2017) found that the dust' sphericity assumption underestimated dust extinction efficiency by ~20–60% for the dust particle larger than 1µm. Tian et al. (2020) found that using a dust ellipsoid model could increase the concentration of coarse dust particle (5-10 µm) by ~5% in eastern china and ~10% in the Taklimakan area because of the decrease in gravitational settling, comparing with the simulations with dust sphericity model. Nevertheless, the aspect ratio of the spheroid dust is uncertain. Even after applying the spheroidal approximation, Soorbas et al. (2015) found that the model underestimated 550 nm aerosol scattering and backscattering values by 49% and 11% because of the uncertainties in particle's axial ratio, complex refractive index, and the particle size distribution. To date, the assumption of spherical particles has been widespread in models (including WRF-Chem) for computational efficiency. The impact of dust morphology on DA deserves further investigation.

To reduce the overestimate in PMx concentrations, we set the gridded standard deviation in place of the latitude-binning standard deviation, as discussed in Section 2.3. Figure 11 shows the analyzed vertical profiles of PMx concentrations. Higher $PM_{10}$ concentrations were shown in the low atmosphere than at the surface for the assimilation experiments. These vertical error profiles decreased the surface $PM_{10}$ particles and increased the $PM_{2.5}/PM_{10}$ ratio. The BEC tuning was not sufficient to increase the mass extinction efficiency to the observed value. The mass extinction efficiency in the analysis was almost equivalent to the background value (Table 4). Finer aerosol size representation and a better advanced aerosol optical calculation for dust could be considered solutions.

Assimilating the AOD seems to increase the diurnal variation in the DA analyses, but this variation was not conclusive since different amounts of AOD data for DA at 0000, 0600, and 1200 UTC. The AOD data were not always available as the data quality control (i.e., cloud screening). There was a higher increase in the concentration at noon (0600 UTC) (Figure 9b), corresponding to a few high AOD during mild dust episodes at that hour. The DA system had to raise the $PM_{10}$ to fit the observed high AOD values. Because the CE318 AOD was only available in the daytime, no DA analysis was performed at 1800 UTC. Also, due to the limited AOD data, assimilating AOD did not substantially increase the correlation of PMx. The analyzed $PM_{2.5}$ and $PM_{10}$ still had low correlations with the observed levels ($R$=0.31~0.35).

**3.4 Assimilating Aerosol Scattering Coefficient**
Assimilating the aerosol scattering coefficient (DA_Esca) yielded overall analyses similar to the phenomenon in DA_AOD. The analyzed 635 nm scattering coefficient (192.1 $Mm^{-1}$) was lower than the observation (231.5 $Mm^{-1}$), with a high correlation of 0.97. The low biases were smaller at short wavelengths (Figure 10b). The wavelength-dependent biases indicated that the current DA system cannot eliminate the bias at each wavelength simultaneously. The analyzed monthly mean AOD was 0.53, better than the AOD of 0.38 when assimilating PMx. However, the surface particle concentrations were overestimated (i.e., positive biases by 14%

for $PM_{2.5}$ and 37% for $PM_{10}$), with a substantial increase in the coarse particle of OIN.
Overestimations appeared during a few mild dust episodes (Figure 7d). It indicated that WRF-
Chem underestimated the dust scattering efficiency, in accordance with the low bias in the
ratio of the scattering coefficient to $PM_{10}$ (0.52 $Mm^{-1}$ $\mu g^{-1}$ $m^3$; Table 4). Thus, the DA
system overfitted the PMx concentration to approach the observed scattering coefficient. The
diurnal $PM_{10}$ in the analysis was similar to the assimilation of PMx, showing a maximum
improvement and a robust nocturnal variation at 1800 UTC (Figure 9c). Assimilating the
scattering coefficient failed to improve the absorption coefficient. The monthly mean
absorption coefficient was 16.5 $Mm^{-1}$, 65% lower than the observed value. The AE responses
in DA_Esca followed those in DA_AOD. The AE values were overfitted (–0.15) for AOD,
slightly improved (0.19) for the scattering coefficient, and got a worse larger (1.95) for
absorption coefficient.
**3.5 Assimilating Aerosol Absorption Coefficient**
In contrast to the above results, assimilating the absorption coefficient (DA_Eabs)
deteriorated all the analyses other than the absorption coefficient. The analyses showed
substantial daily variations, and strong positive biases appeared in the dust episodes (Figure
7). The $PM_{2.5}$ was overestimated by a factor of three, and the $PM_{10}$ was overestimated by a
factor of four. The increases occurred each hour (Figure 9d). Because of the constant ratio
between mass and number concentration, the particle number concentration increased. As a
result, the aerosol scattering coefficient was overfitted to 612.2 $Mm^{-1}$, higher than the
observed levels by a factor of three. The monthly mean AOD improbably rose to 1.73.
Nevertheless, the absorption coefficient (40 $Mm^{-1}$) was improved to the observed level (47.4
$Mm^{-1}$). The AE responses were similar to the results in DA_AOD, showing an overfitted (–
0.01) for AOD, a little better value for the scattering (0.48), and a worse larger for the
absorption (2.01).
Improving the absorption coefficient at the cost of $PM_{10}$ overestimation indicates the model
biases in the representation of the particle mixture and the other absorbing particles (e.g.,
black carbon, brown carbon, and aged dust). The leading absorption aerosol in WRF-Chem is
BC, which had the maximum absorption and hence the maximum DA modification in the
second size (0.156–0.625 $\mu m$; Figure 8e). Because the BC had a small background
concentration, the BC showed a small DA improvement (<1.5 $\mu g$ $m^{-3}$) and did not largely
enhance the particle absorption. Meanwhile, the coarse dust particle concentration was
primarily increased but did not have a strong absorption as BC. As a result, the model lowered
the absorption coefficient's ratio to $PM_{10}$ by order of magnitude (0.05; Table 4). Because of
the observed absorption coefficient constraint, the DA system dramatically overestimated the
particle concentrations and induced too much higher aerosol scattering coefficient and AOD.
The overestimated $PM_{10}$ lowered the mass scattering and absorption efficiencies. The mass
absorption efficiency was much lower at a short wavelength (Figure 10c), opposing the lower
bias at a long-wavelength for the mass scattering efficiency (Figure 10b). The low biases were
dependent on the wavelength, indicating an elaborate tuning that simultaneously eliminates
the wavelength-dependent bias. It requires the DA system, for example, to add aerosol
number concentration as an additional control variable and specify complex refractive index
at each wavelength more precisely. The WRF-Chem aerosol simulation uses a high number of
size bin representation is also helpful.
As the strong positive biases in PMx were concerned, the scattering coefficient's
overestimation was higher than that of the absorption coefficient in DA_Eabs (Table 2). As a
result, DA_Eabs gave the highest SSArf (0.9; Table 3) in all DA experiments, opposite to our
expectation that the assimilation of absorption coefficient should decrease the positive bias in
SSA.
To understand the DA_Eabs's failure, we performed a few sensitivity experiments by
changing the imaginary part of the dust refractive index on 1200UTC on April 9. The dust's
imaginary part that we set in the experiments covers the retrieved value range of imaginary
index for typical desert dust as shown in Di Biagio et al. (2019). The results are presented in
the supplementary table S4a and S4b. The sensitivity experiments show that a high imaginary
part of the dust refractive index decreases the aerosol absorption coefficient (Table S4b). This
paradox is due to the BC's reduction. Specifically, a high imaginary part increases coarse
dust's absorption efficiency and decreases the coarse dust number concentration (num_a04;
Table S4a). This reduction led to less fine aerosol number concentrations (e.g., num_a02)
because of the inter-size bin correlation. BC is abundant in the second and third size bins, and
its imaginary part of the refractive index is two orders of magnitude higher than dust. Less BC
caused a weak absorption coefficient. On the contrary, the low dust imaginary part would not
largely increase dust numbers in the coarse size bin because the DA system also attempts to
increase BC in the fine particles to enhance the absorption coefficient. In an extreme case
with zero value of imaginary part of dust, the improvement of absorption coefficient
exclusively relies on BC; the num_a02 is increased by order of magnitude (Table S4a), and
660 nm Eabs rose to 92.5 $Mm^{-1}$ (Table S4b), much higher than the observed level.
At Kashi, BC has a low background concentration and low background error. The innovation
of BC was limited. Thus, tuning the imaginary part of dust's complex refractive index would
not significantly change the SSAsrf value (0.89 to 0.92). Excluding the contribution from OIN
in $PM_{10}$, the scattering coefficient was associated with sulfate. The sulfate's background error
was higher than the BC's by order of magnitude. The DA system prioritized sulfate
modification even when assimilating absorption coefficient, resulting in a smaller BC mass
fraction in $PM_{10}$ (Figure 12f) and a high SSAsrf of 0.90.
We did another set of sensitivity experiments by increasing the original BC's BEC per size
bin. As shown in the supplementary Table S5, increasing the BC's BECs would not much
deteriorate the absorption coefficient and significantly decrease the positive biases in PMx,
AOD, and scattering coefficient; the SSAsrf approached the observation. Increasing BC's
BECs by a factor of seven (DA_Eabs_BC*7) shows the best analyses. This experiment
suppressed the positive biases without decreasing the absorption coefficient's accuracy
(Figure 7), and the BC mass fraction increased (Figure 12g). The absorption AE decreased to
1.41 (Table 3). Although the decrease was small, this change was opposite to the increase in
the absorption AE in the other DA experiments. Nevertheless, the disadvantage of the
enlargement of BC'BEC is noticeable. The simultaneous assimilation of scattering and
absorption coefficient is not convergent as well as before. After four outer loops and each
with 50 inner iterations, the analyzed absorption coefficient in DA_Eabs_BC*7 was still
higher than the observed value by 47% (Figure S1j). These results indicate a low bias in BC's
background concentration that violates the prerequisite unbiased condition for the control
variable in Eq (1), and this background bias is too large to be consumed in BEC.

**3.6 Assimilating Multi-source Observations**
Assimilating an individual observation improves the corresponding model parameter (i.e.,
$PM_{2.5}$, $PM_{10}$, Esca, Eabs, and AOD) but may worsen other parameters. The reasons for the
inconsistent improvements are relevant to the aerosol model itself. These are: (1) the model
parameters have opposite signs in biases (e.g., one model parameter has a positive bias while
another has a negative bias); (2) the model biases have vast differences in magnitude (e.g., a
good fit of a parameter may lead to another's overfit) and the different biases in magnitude
cannot be reconciled because the forward operator is inaccurate to represent the linkage
between aerosol mass and aerosol optics (e.g., lower particle mass extinction efficiency).

In our case, simultaneous assimilation of the scattering and absorption coefficients
(DA_Esca_Eabs) resulted in the analyses when assimilating the scattering coefficient alone
(DA_Esca), and the inferior analysis in DA_Eabs vanished. This was because incorporating
the scattering coefficient constrained the aerosol number concentrations. DA_PMxAOD
substantially improved the AE for AOD, with an analyzed value of 0.17, consistent with the
observed value of 0.18 (Table 3). The scattering AE was somewhat improved (0.79), though it
was still far from the observed value of –0.43. The absorption AE (1.89) was worse than the
background (1.77), far deviating from the observed value of 1.65. Among the DA
experiments, simultaneous assimilation of PMx and AOD (DA_PMxAOD) gave the best DA
results, in which all the analyses except the absorption coefficient were not significantly
different in the month mean values from the observations. Simultaneous assimilation of all
observations (DA_PMx_Esca_Eabs_AOD) did not substantially improve the analyses
compared with DA_PMxAOD because the surface coefficients and AOD had overlapped
information of the light attenuation. A redundant information source did not introduce extra
constraints on the DA system.

| Table 3, 4; Figure 6, 7, 8, 9, 10 |
| --- |


**3.7 Vertical Profiles of Aerosol Concentrations**
Figure 11 shows the vertical concentration profiles of $PM_{2.5}$ and $PM_{10}$. The DA system
increased the aerosol concentrations up to a height of 4 km, consistent with previous studies
on the Taklamakan Desert. Meng et al. (2019) simulated a deep dust layer thickness in spring,
with a 3–5 km depth. Ge et al. (2014) analyzed the Cloud-Aerosol Lidar Orthogonal
Polarization data in 2006–2012 in the desert. They showed that dust could be lifted to 5 km
above the Tarim Basin and even higher along the northern slope of the Tibetan Plateau.
Among our DA experiments, the vertical $PM_{10}$ concentration increased quickly in the lowest
three model layers and maintained high values at heights of less than 3 km. This vertical
profile corresponded to the background vertical error profile, reflecting the deep dust
transporting layer. The $PM_{2.5}$ vertical profiles showed a rapid reduction with an increase in
altitude. The figure clearly shows that DA_PMx improved the $PM_{2.5}$ and $PM_{10}$ better, whereas
DA_AOD preferentially adjusted the coarse particles and overestimated the $PM_{10}$. Also
shown in the figure are the vertical profiles normalized to their own respective surface
particulate concentrations. The assimilations added a larger fraction of the mass in these
layers and adjusted the shapes of the $PM_{10}$ profiles within 3 km above the ground (Figure
11d).

Figure 11


**4. Discussions**
**4.1 DA Impact on Aerosol Chemical Composition**
Due to the control variable design, our DA system modifies each aerosol's chemical
composition according to the BEC values. The $PM_{10}$ chemical fractions remain close to their
background values (Figure 12). As discussed in section 3.5, the assimilation of the aerosol
absorption coefficient alone (DA_Eabs) increased the sulfate fraction. Sulfate was the
predominant anthropogenic aerosol at Kashi and had a high background error value. The DA
system prioritized sulfate modification and prevented a rise in the BC fraction in DA_Eabs.
For the enlarged BC's BEC in DA_Eabs_BC*7, the BC mass fraction showed the largest
increase. The magnitude of the background error determines the analyzed aerosol chemical
fraction. The total aerosol quantities' assimilation cannot eliminate the intrinsic bias in aerosol
composition. Accurate aerosol chemistry and optical modules are crucial to attaining better
background aerosol chemical data for DA analysis (Saide et al., 2020).

Figure 12


**4.2 DA Impact on Aerosol Direct Radiative Forcing**
Table 5 shows the instantaneous clear-sky ADRF in the background data and the analyses of
DA_PMx and DA_PMxAOD. The DA effect gradually faded away after restarting the model
run. Because AOD and the surface particle concentrations had different DA frequencies, we
focused on the instantaneous radiative forcing values one hour after assimilating AOD data in
the two DA experiments to ensure that the comparison was based on similar analysis times.
The immediate data after DA also show the effective DA effects.
Aerosol redistributes the energy between the land and the atmosphere. The atmosphere gains
more shortwave energy as the dust and black carbon particle absorption; the warming
atmosphere emits more longwave energy as it absorbs shortwave energy. The change in
energy budget at the surface is correspondingly the opposite of that in the atmosphere. As
shown in Table 5, the enhancements in surface cooling forcings were slightly stronger than
those of atmospheric warming. The difference between the surface forcing and atmospheric
forcing is the ADRF at the top of the atmosphere (TOA). When assimilating the surface
particle concentrations, the TOA ADRF enhanced by 21% in the shortwave, 100% in the
longwave, and 18% in the net forcing values, and when assimilating the AOD, enhanced by

34%, 67%, and 32%, respectively. At Kashi, the total net (shortwave plus longwave) clear-sky ADRF with assimilating surface particles and AOD were $-10.4$ $Wm^{-2}$ at the TOA, $+20.8$ $Wm^{-2}$ within the atmosphere, and $-31.2$ $Wm^{-2}$ at the surface, and enhanced by 55%, 48%, and 50% respectively, compared to the background ADRF values.

It is noteworthy that the ADRF estimation remains uncertain even after DA. The AOD observation is only sporadically available because of cloud screening in retrieval data. The DA experiments cannot eliminate the low bias in AOD in WRF-Chem. The ADRF values in the DA experiments are likely to be weaker than the plausible aerosol radiative forcing at Kashi. Neither DA experiment lowers SSAsrf to approach the observation. Penner et al. (2001) claimed that under average conditions, an SSA less than ~0.85 tends to lead to net warming. The observed SSAsrf (0.78) indicates likely aerosol warming forcing at Kashi, while WRF-Chem and the DA analyses tend to impose aerosol cooling forcing. The ADRF uncertainty is associated with the background aerosols. WRF-Chem simulates aerosol size up to 10 μm, whereas larger particles (>10 μm) exhibit substantial absorption relative to scattering in the visible wavelength (Kok et al., 2017). Anthropogenic emission inventories need an update for the year 2019, reducing the potential low bias in BC concentration. Additionally, the revised GSI does not consider the change in particle effective radius per size bin when calculating the aerosol number concentration in each outer loop. Low absorption cross section raises aerosol number concentration as compensation, increasing aerosol scattering coefficient too much. If our tangent operator considered the change in particle effective radius per size bin, we could use aerosol mass and number concentration as control variables simultaneously. The DA system would have higher flexibility to balance the particle radius and number concentration and improve the absorption coefficient. All these need further research in the future.

**5. Conclusions**

This study described our revised GSI DA system for assimilating observed aerosol data for the four-size bin sectional MOSAIC aerosol mechanism in WRF-Chem. The DA system has new design tangent linear operators for the multi-wavelength AOD, aerosol scattering, and absorption coefficients measured by the sun-sky radiometer, nephelometer, and aethalometer, respectively. We examined the DA system for Kashi city in northwestern China by assimilating the multi-wavelength aerosol optical measurements gathered by the Dust Aerosol Observation–Kashi field campaign of April 2019 and the concurrent hourly measurements of surface $PM_{2.5}$ and $PM_{10}$ concentrations.

Our DA system includes two main aspects. Firstly, the control variable is the aerosol chemical composition per size bin corresponding to the WRF-Chem output data. This design allows modifying the composition of each aerosol based on their background error covariances. The number of control variables could be reduced by intentionally excluding a few aerosol compositions in a specific case if these compositions had low concentrations (e.g., chlorine and sodium in this study). Second, the DA system incorporates the observed AOD by assimilating the column mean aerosol extinction coefficient. This transfer avoids handling sensitivity from light attenuation length to the aerosol mass concentration in the tangent linear

operator, which is difficult to be accurately estimated and introduces significant errors in the operator. The tangent linear operator for AOD has two variants that can incorporate nephelometer and aethalometer measurements at the surface.

The most abundant aerosol at Kashi in April 2019 was dust. The WRF-Chem model captured the main dust episodes but underestimated the monthly mean concentrations of $PM_{2.5}$ and $PM_{10}$ by 17% and 41%, respectively. The model failed to capture the peak concentrations from a dust storm on 24 April. The aerosol scattering/absorption coefficients and AOD in the background data showed strong low biases and weak correlations with the observed levels. The DA systems effectively assimilate the surface particle concentrations, aerosol scattering coefficients, and AOD. Some deficiencies in the DA analysis were related to the forward model bias in transferring the aerosol mass concentrations to the aerosol optical parameter. Simultaneous assimilation of the $PM_{2.5}$ and $PM_{10}$ concentrations improved the model aerosol concentrations, with significant increases in the coarse particles; meanwhile, the analyzed AOD was 42% lower than observed levels. The assimilation of AOD significantly improved the AOD but overestimated the surface $PM_{10}$ concentration by 68%. Assimilating the aerosol scattering coefficient improved the scattering coefficient in the analysis but overestimated the surface $PM_{10}$ concentration by 37%. Therefore, it seems that WRF-Chem underestimated the aerosol extinction efficiency. As a compensation, the DA system overestimated the aerosol concentration to fit the observed optical values, yielding overly high particle concentrations.

A notable problem was the assimilation of the absorption coefficient, which greatly overestimated the monthly mean values by a factor of four in $PM_{10}$. The aerosol absorption coefficient was improved but was still 16% lower than observed values. The failure of DA analysis when assimilating the absorption coefficient is associated with many factors, including the biases of the model in aerosol particle mixture and aged dust, the uncertainties in the imaginary part of dust complex refractive index, the uncertain background error of BC, and the likely low bias in anthropogenic emissions. The most effective DA is the simultaneous assimilation of surface particle concentration and AOD, which provides the best overall DA analysis.

Our control variables' design allowed the DA system to adjust the aerosol chemical compositions individually. However, the analyzed anthropogenic aerosol chemical fractions were almost equivalent to the background chemical fractions. The reason is that the hydrophilic aerosols have equivalent or comparable refractive indices and hygroscopic parameters in the forward operator; they, therefore, have comparable tangent linear operator values when assimilating the aerosol optical data. It may be possible to separate the chemical compositions based on their background errors. The model anthropogenic aerosols were low at Kashi, probably due to the anthropogenic emissions' low biases. The low background concentrations led to low background errors and few increments for all chemical compositions. As a result, the chemical fractions of the anthropogenic aerosols remained close to their background values.

When assimilating surface particles and AOD, the instantaneous clear-sky ADRF (shortwave plus longwave) at Kashi were –10.4 $Wm^{-2}$ at the TOA, +20.8 $Wm^{-2}$ within the atmosphere, and –31.2 $Wm^{-2}$ at the surface, respectively. Since the DA analyses still underestimated the AOD value and overestimated SSA, the aerosol radiative forcing values assimilating the observations were underestimated in the atmosphere and the surface.

The limitations that necessitate further research include:

(1) The desired binning strategy should link the circulation flow and particle emission sources. A better hybrid DA coupled with the ensemble Kalman filter will be more effective for estimating the aerosol background error.

(2) The observational error could be elaborated further. The $PM_{10}$ included the anthropogenic coarse particles, which should be separated from the dust originating from the desert (Jin et al., 2019). We set the observation errors for PMx and AOD to the conventional values. The observational errors of the nephelometer and aethalometer were slightly arbitrary in this study, necessitating further consideration.

(3) The anthropogenic aerosols' background errors are needed to harmonize better to assimilate the aerosol absorption coefficient or absorption AOD.

(4) The DA system was based on four-size bin MOSAIC aerosols, but it can be extended to work with eight-size bin MOSAIC aerosols in WRF-Chem. When assimilating aerosol optical data, the DA quality is strongly dependent on the forward model. The responses of our DA analysis to the bias and uncertainty in the forward aerosol optical model in WRF-Chem need further investigation.

**Author contributions**

WC developed the DA system, performed the analyses, and wrote the paper. ZL led the field campaign and revised the paper. YZ and KL implemented the observations and the data quality control. YZ helped to design the new tangent linear operator. JC verified the DA system.

**Competing interests**

The authors declare that they have no conflict of interest.

**Code/Data availability**

The official GSI code is available at https://dtcenter.org/community-code/gridpoint-statistical-interpolation-gsi/download. The revised GSI code is available at https://github.com/wenyuan-chang/GSI_WRF-Chem_MOSAIC. The aerosol measurements at Kashi belong to the Sun-sky radiometer Observation NETwork (SONET), which is accessible at http://www.sonet.ac.cn/en/index.php.

**Acknowledgments**

This work is supported by the National Key Research and Development Program of China (Grant number 2016YFE0201400). We appreciate the two anonymous reviewers' constructive comments that improve the paper.

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

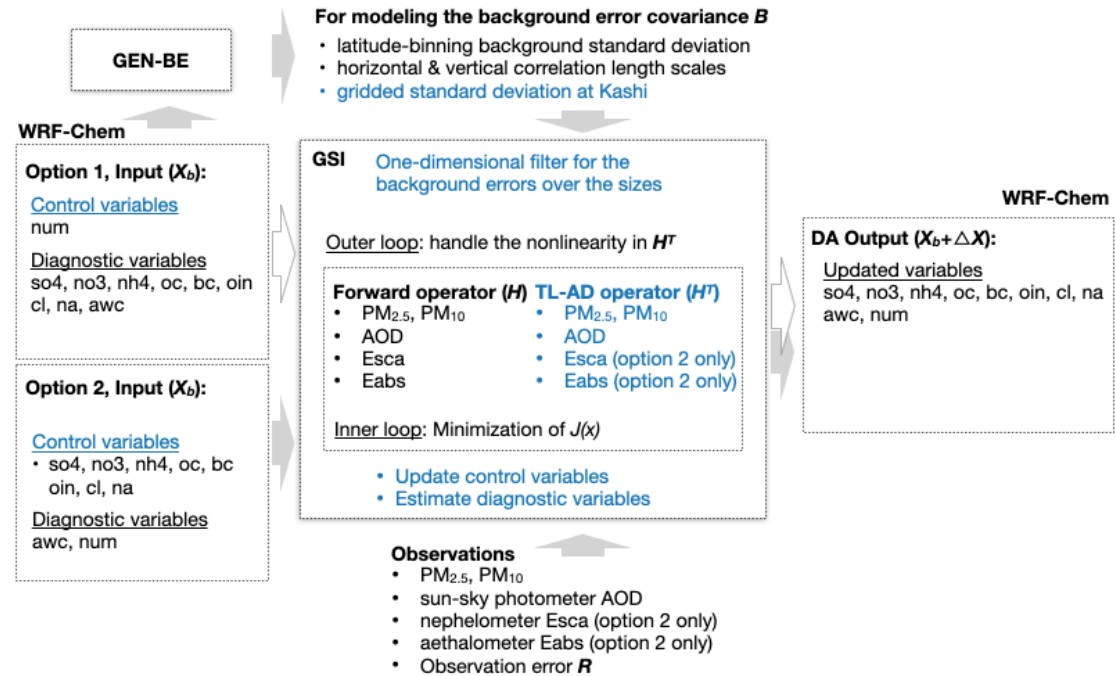



Figure 1. The workflow of aerosol DA in the revised GSI system for the sectional
MOSAIC aerosols in WRF-Chem. The contents in blue are the portions we
developed. The arrows in gray indicate the workflow of option 2 which was
performed in this study to assimilate the aerosol scattering/absorption coefficients.
Abbreviations: so4, sulfate; nh4, ammonium; oc, organic carbon; bc, black carbon;
oin, other inorganic matter; awc, aerosol water content; num, aerosol number
concentration; no3, nitrate; cl, chlorine; na, sodium; Esca, aerosol scattering
coefficient; Eabs, aerosol absorption coefficient.

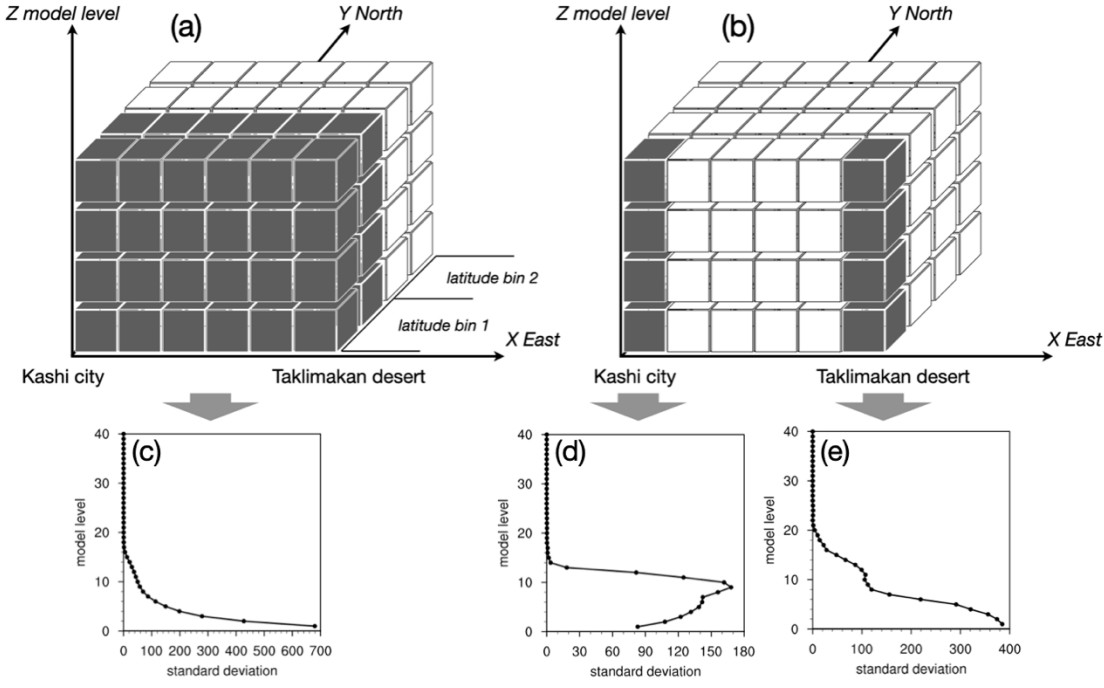



Figure 2. Schematic diagram of the binning strategy for modeling background error
covariance matrix on (a) the latitude binning data or (b) the gridded data; and the
vertical profiles of standard deviations ($\mu g\ kg^{-1}$) of the fourth size-bin OIN
component concentration at 0600UTC over a few mild dust episodes in April 2019 (c)
on average over the latitude bins, (d) at Kashi city grid and (e) at the Taklimakan
desert grid (i.e., 1.5 degrees east to the Kashi city).

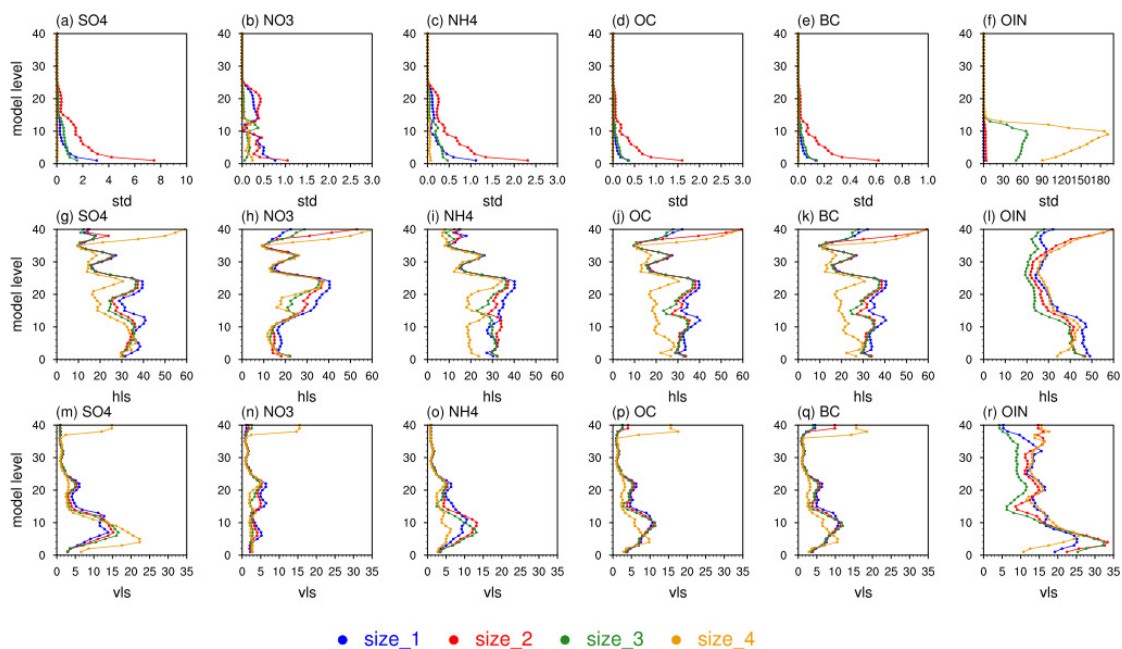

Figure 3. Background error standard deviations at Kashi grid (std, a-f, µg kg⁻¹), horizontal correlation length scales (hls, g-l, km), and vertical correlation length scales (vls, m-r, km) at 0000 UTC in April 2019 for the sectional sulfate (SO4), nitrate (NO3), ammonium (NH4), organic aerosol (OC), black carbon (BC), and other inorganic aerosols (OIN, including dust) in the model domain 2. The horizontal and vertical correlation length were computed based on the latitude bins with a half degree width.

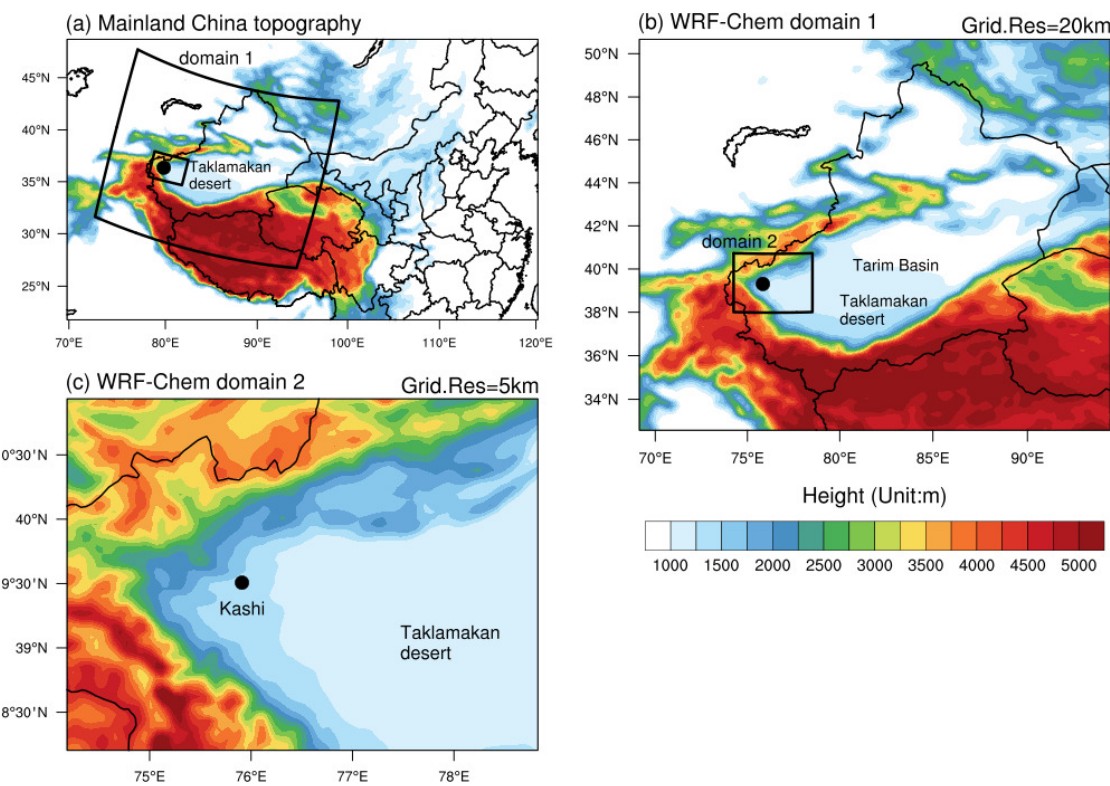



Figure 4. Topography in China (a) and the model domains with the grid resolution of
20 km (b) and 5 km (c) in WRF-Chem.

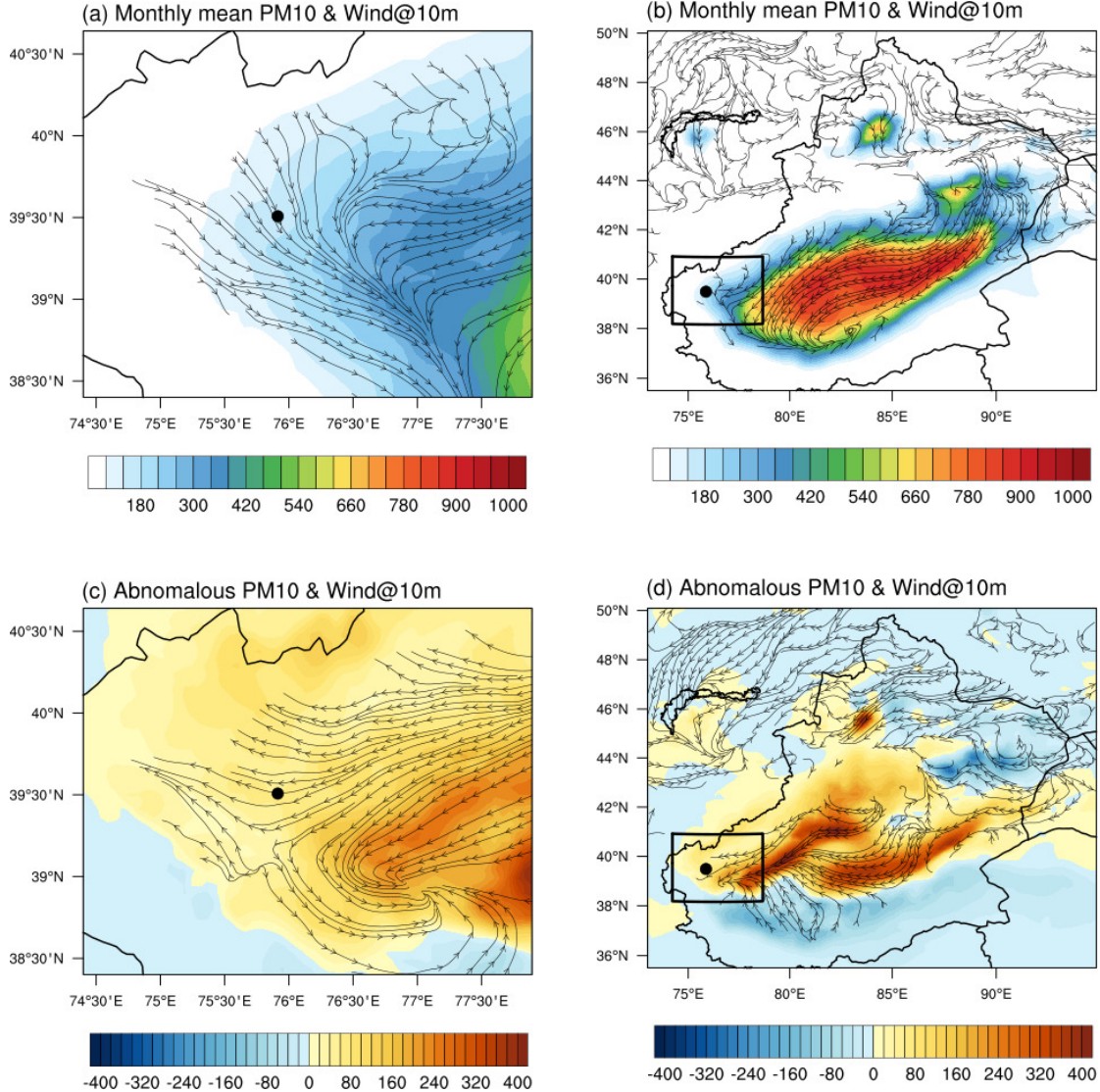



Figure 5. Monthly mean PM$_{10}$ concentration ($\mu$g m$^{-3}$) and the streamlines of the 10-m
wind (m s$^{-1}$) in April (a, b) and their daily mean anomalies (c, d) during a dust storm
on 24 April to the monthly mean values. Only the streamlines at the topographical
height lower than 2500 meters are shown for clarity. The rectangles in figures (b) and
(d) denote the fine model domain 2, which was the geographical range in the figures
(a) and (c). The black points indicate the Kashi city.



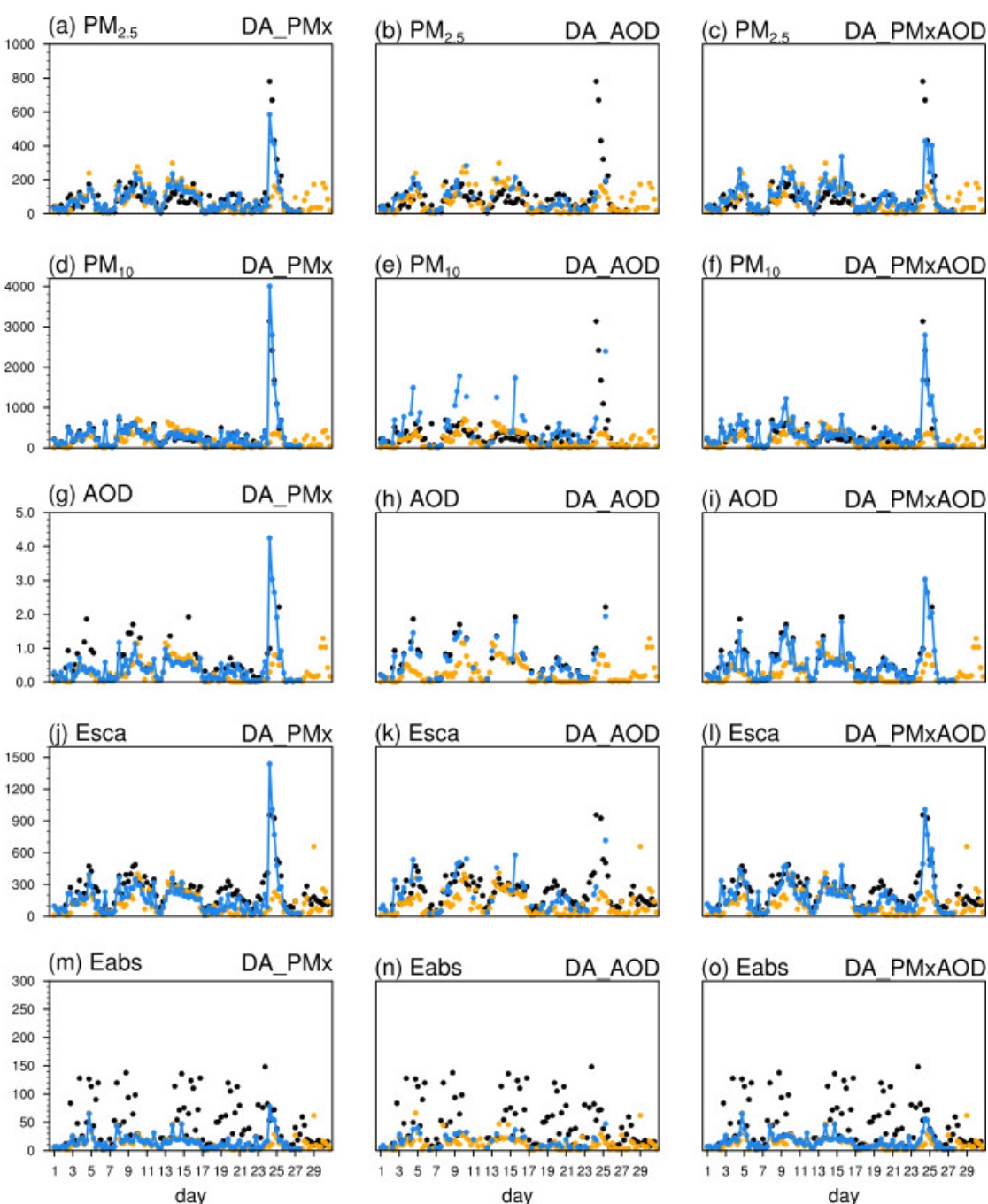



Figure 6. Comparison of PM$_{2.5}$ (µg m$^{-3}$; a-c), PM$_{10}$ (µg m$^{-3}$; d-f), 870 nm AOD (g-i),
635 nm aerosol scattering coefficient (Esca, Mm$^{-1}$; j-l), and 660 nm aerosol
absorption coefficient (Eabs, Mm$^{-1}$; m-o) in the observation (black solid point), the
background simulation (orange solid point), and the DA analyses (blue line) when
assimilating the observed PM$_{2.5}$ and PM$_{10}$ (DA_PMx), AOD (DA_AOD), and
simultaneously assimilating PMx and AOD (DA_PMxAOD) at Kashi in April 2019.

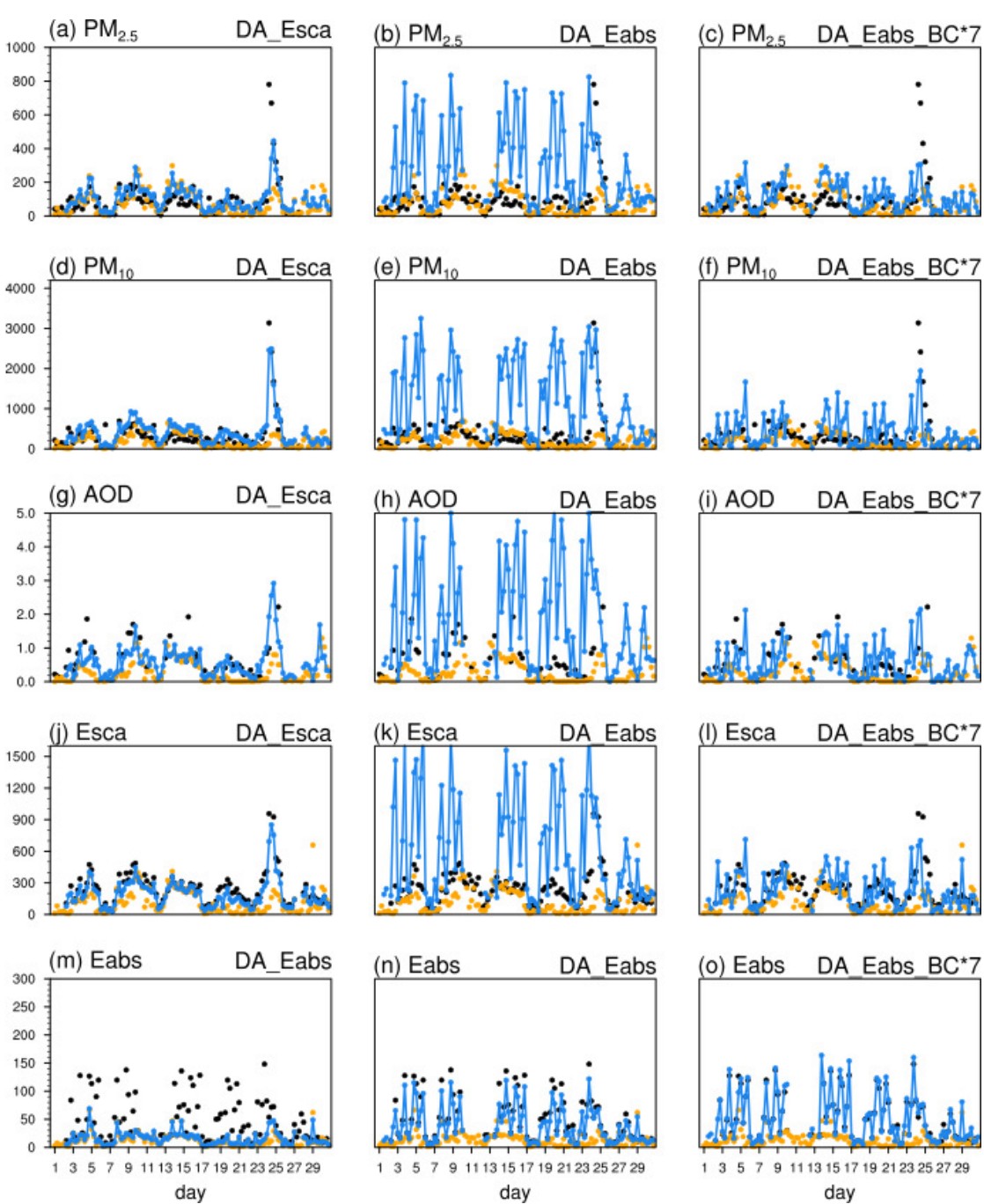



Figure 7. Comparison of PM$_{2.5}$ (µg m$^{-3}$; a-c), PM$_{10}$ (µg m$^{-3}$; d-f), 870 nm AOD (g-i),
635 nm aerosol scattering coefficient (Esca, Mm$^{-1}$; j-l), and 660 nm aerosol
absorption coefficient (Eabs, Mm$^{-1}$; m-o) in the observation (black solid point), the
background simulation (orange solid point), and the DA analyses (blue line) when
assimilating the aerosol scattering coefficient (DA_Esca), aerosol absorption
coefficient (DA_Eabs), and absorption coefficient with the background error of BC
enlarged by a factor of 7 (DA_Eabs_BC*7) at Kashi in April 2019.

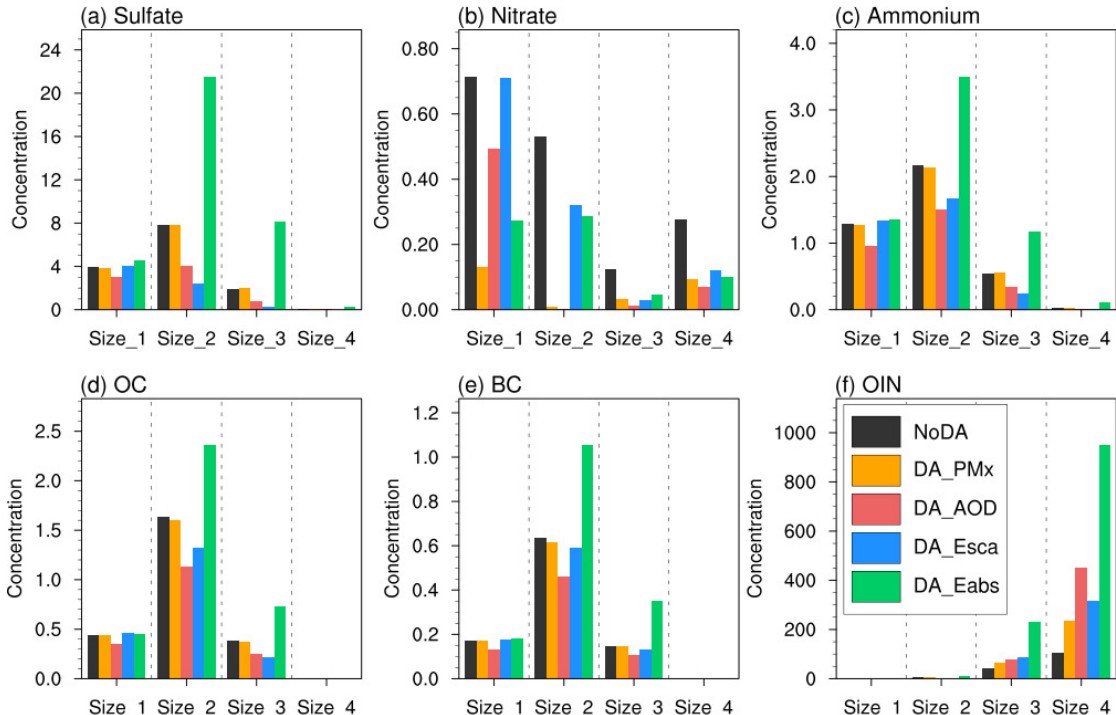

Figure 8. Mean aerosol concentrations ($\mu g\ m^{-3}$) per size bin in the background (NoDA) and the DA analyses when assimilating each individual observation at Kashi in April 2019.

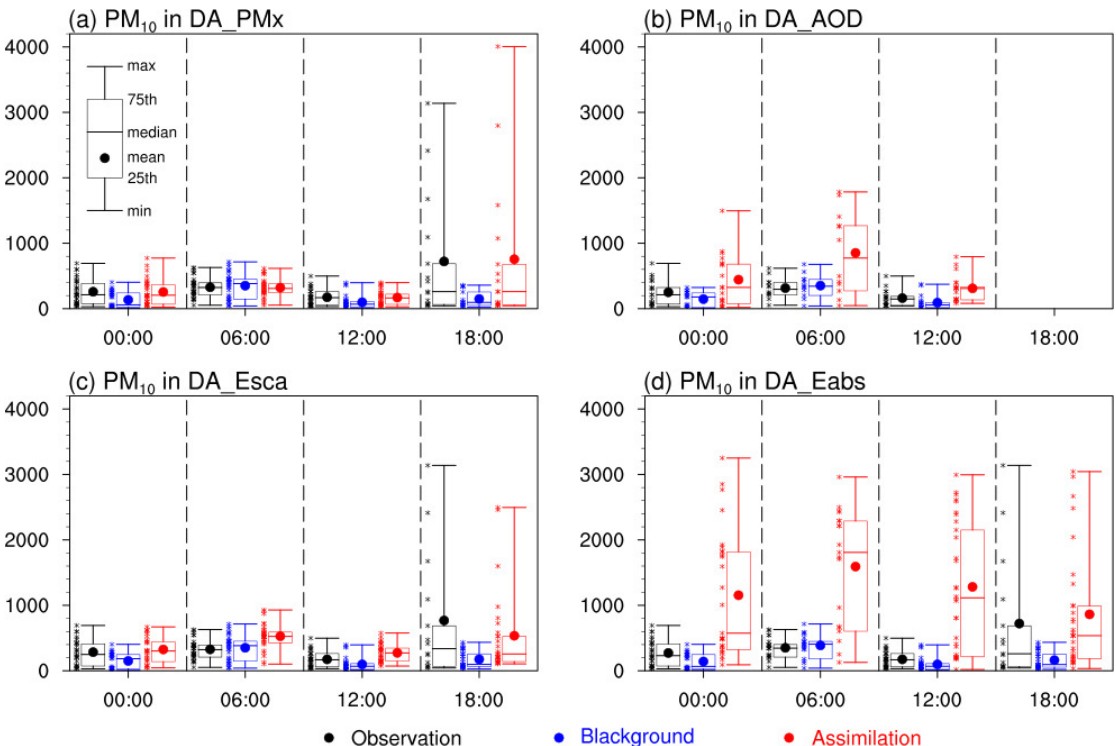

Figure 9. Surface PM$_{10}$ concentrations (µg m$^{-3}$) in the observation (black),
background simulation (blue) and the DA analyses (red) at 0000, 0600, 1200, 1800
UTC in April 2019 when assimilating the observations of (a) PMx, (b) AOD, (c)
aerosol scattering coefficients (Esca), and (d) aerosol absorption coefficient (Eabs),
respectively. The DA_AOD had no analysis at 18:00 UTC that was local midnight.
Kashi is 6 hours ahead of UTC (UTC+6).

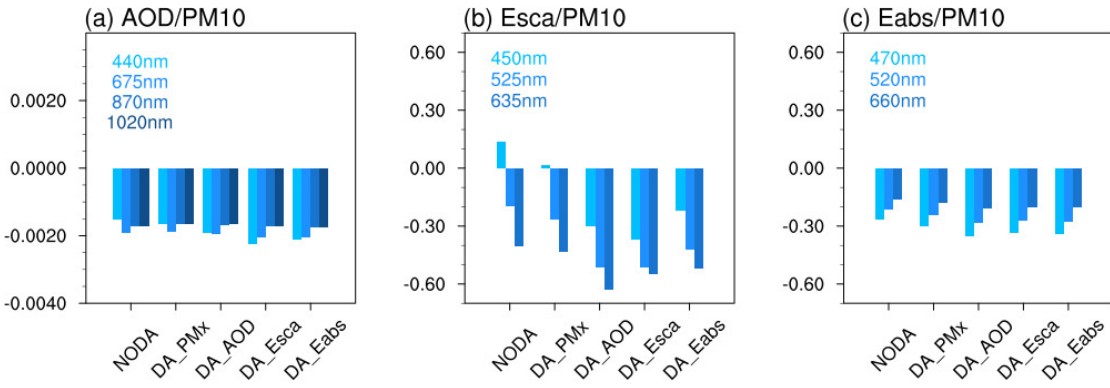



Figure 10. Mean biases in the ratio of AOD to $PM_{10}$, the mass scattering efficiency
(Esca/$PM_{10}$, $Mm^{-1}$ $\mu g^{-1}$ $m^3$), and the mass absorbing efficiency (Eabs/$PM_{10}$, $Mm^{-1}$
$\mu g^{-1}$ $m^3$) at Kashi in April 2019.


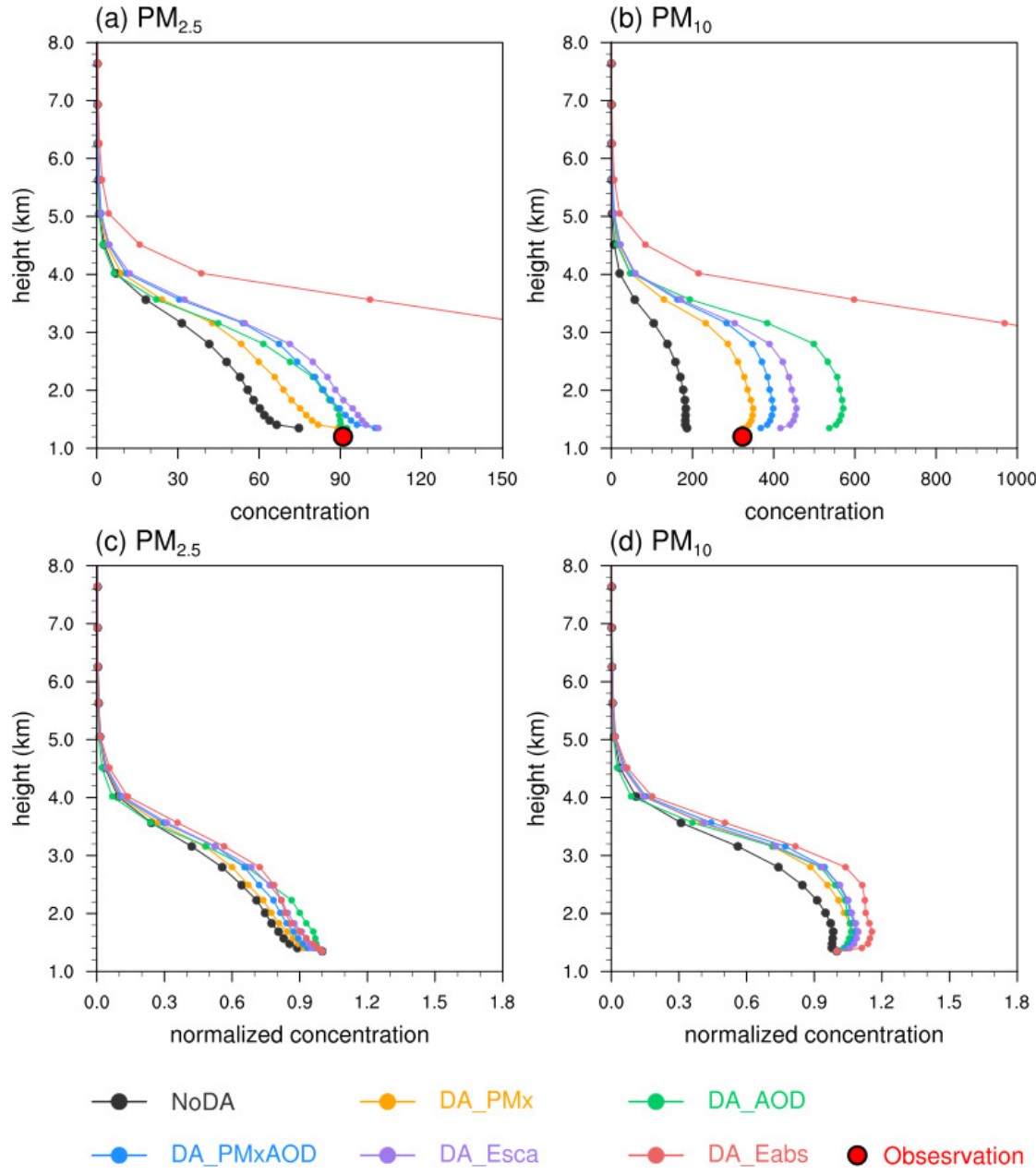



Figure 11. Mean vertical profiles of (a) $PM_{2.5}$ (µg m$^{-3}$), (b) $PM_{10}$ (µg m$^{-3}$) and their
normalized concentration respect to their own surface concentrations (c, d) at Kashi in
April 2019.

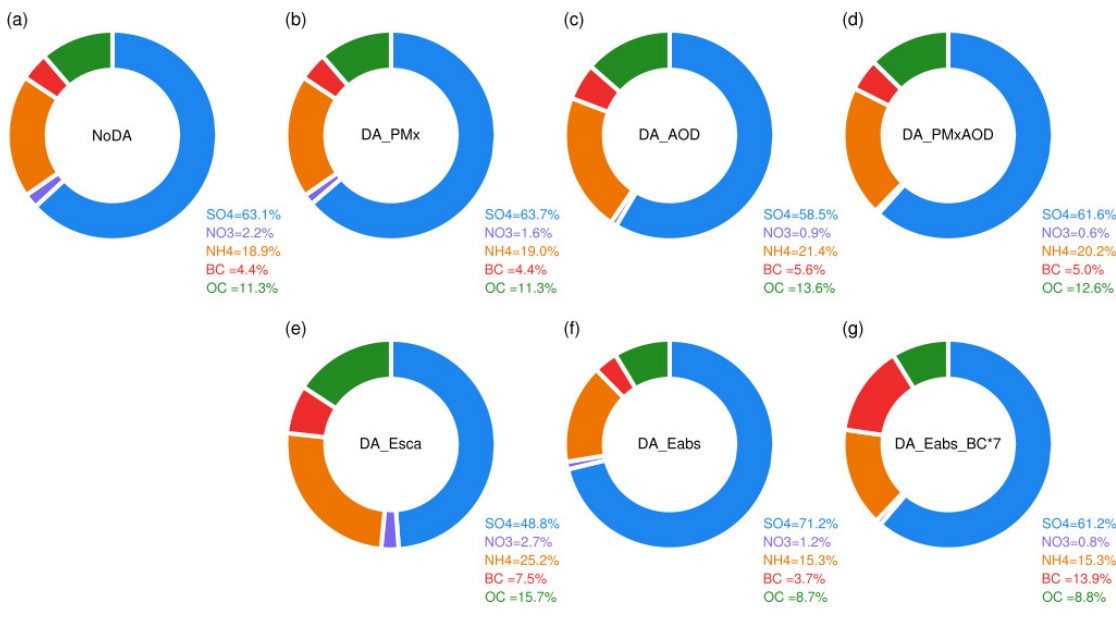




Figure 12. Mean mass percentage (%) of chemical composition in $PM_{10}$ excluding the OIN component at Kashi in April 2019.



Table 1. The observed surface particle concentration, aerosol scattering coefficient
(Esca), aerosol absorption coefficient (Eabs), and AOD used for the DA analysis and
their observational errors.

|  | Data time range | Wavelength (nm) | Observation error ($e$) |
|---|---|---|---|
| PM$_{2.5}$ & PM$_{10}$ (μg m$^{-3}$) | Apr 1 – Apr 30 |  | $e = \sqrt{e_1^2 + e_2^2}$ <br><br> $e_1 = 1.5 + 0.0075 \cdot PM_x$ <br><br> $e_2 = 0.5 \cdot e_1 \cdot \sqrt{\dfrac{d}{3000}}$ <br><br> $d$: grid spacing in meter |
| AOD | Mar 29 – Apr 25 | 440, 675, 870, 1020 | $e = 0.01/\text{height} \times 10^8$ |
| Esca (Mm$^{-1}$) | Apr 2 – Apr 30 | 450, 525, 635 | $e = 10$ |
| Eabs (Mm$^{-1}$) | Apr 2 – Apr 30 | 470, 520, 660 | $e = 10$ |


Table 2. The mean values of the PM$_{2.5}$ and PM$_{10}$ concentrations (µg m$^{-3}$), 635 nm
aerosol scattering coefficient (Esca, Mm$^{-1}$), 660 nm aerosol absorption coefficient
(Eabs, Mm$^{-1}$) and 870 nm AOD in the background and analysis data and their
correlation coefficients (in brackets) with the observations at 0000, 0600, 1200, 1800
UTC at Kashi in April 2019. The underlined number in bold denotes the mean value
that is not significantly different from the observation, and the dashed line denotes an
insignificant correlation. Both the statistical tests of the mean difference and
correlation are conducted at the significance level of 0.05.

| DA experiment | PM$_{2.5}$ (µg m$^{-3}$) | PM$_{10}$ (µg m$^{-3}$) | 870 nm AOD | 635nm Esca (Mm$^{-1}$) | 660nm Eabs (Mm$^{-1}$) |
|---|---|---|---|---|---|
| Observation | 91.0 | 323.2 | 0.66 | 231.5 | 47.4 |
| Background | **75.3** (0.28) | 190.7 (0.24) | 0.24 (0.60) | 123.3 (0.36) | 12.9 (0.34) |
| | | | | | |
| DA_PM$_x$ | **89.3** (0.89) | **329.3** (0.99) | 0.38 (0.35) | 170.4 (0.89) | 15.8 (0.42) |
| DA_AOD | **92.6** (0.35) | 541.7 (0.31) | **0.59** (0.98) | **222.6** (0.61) | 17.0 (0.26) |
| DA_PMxAOD | **103.6** (0.61) | **372.7** (0.86) | **0.59** (0.98) | **192.2** (0.86) | 16.7 (0.45) |
| | | | | | |
| DA_Esca | **103.6** (0.67) | 442.1 (0.93) | **0.53** (0.62) | **192.1** (0.97) | 16.5 (0.47) |
| DA_Eabs | 298.8 (0.36) | 1281.2 (0.34) | 1.73 (----) | 612.2 (0.54) | **40.0** (0.98) |
| DA_Eabs_BC*7 | 106.7 (0.48) | 463.7 (0.45) | **0.75** (0.50) | **226.2** (0.52) | **51.9** (0.90) |


Table 3. The Ångström exponent values based on the AOD (440 nm and 1020 nm;
AEaod), aerosol scattering coefficients (450 nm and 635 nm; AEsca), and aerosol
absorption coefficients (470 nm and 660 nm; AEabs), and the surface single scattering
albedo (SSAsrf=Esca525/(Esca525+Eabs520)) at Kashi in April 2019

|  | 440-1020 nm AEaod | 450-635 nm AEsca | 470-660 nm AEabs | SSAsrf |
|---|---|---|---|---|
| Observation | 0.18 | −0.43 | 1.65 | 0.78 |
| Background | 0.54 | 1.32 | 1.77 | 0.86 |
|  |  |  |  |  |
| DA_PMx | 0.30 | 0.96 | 1.84 | 0.88 |
| DA_AOD | −0.01 | 0.44 | 1.97 | 0.88 |
| DA_PMx_AOD | 0.17 | 0.79 | 1.89 | 0.89 |
|  |  |  |  |  |
| DA_Esca | −0.15 | 0.19 | 1.95 | 0.88 |
| DA_Eabs | −0.01 | 0.48 | 2.01 | 0.90 |
| DA_Eabs_BC*7 | 0.33 | 0.89 | 1.41 | 0.82 |


Table 4. The ratios of AOD, aerosol scattering/absorption coefficient to $PM_{10}$
concentration (mean ± standard deviation) in the observations, the model background
data, and the DA analyses.

|  | Ratios of 870 nm AOD to $PM_{10}$ ($\mu g^{-1}$ $m^3$) | Ratios of 635 nm aerosol scattering coefficient (Esca) to $PM_{10}$ ($Mm^{-1}$ $\mu g^{-1}$ $m^3$) | Ratios of 660 nm aerosol absorption coefficient (Eabs) to $PM_{10}$ ($Mm^{-1}$ $\mu g^{-1}$ $m^3$) |
|---|---|---|---|
| Observation | 0.0030±0.0020 | 1.05±0.57 | 0.25±0.22 |
| Background | 0.0013±0.0009 | 0.65±0.18 | 0.09±0.05 |
|  |  |  |  |
| DA_PMx | 0.0013±0.0008 | 0.61±0.22 | 0.07±0.05 |
| DA_AOD | 0.0013±0.0011 | 0.51±0.24 | 0.05±0.04 |
| DA_PMxAOD | 0.0015±0.0010 | 0.61±0.24 | 0.06±0.05 |
|  |  |  |  |
| DA_Esca | 0.0015±0.0010 | 0.52±0.21 | 0.05±0.05 |
| DA_Eabs | 0.0015±0.0010 | 0.58±0.37 | 0.05±0.06 |
| DA_Eabs_BC*7 | 0.0023±0.0085 | 0.74±0.51 | 0.30±0.48 |


Table 5. The mean instantaneous clear-sky shortwave (SW), longwave (LW) and the
net (SW+LW) direct radiative forcing (Wm$^{-2}$) at the top of atmosphere (TOA), in the
atmosphere (ATM) and at the surface (SRF) in the background and the simulations
restarted from the analyses of DA_PMx and DA_PMx_AOD at one hour after the
analysis times of AOD at Kashi in April 2019.

|  | SW (Wm$^{-2}$) | | | LW (Wm$^{-2}$) | | | SW+LW (Wm$^{-2}$) | | |
|---|---|---|---|---|---|---|---|---|---|
|  | TOA | ATM | SRF | TOA | ATM | SRF | TOA | ATM | SRF |
| Background | -7.0 | +17.0 | -24.0 | +0.3 | -2.9 | +3.2 | -6.7 | +14.1 | -20.8 |
| DA_PMx | -8.5 | +22.7 | -31.2 | +0.6 | -6.3 | +6.9 | -7.9 | +16.4 | -24.3 |
| DA_PMxAOD | -11.4 | +28.6 | -40.0 | +1.0 | -7.8 | +8.8 | -10.4 | +20.8 | -31.2 |

