# Peer review of "Improving the Sectional MOSAIC Aerosols of WRF-Chem with the Revised"

_Atmospheric Chemistry and Physics, 2020_

## Referee Comment (RC1) · Anonymous Referee #1 · 19 Oct 2020

The manuscript presents the development of assimilation of aerosol observations into WRF-Chem using the GSI system using approaches that are different to those used in previous studies. These developments are then tested for a case of assimilating ground-based observations of particle mass concentration, scattering and absorption coefficients, and AOD, performing sensitivity simulations on assimilating datasets in-

dependently and jointly. This is done for a single site located in Kashi, representative of dust conditions. This study it's within the scope of ACP and represents good contributions to the field as it develops a tool that could be used by the community and highlights shortcomings in the techniques a and how could they be improved. I think the paper needs a bit more work before it's ready for publication based on the comments below.

My main comments are the following.

- While the WRF-Chem optical properties module assumes Mie theory which is based on particles being spherical, the testing of the tool is focused on dust which are mostly non-spherical particles. This is briefly mentioned in the article, but I would like to see more on the subject, including looking into literature that has explored this topic and discussion on what discrepancies obtained in this study could be explained by this issue. See more on by line comments

- I believe that what the authors defined as Adjoint operators are really the tangent linear models, i.e., the derivative of the observables with respect to the inputs (aerosol mass). The adjoint operates on perturbations on the observables and outputs the expected perturbations on inputs. Please verify with the literature and correct accordingly.

- Assess representation of some intensive properties such as size (e.g., angstrom exponent, ratio of pm2.5 to pm10), single-scattering albedo, and mass scattering efficiency to try to understand mismatches when doing assimilation. A little bit is done but it would be very helpful to expand this topic and use the nomenclature used in the literature. See more on by line comments

- Absorption seems completely biased even after assimilation, this points to issues probably related to underestimation of imaginary refractive index of dust. Look for literature on this depending on the deserts, I believe Chinese deserts tend to have darker (i.e., more absorbing) sands.

Comments by line:

103-108. For completion, consider citing and discussing the study by Kumar et al (2019) that also uses GSI with CMAQ but does not use the CRTM as Tang study. This study also provides an alternative way of computing the BEC matrix (other than the NMC method) which you discuss in section 2.3

545-552. Are all of these observations in the same location? If not how far apart are they? How many PM2.5/PM10 sites are used? Also, what's the inlet cutoff size used for the scattering and absorption measurements? This is important to related mass and optical properties properly

554-561. Could you add justification for the PM2.5/PM10 observation errors stated in Table 1? There is no explanation how the errors were picked. Also, why do you only use representative error for PM2.5/PM0 but not for the other observations?

569. Can you add a bit more info on the vertical resolution? For instance, thickness of the 1st level and number of levels within 1km.

594-596. Can you clarify if you did 2 simulations every cycle with and without aerosol interacting with radiation, or it was a single simulation with two calls to the radiation code within the same simulation?

632-638. You are also missing some processes of potential importance such as secondary organic aerosol formation and heterogeneous sulfate formation influencing low-dust days.

645-647. I think a better fit to PM2.5 could be achieved if you relaxed the interbin correlation. It looks like PM10 is fitting pretty well but it's going a bit over the observation, so this is restricting increases in PM2.5 due to the correlation. Since bin 4 is 2.5-10um, in theory, if no interbin correlation was present, PM10 and PM2.5 should be able to fit independently. For this study it would make sense to relax the interbin correlation due to the known issues in dust size distributions (see next comment)

[Figure]

647-652. Literature on dust modeling states that parameterizations tend to overpredict the fine dust and underpredict the coarse dust (see Kok et al., 2011, Adebiyi and Kok 2020). So the joint assimilation of PM2.5 and PM10 could be somewhat correcting for that, which is a additional possible explanation to the behavior explained in these sentences.

672-685. Another reason for the discrepancy is related to the size distribution. Are you assimilating multi-wavelength AOD here, right? If so, I would expect some modifications to the size distribution. It looks you are effectively modifying size distr. as the ratio of PM2.5 to PM10 ratio is reduced from 0.31 in the background to 0.11 in the DA_AOD simulation but it might be going a bit too far as the observed ratio is 0.28. You can also check angstrom exponent. You can also explore the point you make at the end related to the dust mass extinction efficiency, you have observations to compute this at the surface. Additionally, there is also potential for your vertical distribution to be off and be generating these issues. You can diagnose this by comparing the ratio of surface extinction vs AOD. It seems the model is overpredicting this ratio, which could mean too much aerosol close to the surface.

Related to this point. You are actually already computing mass scattering efficiency (2nd column in Table 3). The background already underpredicts it, and the assimilation makes it worse as you are increasing the coarser fraction. You could explore if there is an underprediction of the dust refractive index. You could look into values provided in the literature for the region studied and compare to what WRF-Chem uses.

678-683. I think there is no need for this very long description of the Ma paper as these results are not that relevant to the area study as RH is likely low in the desert and dust aerosols tend to be hydrophobic

690-692. AOD to PM10 ratios depends on many variables. Since you are blaming discrepancies to issues in mass scattering/absorption efficiency it makes more sense to do direct comparissons to this variable as you have in-situ measurements of scattering

and absorption

693-696. You can assess issues with size distribution by using the angstrom exponent.

Table 2 and 3. Is there any reason behind using the lower wavelength (440-450nm) for these comparisons? Since the focus of this work is on dust, it would be preferable to compute optical properties for longer wavelengths where coarse aerosols contribute more to the scattering

702-706. There is extensive literature on how optical properties of dust particles deviate from Mie theory (e.g., Dubovik et al, 2006, Nousiainen et al 2015). It would be good for the authors to reference this work and attempt to explain what could be the implications of using Mie theory, and if those can explain any of the discrepancies found when assimilating multiple datasets in this study

Figure 12. It would help to see an additional panel with these profiles being normalized, so we can more easily assess by how much the assimilation of the different datasets is changing the vertical distribution.

730 You know it overestimated PM10, not sure about aerosol number concentration (you would need a different observation for assessing that)

734. Use single-scattering albedo for this

738-746. This is a misconception, aerosol light extinction and AOD does not depend on sun light intensity (for instance, you can sample both at night time with different methods). What's going to change with sunlight are the radiative effects. There are likely other reasons to explain this diurnal behavior. Look into the diurnal evolution of your BEC, and also into diurnal evolution of dust reaching the city. Similar misconnection is mentioned in lines 770-771.

960-963. This is probably due to underprediction of dust imaginary refractive index

Minor Edits

Fig 5 caption. It reads like a) and b) represent PM10 and winds, respectively, but I think that's not the case. Please revise

623. Did you mean "underestimates" instead of "lowered"?

780. Do you mean "particles that absorb radiation" rather than "aborting particles"? Also, I would like black carbon in that list as well.

781-791. I believe primary dust in WRF-Chem is also considered to be a bit absorbing (has a imaginary refractive index above 0). As mentioned in a previous comment, this number might be too low for dust in this region.

802-804. I disagree with this statement. If the model has biases that the assimilation is not able to correct (for instance, inaccurate real and imaginary refractive indexes) then assimilating multiple observation could also create unrealistic modifications to the model.

806-828. I wouldn't put DA_Esca_Eabs as an improvement over DA_Esca, they show pretty much the same results. This means that the absorption observations are not really generating any differences in the results. Also, DA_PMx_AOD matches better the assimilated variables (which off course is expected) and the better agreement with scattering you happened to underpredict it with PM assimilation, and overpredict it with AOD assimilation, so assimilating both yields you something in between.

832-834. As mentioned earlier, it would be better to check this using normalized profiles. The background profiles already had aerosols up to 4km, so is likely that the assimilation is just scaling this profile upwards rather than adding a larger fraction of the mass in these layers

Section 4.1. I don't think this section is very relevant, the aerosols are so dominated by dust and your BEC is constructed in a way dust aerosols will be the ones largely modified. So just briefly mentioning that the composition of these other aerosols doesn't change would do.

Section 4.2. Might want to discuss in this section how the large underprediction of dust absorption would impact these results.

References

Adebiyi, A. A., and J. F. Kok (2020), Climate models miss most of the coarse dust in the atmosphere, Science Advances, 6, eaaz9507.

Dubovik, O. et al. Application of spheroid models to account for aerosol particle non-sphericity in remote sensing of desert dust. J. Geophys. Res. Atmos. 111, D11208 (2006).

Kok, J. F.: A scaling theory for the size distribution of emitted dust aerosols suggests climate models underestimate the size of the global dust cycle, P. Natl. Acad. Sci. USA, 108, 1016–1021, https://doi.org/10.1073/pnas.1014798108, 2011.

Kumar, R., Delle Monache, L., Bresch, J., Saide, P. E., Tang, Y., Liu, Z., da Silva, A. M., Alessandrini, S., Pfister, G., Edwards, D., Lee, P., and Djalalova, I.: Toward Improving Short-Term Predictions of Fine Particulate Matter Over the United States Via Assimilation of Satellite Aerosol Optical Depth Retrievals, Journal of Geophysical Research: Atmospheres, 124, 2753-2773, 2019.

Lennartson, E. M., Wang, J., Gu, J., Castro Garcia, L., Ge, C., Gao, M., Choi, M., Saide, P. E., Carmichael, G. R., Kim, J., and Janz, S. J.: Diurnal variation of aerosol optical depth and PM2.5 in South Korea: a synthesis from AERONET, satellite (GOCI), KORUS-AQ observation, and the WRF-Chem model, Atmos. Chem. Phys., 18, 15125-15144, https://doi.org/10.5194/acp-18-15125-2018, 2018.

Nousiainen, T. & Kandler, K. Light scattering by atmospheric mineral dust particles. Light Scatt. Rev. 9 3–52 (2015).

---

## Referee Comment (RC2) · Anonymous Referee #2 · 28 Oct 2020

The study of Chang et al. developed the GSI 3Dvar capability to assimilate AOD, scattering/absorbing coefficients for MOSAIC scheme. A few DA tests (both simultaneously and separately experiments) were conducted for northwestern China and compared with surface observations at Kashi. The authors should have spent great efforts on the system development and presented very comprehensive results.

Based on my current understanding, some more work need to be done to facilitate the readers to understand, including some essential considerations of the DA core details and the clarifications of the texts. In this way, the system would be better understand/promoted and readers would be more convinced.

My general comments are as below:

1. Actually GOCART is understood for the better performance of dust simulation and the relevant optical properties had been well verified; while the MOSAIC scheme is thought to be more suitable for anthropogenic emission related simulation, but the optical simulation is rather complex.

In this study, the system is developed for MOSAIC but the verification is conducted for a site in desert. This required intensive investigation of the DUST related properties representation in the MOSAIC scheme, for example,

(a) the refractive index of OIN since it is mostly treated as DUST (while there should be distinctive differences between the two);

(b) the species partitioning (NO3 is not changed in option 2 which might not be reasonable and lead to unbalanced chemistry partitioning ), (c) the size distribution (d) the number concentration, since the three factors determining the absorbing and scattering efficiency;

(e) aerosol water content which are not considered but actually may change the optical properties. With very limited observational data to verify the above-mentioned information, the results in this study is really hard to interpret.

2. Some descriptions about DA core and observational data should be provided. For example, it seemed not only AOD, but also wavelength depended absorbing and scattering efficient were all assimilated, the corresponding observational operators and the errors should be given in more detail.

Comments by lines:

1. Line 65 and other places. Adjoint operator, is it referred as TL-AD? Please clarify

2. Line 86 GIS ?

3. Line 100 Zang et al 2016, acutally a different DA system was used other than GSI in this study. Please check.

4. Line 181 regarding of the low anthropogenic and biogenic emissions in the desert, why not use GOCART instead?

5. Line 184-190. Actually the optical properties of NH4SO4, OC, dust, NaCl, H2O are treated as wave-length depended in the model, this information should be investigated and provided. As it seemed that multi-wavelength aerosol scattering and absorption coefficients are assimilated. The uncertainties of the assumption in the model and observational data should be provided.

6. Line 228. Why NO3 is not considered? In this case, it may lead to unbalanced chemistry partitioning.

7. Section 2.2.3 It seemed that scattering and absorbing coefficients are also observational assimilated. Please provide details.

8. Line 101: are the $M_{i,z,k}$ in the two terms the same, maybe possibly dry and wet mass concentration respectively? If not, please clarify.

9. Line 315: is rwet related with aerosol water content, considering the hygroscopicity? Any uncertainty by not considering aerosol water content. Please clarify.

10. Line 352, please clarify mizk as dry or wet mass?

11. Line 367. Any uncertainty by considering constant radius?

12. Line 703-706. Please dig more on this issue.

13. Line 765. Please investigate the uncertainties of the modeled and observed absorption coefficients.

14. Figure2. Why the domain averaged standard deviation (c) is significantly larger than that of column averages (d, e)?

15. Figure 3. Why background error standard deviation of the OIN is two magnitudes larger than the other species? Indicating dominating contribution of dust? In this case, is it meaningful to investigate other species changes?

16. Table 1. Please explain how the errors are determined?

---

## Author Comment (AC3) · 29 Dec 2020

We appreciate the reviewer's constructive comments. Please check our replies in the supplement.

Please also note the supplement to this comment:
https://acp.copernicus.org/preprints/acp-2020-825/acp-2020-825-AC3-supplement.pdf

---

## Author Response (AR1)

The authors appreciate the reviewer's constructive and friendly comments. We have substantially revised the manuscript. New data and figures are present in the main text. A new supplementary document is included in the revision. We reply to the reviewer's comments point by point.

**Anonymous Referee #1**

Received and published: 19 October 2020

The manuscript presents the development of assimilation of aerosol observations into WRF-Chem using the GSI system using approaches that are different to those used in previous studies. These developments are then tested for a case of assimilating ground-based observations of particle mass concentration, scattering and absorption coefficients, and AOD, performing sensitivity simulations on assimilating datasets independently and jointly. This is done for a single site located in Kashi, representative of dust conditions. This study it's within the scope of ACP and represents good contributions to the field as it develops a tool that could be used by the community and highlights shortcomings in the techniques a and how could they be improved. I think the paper needs a bit more work before it's ready for publication based on the comments below.

My main comments are the following.

- While the WRF-Chem optical properties module assumes Mie theory which is based on particles being spherical, the testing of the tool is focused on dust which are mostly non-spherical particles. This is briefly mentioned in the article, but I would like to see more on the subject, including looking into literature that has explored this topic and discussion on what discrepancies obtained in this study could be explained by this issue. See more on by line comments.

Response: Yes, the spherical dust particle in WRF-Chem introduces uncertainty. We reviewed a few literatures and added a paragraph to discuss the impact of non-spherical particles in section 3.3.

- I believe that what the authors defined as Adjoint operators are really the tangent linear models, i.e., the derivative of the observables with respect to the inputs (aerosol mass). The adjoint operates on perturbations on the observables and outputs the expected perturbations on inputs. Please verify with the literature and correct accordingly.

Response: The author appreciates the reviewer's kindly comment. We have changed the misstatement of "adjoint operator" to "tangent linear operator".

- Assess representation of some intensive properties such as size (e.g., angstrom exponent, ratio of pm2.5 to pm10), single-scattering albedo, and mass scattering efficiency to try to understand mismatches when doing assimilation. A little bit is done but it would be very helpful to expand this topic and use the nomenclature used in the literature. See more on by line comments.

Response: According to this comment, the revised manuscript shows additional assessments of angstrom exponent and SSA (Table 3). A new figure 10 shows the multi-wavelength mass scattering/absorption coefficient. Hope this additional content makes this study more convincible.

- Absorption seems completely biased even after assimilation, this points to issues probably related to underestimation of imaginary refractive index of dust. Look for literature on this depending on the deserts, I believe Chinese deserts tend to have darker (i.e., more absorbing) sands.

Response: We used the generic model value of dust refractive index in the first version manuscript. In the revision, we increased the imaginary part of the dust refractive index, which is higher than the imaginary part of the Taklimakan desert that has been retrieved by Di Biagio et al. (2019). We find that tuning dust DA is not helpful for removing the DA bias in absorption coefficient. We redid a lot of DA experiments and found a negligence of anthropogenic emission in the WRF-Chem simulation. This strong bias in absorption DA is relevant to the low concentration of black carbon (BC) and the low BC's background error. We have rewritten the relevant content in section 3.5.

Di Biagio, C., et al.: Complex refractive indices and single-scattering albedo of global dust aerosols in the shortwave spectrum and relationship to size and iron content, Atmos. Chem. Phys., doi:10.5194/acp-19-15503-2019, 2019.

**Comments by line:**

103-108. For completion, consider citing and discussing the study by Kumar et al (2019) that also uses GSI with CMAQ but does not use the CRTM as Tang study. This study also provides an alternative way of computing the BEC matrix (other than the NMC method) which you discuss in section 2.3 Response: We added a few words about Kumar et al. (2019) study in the revised introduction and section 2.3.

545-552. Are all of these observations in the same location? If not how far apart are they? How many PM2.5/PM10 sites are used? Also, what's the inlet cutoff size used for the scattering and absorption measurements? This is important to related mass and optical properties properly.

Response: All the observations (PM2.5, PM10, AOD, scattering/absorption coefficient) were carried out at a single site. There was no inlet cutoff for the scattering and absorption measurements. In the revised manuscript:

"The site was placed in the Kashi campus of the Aerospace Information Research Institute, Chinese Academy of Sciences (39.50°N, 75.93°E; Li et al., 2018), about 4 km in the northwest to the Kashi city. ... All the instruments were deployed at the roof of a three stories height building on the campus."

554-561. Could you add justification for the PM2.5/PM10 observation errors stated in Table 1? There is no explanation how the errors were picked. Also, why do you only use representative error for PM2.5/PM0 but not for the other observations? Response: In the revised section 2.4:

"The observation errors of PMx are handled in the conventional way (Schwartz et al., 2012; Chen et al., 2019), which contains the measurement error (e1) and the representative error (e2). The measurement error is the sum of a baseline error of 1.5  $\mu$ g m-3 and 0.75% of the observed PMx concentration. The representative error is the measurement error multiplied by the half-squared ratio of the grid spacing to

the scale distance. The scale distance denotes the site representation in GSI and has four default values of 2, 3, 4, and 10 km, corresponding to the urban, unknown, suburban, and rural sites. We used 3 km for the scale distance in this study. As we had a single site in Kashi, it is difficult to estimate the site representation error. Since the DA analysis was based on the child model domain with a horizontal resolution of 5 km, close to the site distance to the Kashi urban area, we assumed the aerosol optical measurement had good representativeness of the model grid covering the site. The observation error of CE318 AOD took the AERONET AOD uncertainty of 0.01 in cloud-free conditions (Holben et al., 1998)."

569. Can you add a bit more info on the vertical resolution? For instance, thickness of the 1st level and number of levels within 1km.

Response: In the revised section 2.5:

"Both domains had 41 vertical levels extending from the surface to 50 hPa. The lowest model layer at the site was approximately 25-meter height from the ground."

594-596. Can you clarify if you did 2 simulations every cycle with and without aerosol interacting with radiation, or it was a single simulation with two calls to the radiation code within the same simulation? Response: It was a single simulation with two calls to the radiation code. In the revised section 2.5:

"To study the impact of DA on aerosol direct radiative forcing (ADRF), we modified the WRF-Chem code to calculate the shortwave irradiance with and without aerosols at each model integration step. The modified WRF-Chem model restarted from each DA analysis and ran to the next analysis time. Each running performed the radiation transfer calculation twice, and each calculation saw the aerosols and clean air, respectively. The irradiance difference between the two pairing calls was aerosol radiative forcing."

**632-638. You are also missing some processes of potential importance such as secondary organic aerosol formation and heterogeneous sulfate formation influencing low-dust days.**

Response: Yes, our simulation did not have SOA, and the heterogenous sulfate formation in WRF-Chem may bias. Nevertheless, we accidently lowered the anthropogenic emissions in the original WRF-Chem simulation. Because of the ambition of haze abatement in China since 2013, the atherogenic emissions had dramatic reductions in 2013-2019. So far as we know, a timely update of emission inventories is not available, and we used the open MEIC emission inventories for the year 2010 when the anthropogenic emissions had peak values. A general way to handle this emission reduction is to scale the historical emissions, which were not appropriately handled in our first manuscript. The anthropogenic emissions that we set for Kashi in the 2019 simulation were too low. As lack of aerosol measurement at Kashi, the low bias was not identified at the first glance. In the revised manuscript, we just ignore the yearly emission differences. We redid all simulations with the MEIC emission inventories for 2010.

The revised model concentrations of PM2.5 and PM10 are almost equivalent to the old data (Figure 1) because dust is the dominant component at Kashi. Besides, the real part of the refractive index of sulfate, nitrate, ammonium, and dust are comparable in the model. Thus, the new results do not change

the conclusion. The new advantage is that the DA bias in absorption coefficient can be somewhat attributed to black carbon when the BC's background error was amplified. We rewrote the DA of the absorption coefficient in section 3.5.

Figure 1. Comparisons of PM2.5 (left) and PM10 (right) in the WRF-Chem simulations with high (y-axis) and low anthropogenic emissions (x-axis) at Kashi in April 2019

645-647. I think a better fit to PM2.5 could be achieved if you relaxed the interbin correlation. It looks like PM10 is fitting pretty well but it's going a bit over the observation, so this is restricting increases in PM2.5 due to the correlation. Since bin 4 is 2.5-10um, in theory, if no interbin correlation was present, PM10 and PM2.5 should be able to fit independently. For this study it would make sense to relax the interbin correlation due to the known issues in dust size distributions (see next comment) Response: Based on lots of experiments, we find that the analyses are not sensitive to the inter-size bin correlation length in this case, though the analyses changed a lot when we turned off the inter-size bin correlation. We find that the magnitude of large background error of coarse dust is more effective in affecting the analysis of PM2.5. Reducing the background error of the fourth size bin OIN (oin\_a04) will increase PM2.5 and decrease PM10. We added table S3 in the supplementary document, which shows the PMx response to the different magnitudes of oin\_a04's background error.

In the revised section 3.2:

"Applying the inter-size bin correlation length caused the interlinked analyses of  $PM_{2.5}$  and  $PM_{10}$ . In the desert area, the coarse and fine dust are readily affected by the magnitude of BEC of the fourth size-bin OIN (oin\_a04). We intentionally decreased the BEC of oin\_a04 by 10% each time to 30% of its original value. The magnitude of 30% of oin\_a04 was comparable to the magnitude of the third size-bin (oin\_a03) OIN's background error. As shown in Table S3, because the oin\_a04's BEC reduction relaxes the constraint on the coarse particle, the  $PM_{10}$  bias becomes more negative along with the decrease in on\_a04's BEC. Meanwhile, the  $PM_{2.5}$  bias becomes more positive. Correspondingly, the ratio of  $PM_{2.5}$  to

 $PM_{10}$  was exaggerated to 0.33 with 30% of oin\_a04's BEC, higher than the observed value of 0.28. Overall, the original BEC of oin\_a04 is a reasonable tradeoff in our DA experiments."

647-652. Literature on dust modeling states that parameterizations tend to overpredict the fine dust and underpredict the coarse dust (see Kok et al., 2011, Adebiyi and Kok 2020). So the joint assimilation of PM2.5 and PM10 could be somewhat correcting for that, which is a additional possible explanation to the behavior explained in these sentences.

Response: Thanks for the hint. We cite the two pieces of literature in the revised section 3.2:

"As a result, the ratio of  $PM_{2.5}$  to  $PM_{10}$  decreased from 0.39 in the background to 0.27 in  $DA\_PMx$ , approaching the observed ratio of 0.28. Such improvement was consistent with the correction required to the model desert dust in literature. Kok et al. (2011) found that regional and global circulation models underestimate the fraction of emitted coast dust (>~5 µm), overestimates the fraction of fine dust (<2µm diameter). Adebiyi and Kok (2020) claimed that too rapid deposition of coarse dust out of the atmosphere accounts for the missing coarse dust in models. Similarly, WRF-Chem assimilated too much smaller dust particles than the observed. According to Kashi's AOD between 440 nm and 1020 nm, the observed Ångström exponent (AE) was 0.18 in this case, but the background value was 0.54 (Table 3). DA\_PMx reduced the AE value to 0.30, a little improvement but not sufficient."

672-685. Another reason for the discrepancy is related to the size distribution. Are you assimilating multi-wavelength AOD here, right? If so, I would expect some modifications to the size distribution. It looks you are effectively modifying size distr. as the ratio of PM2.5 to PM10 ratio is reduced from 0.31 in the background to 0.11 in the DA\_AOD simulation but it might be going a bit too far as the observed ratio is 0.28. You can also check angstrom exponent. You can also explore the point you make at the end related to the dust mass extinction efficiency, you have observations to compute this at the surface. Additionally, there is also potential for your vertical distribution to be off and be generating these issues. You can diagnose this by comparing the ratio of surface extinction vs AOD. It seems the model is overpredicting this ratio, which could mean too much aerosol close to the surface.

Response: In the revised manuscript, we check the PM2.5/PM10 ratio, mass extinction efficiency, angstrom exponent (AE), and SSA. We do not check the ratio of surface extinction and AOD because it requires the interpolation of surface extinction and AOD to similar wavelength. The model has a large bias in AE, resulting in an unreliable interpolation. We rewrote a lot in sections 3.2 and 3.3. Please refer to the revised manuscript.

Related to this point. You are actually already computing mass scattering efficiency (2nd column in Table 3). The background already underpredicts it, and the assimilation makes it worse as you are increasing the coarser fraction. You could explore if there is an underprediction of the dust refractive index. You could look into values provided in the literature for the region studied and compare to what WRF-Chem uses.

Response: We add a supplementary document to give the complex refractive indexes for all aerosols in this study. A part of the table is shown below. We set the dust's refractive index referring to the generic

model values in literature. The imaginary part in our study is higher than the retrieved imaginary part for the Taklimakan desert dust by Di Biagio (2019).

Table S1. Multi-wavelength real and imaginary parts of refractive indexes of aerosol chemical compositions and water in this study (a part of the snapshot of table S1)

| (nm) | 440   | 450   | 470   | 520         | 525          | 550       | 635         | 660    | 675    | 870   | 1020  |
|------|-------|-------|-------|-------------|--------------|-----------|-------------|--------|--------|-------|-------|
|      |       |       |       |             |              |           |             |        |        |       |       |
|      |       |       | 0.    | IN, dust (C | Theng et al. | , 2006; Z | hao et al., | 2010)  |        |       |       |
| Real |       |       |       |             |              | 1.53      |             |        |        |       |       |
| Imag | 0.003 | 0.003 | 0.003 | 0.0025      | 0.0025       | 0.002     | 0.0015      | 0.0015 | 0.0015 | 0.001 | 0.001 |

678-683. I think there is no need for this very long description of the Ma paper as these results are not that relevant to the area study as RH is likely low in the desert and dust aerosols tend to be hydrophobic Response: We have removed the statements in the revision.

690-692. AOD to PM10 ratios depends on many variables. Since you are blaming discrepancies to issues in mass scattering/absorption efficiency it makes more sense to do direct comparissons to this variable as you have in-situ measurements of scattering and absorption

Response: In the revised table 4, we show the ratios of AOD, scattering/absorption coefficient to PM10 per DA experiment. In the revised section 3.3:

"Table 4 shows the ratios of the AOD and aerosol scattering/absorption coefficients to the surface  $PM_{10}$  concentrations. The ratio of AOD to  $PM_{10}$  in the background model result was one-third of the observed levels. The observed mass scattering coefficient (Esca/PM10) was 1.05 Mm-1 µg-1 m3, while the background value was only 0.65 Mm-1 µg-1 m3. DA\_AOD did not eliminate the low bias but enlarged the low bias to 0.51 Mm-1 µg-1 m3. The same thing occurred for Eabs/PM10, which was 0.09 in the background and 0.05 in DA\_AOD, much lower than the observed value of 0.25. Figure 10 shows these mean ratios at the other wavelengths. The low bias in AOD/PM10 was comparable at each wavelength. ..."

693-696. You can assess issues with size distribution by using the angstrom exponent. Table 2 and 3. Is there any reason behind using the lower wavelength (440-450nm) for these comparisons? Since the focus of this work is on dust, it would be preferable to compute optical properties for longer wavelengths where coarse aerosols contribute more to the scattering Response: A new table 3 shows the angstrom exponent and SSA. The revised discussion in the main text is based on 870 nm AOD, 635 nm scattering coefficient, and 660 nm absorption coefficient.

702-706. There is extensive literature on how optical properties of dust particles deviate from Mie theory (e.g., Dubovik et al, 2006, Nousiainen et al 2015). It would be good for the authors to reference this work and attempt to explain what could be the implications of using Mie theory, and if those can explain any of the discrepancies found when assimilating multiple datasets in this study.

**Response: In the revised section 3.3:**

"The irregular morphology had a significant influence on the dust simulation. Okada et al. (2001) found that the aspect ratio (the ratio of the longest dimension to its orthogonal width) of the mineral dust particles (0.1-6  $\mu$ m) in China arid regions exhibited a median of 1.4. Dubovik et al. (2006) suggested the aspect ratio of ~1.5 and higher in desert dust plumes. Kok et al. (2017) found that the dust' sphericity assumption underestimated dust extinction efficiency by ~20–60% for the dust particle larger than 1 $\mu$ m. Tian et al. (2020) found that using a dust ellipsoid model could increase the concentration of coarse dust particle (5-10  $\mu$ m) by ~5% in eastern china and ~10% in the Taklimakan area because of the decrease in gravitational settling, comparing with the simulations with dust sphericity model. Nevertheless, the aspect ratio of the spheroid dust is uncertain. Even after applying the spheroidal approximation, Soorbas et al. (2015) found that the model underestimated 550 nm aerosol scattering and backscattering values by 49% and 11%, respectively, because of the uncertainties in particle axial ratio, complex refractive index, and the particle size distribution. To date, the assumption of spherical particles has been widespread in models (including WRF-Chem) for computational efficiency. Impact of dust morphology to DA deserves a further investigation."

Figure 12. It would help to see an additional panel with these profiles being normalized, so we can more easily assess by how much the assimilation of the different datasets is changing the vertical distribution. Response: The revised figure 12 has additional two panels showing the vertical distributions normalized to the surface PMx concentrations.

**730 You know it overestimated PM10, not sure about aerosol number concentration (you would need a different observation for assessing that)**

Response: We do not have the surface measurement of aerosol number concentration. The amounts of quality assured retrievals of aerosol columnar volume and effective radius by CE318 are limited in DAO-K (<9 days, 1 to 4 data samples per day). It is difficult to give a robust verification. The original statement describes that the GSI tends to increase the aerosol number in response to the high aerosol mass concentration. We changed the statement to:

"The revised GSI updates aerosol number concentration according to the analyzed aerosol mass concentration and the background ratio between mass and number concentrations. Thus, an overestimation of aerosol mass concentration inclines to raise aerosol number concentration, resulting in high scattering/absorption coefficients."

**734. Use single-scattering albedo for this**

Response: We add a new table 4 to show the angstrom exponent and SSA. In the revised section 3.3:

"Additionally, we computed the surface single scattering albedo (SSAsrf) with the 525 nm scattering coefficient and 520 nm absorption coefficient. We did not use the Ångström exponent to interpolate the scattering/absorption coefficients to a similar wavelength because the AE itself had a large model bias even after DA (Table 3). The observed SSAsrf value was 0.78, indicating an emphatic absorption particle,

probably due to the mixture of anthropogenic black carbon and natural desert dust in the local air. The model background SSAsrf was 0.86, while the DA analyses gave even higher SSAsrf (0.88 to 0.9)."

738-746. This is a misconception, aerosol light extinction and AOD does not depend on sun light intensity (for instance, you can sample both at night time with different methods). What's going to change with sunlight are the radiative effects. There are likely other reasons to explain this diurnal behavior. Look into the diurnal evolution of your BEC, and also into diurnal evolution of dust reaching the city. Similar misconnection is mentioned in lines 770-771.

**Response: In the revised 3.3,**

"Assimilating the AOD seems to increase the diurnal variation in the DA analyses, but this variation was not conclusive since there were different amounts of AOD data for DA at 00:00, 06:00, and 12:00. The AOD data were not always available as the data quality control (i.e., cloud screening). There was a higher increase in the concentration at noon (06:00 UTC) (Figure 9b), corresponding to a few high AOD during mild dust episodes at that hour. ..."

The misstatement in the original lines 770-771 has been removed.

**960-963. This is probably due to underprediction of dust imaginary refractive index**

Response: We set the dust refractive index to refer to the generic model values in literature. The imaginary part in our study is higher than the imaginary part for the Taklimakan desert dust retrieved by Di Biagio (2019). The strong bias in absorption coefficient can be largely removed by tuning the background error of black carbon, though additional disadvantage is introduced. Please refer to the revised section 3.5.

**Minor Edits**

Fig 5 caption. It reads like a) and b) represent PM10 and winds, respectively, but I think that's not the case. Please revise

**Response: The figure caption is changed to**

"Figure 5. Monthly mean  $PM_{10}$  concentration ( $\mu g m^{-3}$ ) and the streamlines of the 10-m wind ( $m s^{-1}$ ) in April (a, b) and their daily mean anomalies (c, d) ..."

**623. Did you mean "underestimates" instead of "lowered"?**

Response: changed to "underestimates"

780. Do you mean "particles that absorb radiation" rather than "aborting particles"? Also, I would like black carbon in that list as well.

**Response: Corrected.**

781-791. I believe primary dust in WRF-Chem is also considered to be a bit absorbing (has a imaginary refractive index above 0). As mentioned in a previous comment, this number might be too low for dust in this region.

Response: In the first version manuscript, the imaginary part of dust was in the range of 0.002 to 0.001. In the revision, we increase the imaginary part to the range of 0.003 to 0.001. Our imaginary part is higher than the imaginary part for the Taklimakan desert dust retrieved by Di Biagio (2019). The strong bias when assimilating the absorption coefficient can be largely removed by tuning the background error of black carbon. Please refer to the revised section 3.5.

802-804. I disagree with this statement. If the model has biases that the assimilation is not able to correct (for instance, inaccurate real and imaginary refractive indexes) then assimilating multiple observation could also create unrealistic modifications to the model. Response: The statements have been removed in the revision.

806-828. I wouldn't put DA\_Esca\_Eabs as an improvement over DA\_Esca, they show pretty much the same results. This means that the absorption observations are not really generating any differences in the results. Also, DA\_PMx\_AOD matches better the assimilated variables (which off course is expected) and the better agreement with scattering you happened to underpredict it with PM assimilation, and overpredict it with AOD assimilation, so assimilating both yields you something in between. Response: Because of the problem of individual assimilation of the absorption coefficient, we remove the DA results of DA\_Esca\_Eabs and DA\_PMx\_Esca\_Eabs\_AOD and just keep the result of DA\_PMx\_AOD.

832-834. As mentioned earlier, it would be better to check this using normalized profiles. The background profiles already had aerosols up to 4km, so is likely that the assimilation is just scaling this profile upwards rather than adding a larger fraction of the mass in these layers Response: The revised figure 12 shows the normalized profiles. In the revised section 3.7:

"Also shown in the figure are the vertical profiles normalized to their own respective surface particulate concentrations. The assimilations not only added a larger fraction of the mass in these layers but also adjusted the shapes of the  $PM_{10}$  profiles within 3 km above the ground (Figure 11d), following the BEC's vertical correlation length scales (Figure 3r)."

Section 4.1. I don't think this section is very relevant, the aerosols are so dominated by dust and your BEC is constructed in a way dust aerosols will be the ones largely modified. So just briefly mentioning that the composition of these other aerosols doesn't change would do.

Response: We shorten the revised section 4.1 and present the new anthropogenic aerosols' results.

Section 4.2. Might want to discuss in this section how the large underprediction of dust absorption would impact these results.

Response: It is not easy to quantify the ADRF bias due to the weak absorption with a single WRF-Chem experiment. We admit this uncertainty and add a new paragraph in section 4:

"It is noteworthy to say that the ADRF estimation remains uncertain even after DA. The AOD observation is only sporadically available because of cloud screening in retrieval data. The DA experiments cannot

eliminate the low bias in AOD in WRF-Chem. The ADRF values in the DA experiments are likely to be weaker than the plausible aerosol radiative forcing at Kashi. Neither DA experiment lowers SSAsrf to approach the observation. The observed SSAsrf (0.78) indicates likely warming forcing of aerosol at Kashi, while WRF-Chem and the DA analyses impose cooling forcing. The ADRF uncertainty is associated with the background aerosols. WRF-Chem simulates aerosol size up to 10  $\mu$ m, whereas larger particles (>10  $\mu$ m) exhibit substantial absorption relative to scattering in the visible wavelength (Kok et al., 2017). Anthropogenic emission inventories need an update for the year 2019, which may reduce the potential low bias in BC concentration. Additionally, the revised GSI does not concern the change in particle effective radius per size bin when calculating the aerosol number concentration in each outer loop. Low absorption cross section rises aerosol number concentration as compensation, increasing aerosol scattering coefficient too much. If our tangent operator concerns the change in particle effective radius per size bin, we can use aerosol mass and number concentration as control variables simultaneously. The DA would have a higher degree of freedom to balance the particle radius and number concentration and improve the absorption coefficient. All these need further research in the future." The authors appreciate the reviewer's constructive and friendly comments. We have substantially revised the manuscript. New data and figures are present in the main text. A new supplementary document is included in the revision. We reply to the reviewer's comments point by point.

**Anonymous Referee #2**

**Received and published: 28 October 2020**

The study of Chang et al. developed the GSI 3Dvar capability to assimilate AOD, scattering/absorbing coefficients for MOSAIC scheme. A few DA tests (both simultaneously and separately experiments) were conducted for northwestern China and compared with surface observations at Kashi. The authors should have spent great efforts on the system development and presented very comprehensive results.

Based on my current understanding, some more work need to be done to facilitate the readers to understand, including some essential considerations of the DA core details and the clarifications of the texts. In this way, the system would be better under- stand/promoted and readers would be more convinced.

**My general comments are as below:**

1. Actually GOCART is understood for the better performance of dust simulation and the relevant optical properties had been well verified; while the MOSAIC scheme is thought to be more suitable for anthropogenic emission related simulation, but the optical simulation is rather complex.

Response: Agree. The GOCART dust emission scheme is popular for dust simulation. Here, we applied the GOCART dust scheme to simulate the dust and used the MOSAIC scheme to simulate anthropogenic aerosols.

In this study, the system is developed for MOSAIC but the verification is conducted for a site in desert. This required intensive investigation of the DUST related properties representation in the MOSAIC scheme, for example,

**(a) the refractive index of OIN since it is mostly treated as DUST (while there should be distinctive differences between the two);**

Response: Yes, the OIN is not equivalent to dust. WRF-Chem has a dust option (dust\_opt=13) for simultaneous simulation of dust and anthropogenic aerosols with the GOCART dust scheme and the MOSAIC scheme, respectively. With this option, dust is added to OIN. Surely, this simplification is not perfect, but it did not hinder our verification of the DA system. In fact, even using the GOCART aerosol scheme, WRF-Chem computes aerosol optics with the Mie theory. Improving the dust representation in WRF-Chem needs further code development.

(b) the species partitioning (NO3 is not changed in option 2 which might not be reasonable and lead to unbalanced chemistry partitioning),

Response: NO3 is one of the control variables in the revision. We redid the DA experiments.

(c) the size distribution, (d) the number concentration, since the three factors determining the absorbing and scattering efficiency;

Response: We used multi-wavelength aerosol optical measurements to verify the DA system. The revised manuscript additionally shows the angstrom exponent result in Table 3.

(e) aerosol water content which are not considered but actually may change the optical properties. With very limited observational data to verify the above-mentioned information, the results in this study is really hard to interpret.

Response: Aerosol water content (AWC) is not a control variable in DA but is diagnosed in the GSI system according to the hygroscopic growth scheme, based on the analyzed aerosol dry mass concentrations. This treatment ensures the change in AWC is a physical constraint. Besides, AWC is low in the desert site and does not affect AOD a lot. In the revision, we dig the analyses by studying angstrom exponent, SSA, mass extinction coefficient. Hope the revised manuscript is convincible.

2. Some descriptions about DA core and observational data should be provided. For example, it seemed not only AOD, but also wavelength depended absorbing and scattering efficient were all assimilated, the corresponding observational operators and the errors should be given in more detail. Response: Sorry for the confusion. The observational operators of scattering/absorption coefficient are implicitly involved in the operator of AOD in equation (3). In the revised section 2.2.3, we explicitly

implicitly involved in the operator of AOD in equation (3). In the revised section 2.2.3, we explicitly present the two observational operators in equation (4). We rewrote the statements about observation errors in the revised section 2.4:

"The observation errors of PMx are handled in the conventional way (Schwartz et al., 2012; Chen et al., 2019), which contains the measurement error (e1) and the representative error (e2). The measurement error is the sum of a baseline error of  $1.5 \ \mu g \ m-3$  and 0.75% of the observed PMx concentration. The representative error is the measurement error multiplied by the half-squared ratio of the grid spacing to the scale distance. The scale distance denotes the site representation in GSI and has four default values of 2, 3, 4, and 10 km, corresponding to the urban, unknown, suburban, and rural sites. We used 3 km for the scale distance in this study. As we had a single site in Kashi, it is difficult to estimate the site representation of 5 km, close to the site distance to the Kashi urban area, we assumed the aerosol optical measurement had good representativeness of the model grid covering the site. The observation error of CE318 AOD took the AERONET AOD uncertainty of 0.01 in cloud-free conditions (Holben et al., 1998). The AOD observational error was further divided by the total model layer thickness in GSI."

**Comments by lines: 1. Line 65 and other places. Adjoint operator, is it referred as TL-AD? Please clarify Response: We corrected the statements to "tangent linear".**

**2. Line 86 GIS ? Response: Corrected to "GSI"**

3. Line 100 Zang et al 2016, acutally a different DA system was used other than GSI in this study. Please check.

Response: We corrected the statements in the revised introduction:

"Li et al. (2013) developed a 3D-Var scheme for assimilating the surface  $PM_{2.5}$  and speciated aerosol chemical concentrations for the WRF-Chem MOSACI aerosols. Zang et al. (2016) applied this scheme to incorporate aircraft speciated aerosols in California. They proved that the assimilation of aircraft profile extended the DA benefit to aerosol forecast."

**4. Line 181 regarding of the low anthropogenic and biogenic emissions in the desert, why not use GOCART instead?**

Response: The research purpose is to introduce the new GSI system to work with the MOSAIC aerosol scheme. We used the GOCART scheme to simulate dust. In the revised section 2.1:

"The dust emission was simulated using the GOCART dust scheme (Ginoux et al., 2001), and the dust mass was included in the OIN concentration. We performed the MOSAIC aerosol simulations with foursize bins (0.039–0.156  $\mu$ m, 0.156–0.625  $\mu$ m, 0.625–2.500  $\mu$ m, and 2.5–10.0  $\mu$ m dry diameters) for the anthropogenic aerosols."

5. Line 184-190. Actually the optical properties of NH4SO4, OC, dust, NaCl, H2O are treated as wavelength depended in the model, this information should be investigated and provided. As it seemed that multi-wavelength aerosol scattering and absorption coefficients are assimilated. The uncertainties of the assumption in the model and observational data should be provided.

Response: In the revision, we give the complex refractive index in table S1 in the supplementary document; Section 2.2.6 describes the refractive index; The revised section 2.4 describes the observational errors; the revised section 3.3 and section 4.2 states the uncertainties associated with dust morphology and aerosol radiative forcing. Hope the revisions make the manuscript more complete.

6. Line 228. Why NO3 is not considered? In this case, it may lead to unbalanced chemistry partitioning. Response: Nitrate is a control variable in the revision.

7. Section 2.2.3 It seemed that scattering and absorbing coefficients are also observational assimilated. Please provide details.

Response: The revised section 2.2.3 provides the observation operators of scattering/absorption coefficients.

**8. Line 101: are the Mi,z,k in the two terms the same, maybe possibly dry and wet mass concentration respectively? If not, please clarify.**

Response: Mi,z,k denotes the aerosol composition. It could be aerosol water content when calculating the internal mixing refractive index. In the revised section 2.2.4,

"Note that the dry  $(r_{dry,z,k})$  and wet  $(r_{wet,z,k})$  particle radiuses are both present in Eq (21). Because aerosol water content is not a control variable,  $r_{dry,z,k}$  is used in Eq (19) and appears in Eq (21). Aerosol water

content participates the computation of internal mixing refractive indexes, and thus  $r_{wet,z,k}$  is also present in Eq (21)."

**9. Line 315: is rwet related with aerosol water content, considering the hygroscopicity? Any uncertainty by not considering aerosol water content. Please clarify.**

Response:  $r_{wet}$  is the wet particle radius when aerosol water content (AWC) is counted in the aerosol composition. At the end of the revised section 2.2.1:

"The AWC was diagnosed according to the analyzed aerosol mass concentration and the background relative humidity in each DA outer loop. The hygroscopic growth was calculated using the WRF-Chem code coupled with the revised GSI."

**10. Line 352, please clarify mizk as dry or wet mass?**

Response: Mi, *z*, *k* denotes the aerosol compositions. It could be aerosol water content when calculating the internal mixing refractive index.

**11. Line 367. Any uncertainty by considering constant radius?**

Response: It is hard to estimate the uncertainty of this constant radius in this study. Appling this constant radius is to simplify the mathematical derivation of the tangent linear operator for AOD. This simplification was applied by Saide et al. (2013). We hope to remove this assumption in the future and could discuss the relevant uncertainty.

Saide, P. E., Carmichael, G. R., Liu, Z., Schwartz, C. S., Lin, H. C., da Silva, A. M., and Hyer, E.: Aerosol optical depth assimilation for a size-resolved sectional model: impacts of observationally constrained, multi-wavelength and fine mode retrievals on regional scale analyses and forecasts, Atmos. Chem. Phys., 13, 10425-10444, doi:10.5194/acp-13-10425-2013, 2013.

**12. Line 703-706. Please dig more on this issue.**

Response: We add a paragraph in the revised section 3.3:

"The irregular morphology had a significant influence on the dust simulation. Okada et al. (2001) found that the aspect ratio (the ratio of the longest dimension to its orthogonal width) of the mineral dust particles (0.1-6  $\mu$ m) in China arid regions exhibited a median of 1.4. Dubovik et al. (2006) suggested the aspect ratio of ~1.5 and higher in desert dust plumes. Kok et al. (2017) found that the dust' sphericity assumption underestimated dust extinction efficiency by ~20–60% for the dust particle larger than 1 $\mu$ m. Tian et al. (2020) found that using a dust ellipsoid model could increase the concentration of coarse dust particle (5-10  $\mu$ m) by ~5% in eastern china and ~10% in the Taklimakan area because of the decrease in gravitational settling, comparing with the simulations with dust sphericity model. Nevertheless, the aspect ratio of the spheroid dust is uncertain. Even after applying the spheroidal approximation, Soorbas et al. (2015) found that the model underestimated 550 nm aerosol scattering and backscattering values by 49% and 11%, respectively, because of the uncertainties in particle axial ratio, complex refractive index, and the particle size distribution. To date, the assumption of spherical particles has been widespread in models (including WRF-Chem) for computational efficiency. Impact of dust morphology to DA deserves a further investigation."

13. Line 765. Please investigate the uncertainties of the modeled and observed absorption coefficients. Response: We check the differences in DA analysis as using the different imaginary part of dust refractive index and background error of BC. Please refer to the revised section 3.5.

**14. Figure 2. Why the domain averaged standard deviation (c) is significantly larger than that of column averages (d, e)?**

Response: The vertical profiles in figure 2(c, d, e) are based on different grids. As shown in Figure 5c, Kashi and the desert point we picked up for figure 2(e) are not on the track of dust storm. Thus, the dust variations at the two points (figure 2d, e) are smaller than the average of binning standard deviation (figure 2c).

**15. Figure 3. Why background error standard deviation of the OIN is two magnitudes larger than the other species? Indicating dominating contribution of dust? In this case, is it meaningful to investigate other species changes?**

Response: We accidently lowered anthropogenic aerosols in Kashi. The revised simulations correct the emissions and show that the OIN is still the predominant composition, accounting for 62% of  $PM_{2.5}$  and 82% of  $PM_{10}$  in April. The qualitative conclusion is the same.

**16. Table 1. Please explain how the errors are determined?**

Response: In the revised section 2.4:

[revised manuscript text omitted]
. We set the control variables of six 256 aerosol mass mixing ratios of SO42-, NH4+, NO3-, OC, BC, and OIN per size bin. Chlorine, 257 and sodium had miniscule background concentrations and remained the background values in 258 the DA analysis. There were twenty-four control variables in total for the four-size bin 259 simulations, In Kashi's case near the desert, the OIN was predominant, accounting for 62% of 260 PM2.5 and 82% of PM10. 261 262 Our design of the control variables was different from the AOD assimilation in Saide et al. 263 (2013), with theirs being the natural logarithm of the total mass mixing ratio per size bin, 264 multiplied by the thickness of the model layer. As the high model layer had a significant layer 265 thickness with low aerosol concentrations, the multiplication offset the opposite effects of 266 increasing layer thickness versus decreasing concentrations with increase in altitude. This 267 multiplication prevented the addition of many modifications for the high model layers, where 268 aerosols were low in concentration. The logarithmic transformation was used to decrease the 269 extensive value range in the control variables caused by multiplication. Since the AOD value 270 is often smaller than one, this leads to a significant negative logarithm value and a relatively 271 unconstrained DA system. Saide et al. (2013) introduced two weak constraints in their cost 272 function to cut off the user-defined "extraordinarily high" and "extraordinarily low" 273 concentrations. They repartitioned the increments of the total mass per size bin for 274 composition of each aerosol, with the background aerosol chemical mass fractions. Here, 275 neither the logarithmic transformation, nor the multiplication using layer thickness was set in 276 our DA system. Our control variable was restricted to the WRF-Chem output variable, and the 277 DA system changed the composition of each aerosol per size bin, depending on the aerosol 278 background errors. 279 280 Consistent with the set by Pang et al. (2020), aerosol water content (AWC) was not one of the 281 control variables in our GSI. Otherwise, the AWC might have increased contrary to the 282 physical constraints for the loading of hydrophilic particles, and simply as a mathematical 283 artefact. The AWC was diagnosed according to the analyzed aerosol mass concentration and 284 the background relative humidity in each DA outer loop. The hygroscopic growth was

- 285 calculated using the WRF-Chem code coupled with the revised GSI.
- 286

Deleted: In the case of Kashi situated near the desert, the OIN was predominant, accounting for ~99% of the total particle mass concentrations. The control variable could thus have exclusively comprised the OIN. However, because we were curious about the response of aerosol chemical fractions in the DA constraint, w

**Deleted: five**

Moved (insertion) [3]

Moved up [3]: Saide et al. (2013) repartitioned the increments of the total mass per size bin for composition of each aerosol, with the background aerosol chemical mass fractions. Our control variable

102.2.2 Jangent Linear Operator for PM.Defends and the sum of all across of the PM. (concentration over the size bins, and the sum of
the first free site PM.); (cone at al., 2019; Wang et al., 2020; Accordingly, the Pargent
linear operator for PM.) is the gradient of the PM, concentration to the across of chemical mass
concentration per size bins.Defects: Adjain07
$$\frac{\partial_{12}PM_{11}}{\partial_{1}(corral}} + t = 1, ..., n_{Rise}$$

(2)(2)00where  $n_{dec}$  is the number of size bins and is equal to four in this study; [] denotes the most
concentration ( $\mu_{11} = h^{-1} h_{11} h_{12} e^{-1} h_{12} h_{12} e^{-1} h_{13} h_{12} e^{-1} h_{13} h_{12} e^{-1} h_{13} h_{13} h_{13} e^{-1} h_{13} h_{13}$

343 344  $e_{ext,z,k} = e_{sca,z,k} + e_{abs,z,k}$ 345 (5) 346 347 The extinction cross section  $e_{ext,z,k}$  of a wet particle with radius  $r_{wet,z,k}$  is: 348 349  $e_{ext,z,k} = p_{ext,z,k} \cdot \pi \cdot r_{wet,z,k}^2$ 350 (6) 351 352 where  $p_{ext,z,k}$  is the extinction efficiency, given the desired mixing refractive indexes and the Deleted: coefficient 353 wet particle radius. The  $p_{ext,z,k}$  is attained through the Chebyshev polynomial interpolation: 354  $p_{ext,z,k} = \exp \{\sum_{j=1}^{n_{coef}} c_{ch}(j) \cdot c_{ext,z,k}(j)\}$ 355 356 (7) 357 where  $c_{ch}$  is the coefficient of  $n_{coef}$  order Chebyshev polynomials,  $c_{ext,z,k}$  is the polynomial 358 value for the extinction efficiency of the particle, which is an internal mixture of all aerosol 359 compositions (i.e., the control variables plus chlorine, sodium, and AWC). The radius in the Deleted: nitrate, 360 AOD subroutine code is in a logarithmic transform to handle the broad particle size range 361 from 0.039 µm to 10 µm. The exponential function in Eq. (7) transforms the logarithm radius 362 back to the normal radius. The aerosol number concentration  $n_{z,k}$  and the aerosol dry (wet) 363 mass concentration  $m_{i,z,k}$  have a linkage through the dry (wet) particle radius  $r_{dry,z,k}$  ( $r_{wet,z,k}$ ) and 364 the density  $\rho_i$  of each aerosol chemical composition: 365  $n_{z,k} = \sum_{i}^{n_{wet\_aer}} \frac{m_{i,z,k}}{\rho_i} \cdot \frac{3}{4\pi \cdot r_{wet,z,k}^3} = \sum_{i}^{n_{dry\_aer}} \frac{m_{i,z,k}}{\rho_i} \cdot \frac{3}{4\pi \cdot r_{dry\_z,k}^3}$ 366 367 (8) 368 369 Both the dry and wet particle radius will appear in the tangent linear operator. The difference 370 between the second and the third terms in Eq (8) is whether aerosol water content is counted. 371  $n_{wet aer}$  is the number of aerosol chemical composition plus aerosol water content ( $n_{wet aer}$ 372  $\underline{n_{dry\_aer}+1}$ ). 373 374 2.2.4 Tangent Linear Operator Developed for AOD **Deleted:** Adjoint 375 As per the forward operator in Eq. (3) in WRF-Chem, we developed the tangent linear Deleted: adjoint 376 operator for AOD, which requires the derivative of  $\tau$  in Eq. (3) to the aerosol dry mass 377 concentration (aerosol water content is not a control variable),  $m_{i,z,k}$ : 378  $\frac{\delta \tau}{\delta m_{i,z,k}} = \frac{\delta \tau_z}{\delta m_{i,z,k}} = \frac{\delta e_{ext,z,k} \cdot n_{z,k} \cdot H_z}{\delta m_{i,z,k}} + \frac{e_{ext,z,k} \cdot \delta n_{z,k} \cdot H_z}{\delta m_{i,z,k}} + \frac{e_{ext,z,k} \cdot n_{z,k} \cdot \delta H_z}{\delta m_{i,z,k}}$ 379 380 (9) 381

The first term on the righthand side of Eq. (9) indicates the change in AOD as the perturbation
of extinction cross section. According to Eq. (6), considering that the particle radius is
constant,
$$\delta e_{ext,x,k} = \delta p_{ext,x,k}$$
,  $\pi \cdot r_{wet,x,k}^2$  (10)
 $\delta e_{ext,x,k} = \delta p_{ext,x,k}$ ,  $\pi \cdot r_{wet,x,k}^2$  (10)
 $\delta e_{ext,x,k} = \delta p_{ext,x,k}$ ,  $\pi \cdot r_{wet,x,k}^2$  (10)
 $\delta e_{ext,x,k} = \delta p_{ext,x,k}$ ,  $\pi \cdot r_{wet,x,k}^2$  (10)
 $\delta e_{ext,x,k} = \delta p_{ext,x,k}$ ,  $\pi \cdot r_{wet,x,k}^2$  (10)
 $\delta e_{ext,x,k} = \delta p_{ext,x,k}$ ,  $\pi \cdot r_{wet,x,k}^2$  (10)
 $\delta e_{ext,x,k} = p_{ext,x,k}$ ,  $\{\sum_{j=1}^{n_{corf}} c_{ch}(j) \cdot \delta c_{ext,x,k}(j)\}$
 $\delta p_{ext,x,k} = p_{ext,x,k}$ ,  $\{\sum_{j=1}^{n_{corf}} c_{ch}(j) \cdot \delta c_{ext,x,k}(j)\}$  (11)
By expanding  $\delta c_{ext,x,k}$  in Eq. (11), we have:
 $\delta p_{ext,x,k}(j) = \delta w_{00} \cdot E_{ext,00}(j) + \delta w_{01} \cdot E_{ext,01}(j) + \delta w_{11} \cdot E_{ext,11}(j)$  (12)
 $\delta c_{ext,x,k}(j) = \delta w_{00} \cdot E_{ext,00}(j) + \delta w_{01} \cdot E_{ext,01}(j) + \delta w_{11} \cdot E_{ext,11}(j)$  (12)
 $\delta c_{ext,x,k}(j) = \delta w_{00} \cdot E_{ext,00}(j) + \delta w_{01} \cdot E_{ext,01}(j) + \delta w_{11} \cdot E_{ext,11}(j)$  (12)
 $\delta c_{ext,x,k}(j) = \delta w_{00} \cdot E_{ext,00}(j) + \delta w_{01} = (1 - v_j \delta u - u \delta v)$
 $\delta w_{00} = (v - 1) \delta u + (u - 1) \delta v$   $\delta w_{01} = (1 - v_j \delta u - u \delta v)$
 $\delta w_{00} = (v - 1) \delta u + (u - 1) \delta v$   $\delta w_{01} = (1 - v_j \delta u - u \delta v)$
 $\delta w_{010} = (1 - u_j \delta v - v \delta u)$   $\delta w_{11} = u \delta v + v \delta u$
 $\delta w_{10} = (1 - u_j \delta v - v \delta u)$   $\delta w_{11} = u \delta v + v \delta u$
 $\delta w_{11} = \frac{R_{mix} - R_{low}}{R_{up} - R_{low}}}$   $\delta v = \frac{\delta R_{mix}}{R_{up} - R_{low}}}$
 $\delta v = \frac{\delta R_{mix}}{R_{up} - R_{low}}$   $\delta v = \frac{\delta R_{mix}}{R_{up} - R_{low}}}$  (14)
In Eq. (14),  $R_{mix}$  and  $I_{mix}$  are the aerosol volume-weighted mean real and imaginary parts of
complex refractive indices, respectively,  $R_w (I_w)$  and  $R_{bw} (I_{bw})$  are the nearest upper and
lower limits for  $R_{mix} (I_{mix})$  in the Mie table. Considering  $V_{wet,x}$  is the volume of all acrosol dry,
masses plus acrosol water content, the real and imaginary parts and their drivatives are:
233

424
$$R_{mix,z,k} = \sum_{i} R_i \cdot \frac{m_{i,z,k}}{\rho_i \cdot V_{wet,z,k}} \qquad \delta R_{mix,z,k} = \frac{R_i}{\rho_i \cdot V_{wet,z,k}} \cdot \delta m_{i,z,k}$$
10

$$\begin{array}{cccc} 425 & I_{mix,z,k} = \sum_{i=1}^{n_{wet,aer}} I_i \cdot \frac{m_{i,z,k}}{\rho_i \cdot V_{wet,z,k}} & \delta I_{mix,z,k} = \frac{I_i}{\rho_i \cdot V_{wet,z,k}} \cdot \delta m_{i,z,k} \end{array}$$

$$\begin{array}{c} 426 & (15) \\ 427 \\ 428 & \text{where} \\ 429 & V_{wet,z,k} = \sum_{i=1}^{n_{wet,aer}} \frac{m_{i,z,k}}{\rho_i} \\ 430 & (16) \\ 431 \\ 432 & \text{Put Eq. (12), Eq. (13) into Eq. (11) leads to:} \\ 433 \\ 434 & \delta p_{ext,z,k} = [(v-1)\alpha_{sca,00} + (1-v)\alpha_{sca,01} - v\alpha_{sca,10} + v\alpha_{sca,11}]\delta u + \\ [(u-1)\alpha_{abs,00} - u\alpha_{abs,01} + (1-u)\alpha_{abs,10} + u\alpha_{abs,11}]\delta v \\ 437 & \text{where} \\ 438 & \alpha_{sca,00} = p_{sca,1,k} \cdot \sum_{i=1}^{n_{coef}} c_{ch}(j) \cdot E_{sca,00}(j) \\ 439 & \alpha_{sca,10} = p_{sca,1,k} \cdot \sum_{i=1}^{n_{coef}} c_{ch}(j) \cdot E_{sca,10}(j) \\ n_{coef} & n_{coef} \\ \end{array}$$

$$\begin{array}{ll}
440 & \alpha_{abs,00} = p_{abs,1,k} \cdot \sum_{j=1}^{n_{coef}} c_{ch}(j) \cdot E_{abs,00}(j) & \alpha_{abs,01} = p_{abs,1,k} \cdot \sum_{j=1}^{n_{coef}} c_{ch}(j) \cdot E_{abs,01}(j) \\
441 & \alpha_{abs,10} = p_{abs,1,k} \cdot \sum_{j=1}^{n_{coef}} c_{ch}(j) \cdot E_{abs,10}(j) & \alpha_{abs,11} = p_{abs,1,k} \cdot \sum_{j=1}^{n_{coef}} c_{ch}(j) \cdot E_{abs,11}(j) \\
442 \\
443 & (18)
\end{array}$$

(18)

445 The subscripts of *sca* and *abs* in Eq. (17) and (18) denote "scattering" and "absorption", 446 respectively. The first term on the righthand side of Eq.  $(\underline{9})$  is determined using Eq.  $(\underline{10})$  and 447 Eq.  $(\underline{17})$ . The second term on the righthand side of Eq.  $(\underline{9})$  indicates the linkage of the aerosol 448 number and mass concentrations. It is the derivative of  $\frac{dry \text{ particle in }}{Eq. (\underline{8})}$  by assuming a

444

$$\begin{array}{l}
451 \quad \delta n_{z,k} = \frac{1}{4\pi \cdot r_{dyr,z,k}^3 \cdot \rho_i} \\
452 \\
453
\end{array} \tag{19}$$

454 The third term on the righthand side of Eq.  $(\underline{9})$  contains the derivative of the layer thickness to 455 the concentrations in this layer. This indicates that the light attenuation length based on per

456 unit concentration, which can be intuitively represented by the ratio of layer thickness to the
457 aerosol mass concentration in this layer. Putting Eq. (19) and Eq. (19) into Eq. (9), we have
458 the original formula of the Langent linear operator for AOD for the aerosol dry mass
459 concentration:
460
461
462
$$\frac{\delta \tau}{\delta m_{i,x,k}} = \frac{\delta \tau_{x,x}}{\delta m_{i,x,k}} = \frac{\delta e_{ext,x,k} \cdot n_{x,k} \cdot H_x}{\delta m_{i,x,k}} + \frac{e_{ext,x,k} \cdot \delta n_{x,k} \cdot H_x}{\delta m_{i,x,k}} + \frac{e_{ext,x,k} \cdot n_{x,k} \cdot \delta H_x}{\delta m_{i,x,k}} = \frac{\delta \tau_{x,x}}{\delta m_{i,x,k}} + \frac{1 - \nu n_{x,x,n,0}}{\delta m_{i,x,k}} + \frac{1 - \nu n_{x,x,n,0}}{\rho_1 \cdot V_{wet,x,k} \cdot (R_{v,p,x,k} - R_{i,0w,x,k})} + \frac{1}{4}$$
463  $\{ [(\nu - 1)\alpha_{x,cn,00} + (1 - \nu)\alpha_{x,cn,01} - \nu \alpha_{x,cn,10} + \nu \alpha_{x,cn,11}] \cdot \frac{\pi \cdot r_{wet,x,k}^2 \cdot R_1 \cdot n_{x,k} \cdot H_x}{\rho_1 \cdot V_{wet,x,k} \cdot (L_{up,x,k} - L_{low,x,k})} + \frac{1}{4}$
465  $[(\nu - 1)\alpha_{abs,00} - u\alpha_{abs,01} + (1 - u)\alpha_{abs,1,0} + u\alpha_{abs,1,1}] \cdot \frac{\pi \cdot r_{wet,x,k}^2 \cdot (L_{up,x,k} - L_{low,x,k})}{\rho_1 \cdot V_{wet,x,k} \cdot (L_{up,x,k} - L_{low,x,k})} + \frac{1}{4}$
466  $\frac{3e_{ext,x,k} \cdot H_x}{4\pi \cdot r_{dy,x,k}^2 \cdot \rho_1} + \frac{e_{ext,x,k} \cdot n_{x,k} \cdot H_x}{m_{x,x}} \cdot \rho_1} + \frac{e_{ext,x,k} \cdot n_{x,k} \cdot H_x}{m_{x,x}}} + \frac{e_{ext,x,k} \cdot n_{x,k} \cdot H_x}{(20)}$
470 where  $\beta$  is the factor that changes the unit of mass from µg kg-1 to µg m-3. The last righthand term in Eq. (20) may not have a quick convergence in the DA outer loops because the aerosol mass concentration  $m_{t,x}$  in the denominator often has a low bias, which introduces an error in to the operator. The error is amplified by the layer thickness H, in the numerator. Thus, the perator of Eq. (20) cannot lead to a stable analysis. For this reason, we changed the permutor to account for the columnar mean aerosol extinction coefficient which is described as follows:
477  $\frac{\delta (e_{ext} \cdot n)}{\delta m_{t,x,k}}} = \frac{H_x}{\Sigma H_x} \cdot \frac{\delta (e_{ext,x,k} \cdot n_{x,k}}{\delta m_{t,x,k}}} + \frac{e_{ext,x,k} \cdot n_{x,k}}{\delta m_{t,x,k}} + \frac{e_{ext,x,k} \cdot n_{x,k}}{\delta m_{t,x,k}} + \frac{e_{ext,x,k} \cdot n_{x,k}}{\delta m_{t,x,k}} + \frac{e_{ext,x,k} \cdot n_{x,k$

by observation location. Note that the dry
$$(r_{max,0})$$
 particle radiuses are both,
present in Eq. (2). Because aerosol water content is not a control variable,  $r_{p_{max}}$  is used in Eq.
(19) and appears in Eq. (2). Acrosol water content participates the computation of internal
mixing refractive indexes, and thus  $r_{max}$  is also present in Eq. (2). Equation (2) is the final
angeent linear operator for AOD DA in this study.
**Detect:** adjoint
The aerosol scattering and absorption coefficients measured by the nephelometer and
actual and absorption coefficients measured by the nephelometer and
actual meta-operator for the two coefficients measured by metholometer is
detended scattering coefficient measured by metholometer is described as follows:
$$\frac{\delta(c_{max}, s^+, n_{k,k})}{\delta m_{n,k}} = \{l(v-1)a_{max}m + (1-v)a_{max}m + va_{max}, n + va_{max}, n \}, \frac{3e_{max}m_{max}}{4\pi + n_{max}^2, n + n_{max}^2}, \frac{3e_{max}m_{max}}{4\pi + n_{max}^2, n + n_{max}^2}, \frac{3e_{max}m_{max}}{2}, \frac{1}{2}$$

where the symbols have the same meaning as before, and the subscript once in Eq. (2)
denotes the surface layer. The operator for the aerosol absorption coefficient measured by
actual means of the cost function within the CSI. That is, the cost function is first
minimized with the first analysis contentruins in the aerosof
anumber concentration in the background field, and the number
concentration in updated with the first analysis contentruins. This iterative process in
denoted as the "arter loop," which first analysis contructuring in the cost function the first analysis
(Massart et al., 2010). We agate maximum interving the propriors first and the cool function the cost contructure in the aerosof
atomic at the "autor loop," which are presented several times to attain the first analysis
(Massart et al., 2010). We agate maximum interving the cost function first
minimized with the first analysis constructs a new
operator value, resulting in an examalysis of law subsurating at the present in the conter loops within the CSI.
The WRP-C

**541 2.2.6 Aerosol Complex Refractive Indexes in GSI**

542 Table S1 in the supplementary document shows the complex refractive indexes for each 543 aerosol chemical composition in the revised GSI. The refractive indexes are for eleven 544 wavelengths, including four for CE318, three for nephelometer, three for aethalometer, and 545 one for 550 nm MODIS AOD (not assimilated in this study). The real parts of refractive 546 indexes of sulfate, nitrate, and ammonium are similar and refer to Toon et al.'s (1976) data. 547 The real part is 1.53 at 440 nm and decreases to 1.52 at 1020 nm. The refractive indexes of 548 OC and BC are constant across the wavelengths, being 1.55–0.001*i* for OC (Chen and Bond, 549 2010) and 1.95–0.79i for BC (Bond and Berstrom, 2006). The dust refractive index's real part 550 is a constant value of 1.54 (Zhao et al., 2010). The dust refractive index's imaginary part 551 depends on the dust mineralogy, size distribution, and shape, which are associated with the 552 dust sources. The imaginary part varies a lot at the same dust source. 
[revised manuscript text omitted]
 emission low bias in the residential 790 sector which is a major source of anthropogenic emissions for PM2 s. BC, and OC in the
- 790 sector which is a major source of anthropogenic emissions for PM2.5, BC, and OC in the developing western area. The residential sector accounts for 36–82% of these emissions.
- developing western area. The residential sector accounts for 36–82% of these emissions,
   according to the MEIC emission inventory (Li et al., 2017) and is the primary source of
- uncertainty in anthropogenic emission inventories in China.
- uncertainty in anthropogenic emissions inventories in China.

Figure 5

795

**796 3.2 Assimilating PM2.5 and PM10 Concentrations**

797 Simultaneous assimilation of the observed PMx (DA\_PMx) improved both the fine and 798 coarse particle concentrations, with a substantial improvement in the third and fourth particle 799 sizes of the OIN composition (Figure 8f). The analyzed monthly mean PM10 increased to 800 329.3  $\mu$ g m-3, with a high correlation of 0.99. The analyzed monthly mean PM2.5 was 801 improved to  $89.3 \,\mu g \, m^{-3}$ , although it was still lower than the observed levels, with a high correlation of 0.89. The low bias in PM2.5 and the high bias in PM10 in the analyses were both 802 803 mainly in the dust storm on 24-25 April (Figure 6a, d). Applying the inter-size bin correlation 804 length caused the interlinked analyses of PM2.5 and PM10. In the desert area, the coarse and 805 fine dust are readily affected by the magnitude of BEC of the fourth size-bin OIN (oin\_a04). 806 We intentionally decreased the BEC of oin a04 by 10% each time to 30% of its original 807 value. The magnitude of 30% of oin a04 was comparable to the magnitude of the third size-808 bin (oin a03) OIN's background error. As shown in Table S3, because the oin a04's BEC 809 reduction relaxes the constraint on the coarse particle, the PM10 bias becomes more negative 810 along with the decrease in on\_a04's BEC. Meanwhile, the PM2.5 bias becomes more positive. 811 Correspondingly, the ratio of PM2.5 to PM10 was exaggerated to 0.33 with 30% of oin\_a04's 812 BEC, higher than the observed value of 0.28. Overall, the original BEC of oin\_a04 is a

813 reasonable tradeoff in our DA experiments. The inter-size bin correlation length tunes the

816 cross size-bin modifications, and it indeed does matter to the DA performance compared with

817 those without inter-size bin correlation. Although the correlation length of four in our DA

818 experiment is a little bit arbitrary, we found that the impact on the analysis due to using

- 819 different correlation length is almost ignorable.
- 820

821 The DA system preferentially modified the coarse particle concentrations because of the

coarse particles's high background model error according to our BEC modeling strategy.

823 Intuitionally, our modification that mainly focused on the highest concentration of coarse

- 824 particles was reasonable. It decreased the model biases by raising the heaviest loading
- aerosols. As a result, the ratio of  $PM_{2.5}$  to  $PM_{10}$  decreased from 0.39 in the background to 0.27
- 826 in DA\_PMx, approaching the observed ratio of 0.28. Such improvement was consistent with
- 827 the correction required to the model desert dust in literature. Kok et al. (2011) found that
- 828 regional and global circulation models underestimate the fraction of emitted coast dust (>~5\_
- μm), overestimates the fraction of fine dust (<2μm diameter). Adebiyi and Kok (2020)
- 830 claimed that too rapid deposition of coarse dust out of the atmosphere accounts for the
- 831 missing coarse dust in models. Similarly, WRF-Chem assimilated too much smaller dust
- particles than the observed. According to Kashi's AOD between 440 nm and 1020 nm, the
- 833 observed Ångström exponent (AE) was 0.18 in this case, but the background value was 0.54
- (Table 3). DA\_PMx reduced the AE value to 0.30, a little improvement but not sufficient.
- 835

As the particle concentration increased, the 635 nm aerosol scattering coefficient in DA\_PMx\_

837 moderately increased to 170.4 Mm-1, with a high correlation of 0.89, still lower than the

838 observed level of 231.5 Mm-1. The analyzed 660 nm absorption coefficient was 15.8 Mm-1,

67% lower than observed levels, with a correlation of 0.42. The analyzed AOD showed a

840 monthly mean value of 0.38 in DA\_PMx, 42% lower than observed levels, with a low

841 correlation of 0.35.842

Figure 9a shows the diurnal concentrations of PM10 in the analyses in April. The observed
 PM10 showed a substantial variation at 18:00 UTC, the (local midnight). This substantial

nocturnal variation was partly owing to the dust storm that started on 24 April and ended the

next day. This midnight variation was also related to a nocturnal low-level jet. Ge et al.

- 847 (2016) pointed out that there was a nocturnal low-level jet at a height of 100–400 m, with a
- 848 wind speed of  $4-10 \text{ m s}^{-1}$  throughout the year in the Tarim Basin. They stressed that the low-
- 849 level jet broke down in the morning, transporting its momentum toward the surface, and
- 850 increased dust emissions. The nocturnal low-level jet increased the possibility of dust
- 851 particles moving towards the city at night, causing a high PM10 variation at 18:00 UTC. The
- diurnal changes in the DA analyses followed the observed levels, but had higher mean values.
- 853

**854 **3.3 Assimilating AOD**

- Assimilating AOD (DA\_AOD) improved the monthly mean 870 nm AOD to 0.59,
- approaching to the observed value of 0.66, with a high correlation of 0.98 (Figure 60). The
- monthly mean PM2.5 was improved to  $\frac{92.6}{2.6} \mu \text{g m}^{-3}$ , quite close to the observed level of 91  $\mu \text{g}$
- 858  $\underline{m}^{-3}$ , but the analyzed PM10 was 541.7  $\mu$ g m-3, 68% higher than the observed value. The DA
- system improved the AOD at the price of deteriorating the data quality of surface coarse

862 particle concentrations, Surface particle overestimations have been reported in previous 863 studies (Liu et al., 2011; Ma et al., 2020; Saide et al., 2020). In the arid area of Kashi, the ratio 864 of  $PM_{2.5}$  to  $PM_{10}$  therefore reduced to 0.17 in DA AOD, which was too far comparing with 865 the observed ratio of 0.28. 866 867 The revised GSI updates aerosol number concentration according to the analyzed aerosol 868 mass concentration and the background ratio between mass and number concentrations. Thus, 869 an overestimation of aerosol mass concentration inclines to raise aerosol number 870 concentration, resulting in high scattering/absorption coefficients. In Kashi, the analyzed 635 871 nm scattering coefficient in DA\_AOD was 222.6 Mm-1, slightly lower than the observed 872 value. The analyzed 660 nm absorption coefficient was 17.0 Mm-1, 64% lower than the 873 observed value. It indicates that WRF-Chem strongly underestimated the 874 scattering/absorption cross section. This underestimation resulted in too many coarse particles 875 as compensation to fit the observed AOD, and hence decreased the  $PM_{2.5}/PM_{10}$  ratio further. 876 877 Table 4 shows the ratios of the AOD and aerosol scattering/absorption coefficients to the 878 surface PM10 concentrations. The ratio of AOD to PM10 in the background model result was 879 one-third of the observed levels. The observed mass scattering coefficient (Esca/PM10) was 880  $1.05 \text{ Mm}^{-1} \mu \text{g}^{-1} \text{ m}^3$ , while the background value was only 0.65 Mm-1  $\mu \text{g}^{-1} \text{ m}^3$ . DA AOD 881 did not eliminate the low bias but enlarged the low bias to 0.51  $Mm^{-1}\,\mu g^{-1}\,m^3.$  The same 882 thing occurred for Eabs/PM10, which was 0.09 in the background and 0.05 in DA\_AOD, 883 much lower than the observed value of 0.25. Figure 10 shows these mean ratios at the other 884 wavelengths. The low bias in AOD/PM10 was comparable at each wavelength. All DA 885 experiments yielded close bias in extinction/scattering/absorption efficiency. Such low bias in 886  $AOD/PM_{10}$  imposed the DA system to overestimate the PM10 to fit the observed AOD data. 887 888 Additionally, 
[revised manuscript text omitted]
 up to 1.73. Nevertheless, the 1019 absorption coefficient ( $40 \text{ Mm}^{-1}$ ) was improved to the observed level ( $47.4 \text{ Mm}^{-1}$ ).
- 1019 absorption coefficient ( $\underline{40}$  Mm-1) was improved to the observed level ( $\underline{47.4 \text{ Mm}^{-1}}$ ). 1020
- 1020
- 1021 Improving the absorption coefficient at the cost of PM10 overestimation indicates the model
- biases in the representation of the particle mixture and the other absorbing particles (e.g.,
- 1023 black carbon, brown carbon and aged dust). With respect to the current model, this failure is
- 1024 related to the aerosol absorption represented in WRF-Chem. The leading absorption aerosol in
- 1025  $\,$  WRF-Chem is BC. The BC particle in the second size (0.156–0.625  $\mu m)$  had the maximum
- $1026 \qquad \text{absorption, according to Mie theory, and had the maximum DA modifications in the second-}$
- 1027 size bin (Figure 8e). However, because the BC had a small background concentration, the BC
- 1028 showed a small DA improvement (<1.5  $\mu$ g m-3) and had small effects on increasing the
- 1029 particle absorption. Meanwhile, the coarse dust particle concentration was primarily
- $1030 \qquad \text{increased, but the dust particles did not have a strong absorption as BC. As a result, the model}$
- 1031 lowered the ratio of the absorption coefficient of  $PM_{10}$  by an order of magnitude (Table 4).
- 1032 The lower mass absorption efficiency was comparable at each wavelength and was close to
- 1033 the other DA experiment (Figure 10c). Because of the constraint of the observed absorption
- 1034 coefficient, the DA system dramatically overestimated the particle concentrations and induced

1043 too much higher aerosol scattering coefficient and AOD, Because the overestimation of the 1044 scattering coefficient was higher than that of the absorption coefficient, DA abs even gave 1045 the strongest SSArf (0.9; Table 3) in all DA experiments, opposite to our expectation that the 1046 assimilation of absorption coefficient should improve SSA. 1047 1048 To understand the DA\_Eabs's failure, we performed a few trials by changing the imaginary 1049 part of the dust refractive index on 1200UTC on April 9. The results are present in the 1050 supplementary Table S4a and S4b. The trials show that a high imaginary part of the dust 1051 refractive index decreases the aerosol absorption coefficient. This paradox is due to the BC's 1052 reduction. Specifically, a high imaginary part increases the absorption efficiency of coarse\_ 1053 dust and decreases the coarse dust number concentration (num a04; Table S4a). This 1054 reduction also led to less fine aerosol number concentrations (e.g., num\_a02) because of the 1055 inter-size bin correlation. BC is abundant in the second and third size bins, and its imaginary 1056 part of refractive index is two orders of magnitude higher than dust. Less BC caused a weak 1057 absorption coefficient (Table S4b). On the contrary, the low dust imaginary part would not 1058 largely increase dust numbers in the coarse size bin because the DA system attempts to 1059 increase BC to enhance the absorption coefficient. In an extreme case with zero value of 1060 imaginary part of dust, the improvement of absorption coefficient exclusively relies on BC; 1061 the num a02 is increased by order of magnitude (Table S4a), and 660 nm Eabs rose up to 1062 92.5 Mm-1 (Table S4b), much higher than the observed level. 1063 1064 At Kashi, BC has a low background concentration and low background error. The innovation 1065 of BC was limited. Thus, tuning the imaginary part of dust would not change the SSAsrf 1066 value a lot (0.89 to 0.92). Excluding the contribution from OIN in PM10, the scattering 1067 coefficient was associated with sulfate. The sulfate's background error was higher than the 1068 BC's by order of magnitude. The DA system prioritized sulfate modification even when 1069 assimilating absorption coefficient, resulting in a smaller BC mass fraction in PM10 (Figure 1070 12f) and a high SSAsrf of 0.90. 1071 1072 We did another set of trials by increasing the original BC's BEC per size bin. As shown in the 1073 supplementary Table S5, increasing the BC's BECs would not much degrade the absorption 1074 coefficient but significantly decrease the positive biases in PMx, AOD, and scattering\_ 1075 coefficient; the SSAsrf approached the observation. Increasing BC's BECs by a factor of 1076 seven (DA Eabs BC\*7) shows the best analyses. This trial suppressed the positive biases 1077 without decreasing the accuracy of absorption coefficient (Figure 7), and the BC mass 1078 fraction was increased (Figure 12g). Nevertheless, the disadvantage of the enlargement of 1079 BC'BEC is that the simultaneous assimilation of scattering and absorption coefficient is not 1080 convergent as well as before. After four outer loops and each with 50 inner iterations, the 1081 analyzed absorption coefficient in DA Eabs BC\*7 was still higher than the observed value 1082 by 47% (Figure S1j). It indicates there is a low bias in BC's background concentration that

- 1083 violates the unbiased condition of DA.
- 1084
- 1085 3.6 Assimilating Multi-source Observations

[revised manuscript text omitted]

**1159**

**1160 4. Discussions**

- 1161 4.1 DA Impact on Aerosol Chemical Composition
- 1162 For control variable design, our DA system modifies the chemical composition of each
- 1163 aerosol according to the BEC values. The PM10 chemical fractions remain close to their 1164
- background values (Figure 12). As discussed in section 3.5, the assimilation of the aerosol 1165 absorption coefficient alone (DA Eabs) increased the sulfate fraction. The DA modification
- 166 increased aerosol number concentration, and the rising number concentration increased the
- 1167 tangent linear operator value for the scattering component. Sulfate was the predominant
- 1168 anthropogenic aerosol at Kashi and had a high background error value. The DA system
- 1169 prioritized the modification of sulfate and prevented a rise in the BC fraction in DA\_Eabs. As
- 1170 the enlarged BC BEC in DA\_Eabs\_BC\*7, the BC mass fraction showed the largest increase.
- 1171 The model bias in aerosol background concentration and the background error determine the
- 1172
- analyzed aerosol chemical fraction. Overall, it seems that differences in aerosol chemical 1173
- composition from assimilating the aerosol optical data are smaller than the difference in 1174 model setting (e.g., using other aerosol chemistry mechanisms, or using finer aerosol size
- 1175 bins). The assimilation of the total aerosol quantities cannot eliminate the intrinsic bias in
- 1176 aerosol composition. Thus, accurate aerosol chemistry and optical modules are crucial to
- 1177 attain a better background aerosol chemical data for DA analysis (Saide et al., 2020).
- 1178

1179

**Figure 12**

- 1180 4.2 DA Impact on Aerosol Direct Radiative Forcing
- 1181 Table 5 shows the instantaneous clear-sky ADRF in the background data and the analyses of
- 1182 DA\_PMx and DA\_PMx\_AOD. After the analyses, the DA effect (various DA frequencies for
- 1183 assimilating AOD and the surface particle concentrations) gradually faded away after
- 1184 restarting the model run. We therefore focused on the instantaneous radiative forcing values
- 1185 one hour after